# Observability of Moisture Transport Divergence in Arctic Atmospheric Rivers by Dropsondes

Henning Dorff[1,2], Heike Konow[3,1], Vera Schemann[4], and Felix Ament[1,3]

[1]University of Hamburg, Hamburg, Germany
[2]International Max Planck Research School on Earth System Modelling, Max Planck Institute for Meteorology, Hamburg, Germany
[3]Max Planck Institute for Meteorology, Hamburg, Germany
[4]University of Cologne, Cologne, Germany

**Correspondence:** henning.dorff@uni-hamburg.de

**Abstract.** This study emulates dropsondes to elucidate the extent to which sporadic airborne sondes adequately represent divergence of moisture transport in arctic Atmospheric Rivers (ARs). The convergence of vertically integrated moisture transport ($IVT$) plays a crucial role as it favours precipitation that significantly affects arctic sea ice properties. Long-range research aircraft can transect ARs and drop sondes to determine their $IVT$ divergence. In order to assess the representativeness of future sonde-based $IVT$ divergence in arctic ARs, we disentangle the sonde-based deviations from an ideal instantaneous $IVT$ divergence, which result from undersampling by a limited number of sondes and from the flight duration.

Our synthetic study uses CARRA reanalyses to set up an idealised scenario for airborne AR observations. For nine arctic spring ARs, we mimic flights transecting each AR in CARRA and emulate sonde-based $IVT$ representation by picking single vertical profiles. The emulation quantifies $IVT$ divergence observability by two approaches. First, sonde-based $IVT$ and its divergence are compared to the continuous $IVT$ interpolated onto the flight cross-section. The comparison specifies uncertainties of discrete sonde-based $IVT$ variability and divergence. Second, we determine how temporal AR evolution affects $IVT$ divergence values by contrasting time-propagating sonde-based values with the divergence based on instantaneous snapshots.

For our arctic AR cross-sections, we find that coherent wind and moisture variability contribute by less than 10 % to the total transport. Both quantities are uncorrelated to a great extent. Moisture turns out as the more variable quantity. We show that sounding spacing greater than 100 km results in errors greater than 10 % of the total $IVT$ along AR cross-sections. For $IVT$ divergence, the arctic ARs exhibit similar differences in moisture advection and mass convergence across the embedded front as mid-latitude ARs, but we identify moisture advection being dominant. Overall, we confirm the observability of $IVT$ divergence with an uncertainty of around 25–50 % using a sequence of at least seven sondes per cross-section. Rather than sonde undersampling, it is the temporal AR evolution over the flight duration that leads to high deviations in divergence components. In order to realise the estimation of $IVT$ divergence from dropsondes, flight planning should consider not only the sonde positioning, but also the minimisation of the flight duration. Our benchmarks quantify sonde-based uncertainties as an essential preparatory work for the upcoming airborne closure of the moisture budget in arctic ARs.

# 1 Introduction

Atmospheric Rivers (ARs), which are elongated ($> 2000\,\mathrm{km}$ in length) but narrow ($< 1000\,\mathrm{km}$ in width) water vapour rich corridors with strong moisture transport, occasionally enter the Arctic. Their occurrence accounts for roughly 70 % of poleward moisture transport (Nash et al., 2018). The presence of ARs can induce significant arctic warming (e.g. Neff et al., 2014) causing substantial sea-ice retreat (Woods and Caballero, 2016) as well as Greenland ice sheet melt (Mattingly et al., 2018; Neff, 2018). In addition to near-surface heat fluxes (Woods and Caballero, 2016; You et al., 2022), the melting arises from AR-induced precipitation (Mattingly et al., 2018; Viceto et al., 2022). Lauer et al. (2023b) identified ARs as one of the main contributors to overall arctic precipitation. For the required moisture of ARs impacting the Arctic, the North Atlantic to the south is a dominant uptake zone (Vázquez et al., 2018). Embedded in poleward moving cyclones and their warm conveyor belts (e.g. Dacre et al., 2019), the AR air masses can propagate meridional towards the Arctic Ocean and reach the sea-ice (Papritz et al., 2021), shaping the regional moisture patterns (Nygård et al., 2020). Along the long-distance displacement, the AR embedded moist and warm air masses are subject to transforming thermodynamic vertical structures (You et al., 2022). Komatsu et al. (2018) discovered an amplification of the air mass transformations after the ARs overpass the sea-ice edge.

To illuminate moisture transformation processes occurring in arctic ARs, it is crucial to grasp spatial characteristics of the moisture transport, i.e. the vertically Integrated Water Vapour Transport ($IVT$). For instance, Seager and Henderson (2013) point out that the divergence of $IVT$ links the local temporal development of moisture amount with precipitation formation. When we thus target to derive $IVT$ divergence in ARs, a prerequisite is to resolve the spatial variability of $IVT$. Guan and Waliser (2015) used global ECMWF Interim reanalysis (ERA-Interim, Dee et al., 2011) to investigate strong moisture transport gradients that exist along AR cross-sections, perpendicular to the major $IVT$ orientation. Along such lateral cross-sections through the AR centre, airborne observations in mid-latitude ARs have shown a bell-shaped $IVT$ distribution, having the strongest moisture transport in the AR core (e.g. Ralph et al., 2017; Demirdjian et al., 2020). Using ERA5, Cobb et al. (2021b) confirm the high spatial variability of $IVT$ for mid-latitude ARs. For the conditions in arctic ARs containing weaker moisture transport than in the mid-latitudes, we still lack quantitative knowledge of predominant $IVT$ variability and how this influences $IVT$ divergence.

High-resolution observations of $IVT$ variability are not available for arctic ARs. One reason is the remote and infrequent occurrence of arctic ARs over the ocean basins. Furthermore, the observation of moisture transport requires simultaneous measurements of winds and moisture for the entire troposphere. Radiosondes allow detailed insights of moisture transport profiles of arctic ARs at individual locations (e.g. Viceto et al., 2022), but their observation network in the Arctic is too sparse to obtain the divergence in single ARs (Dufour et al., 2016). Based on a similar principle, dropsondes released from research aircraft provide vertical profiles of relative humidity and wind speed with an accuracy of 1 % and 0.1 m/s, respectively (e.g. George et al., 2021; Konow et al., 2021). The simultaneous measurements allow derivation of moisture transport profiles and $IVT$. According to Zheng et al. (2021), dropsondes over ocean fill a data gap left by spaceborne remote sensing or ground-based observations. To derive the $IVT$ divergence in ARs, it is necessary to release the sondes at close spacing but over horizontally extended areas above the AR, which can only be achieved by long-range research aircraft (Neiman et al., 2014).

The overall goal of this study is to assess the observability of moisture transport divergence in arctic ARs by dropsondes. The assessment targets the facilitation of measurement strategies in dedicated research flights, as e.g. proposed by Wendisch et al. (2021). We include (a) the role of sonde frequency, (b) a concretisation of the need for supplementary measurements based on the spatial variability of moisture and wind, and c) the impact of extended flight duration under evolving AR conditions on the ability of the dropsondes to reproduce $IVT$ divergence in arctic ARs. The following paragraphs set these aspects into context of recent findings based on mid-latitude ARs and unravel four research gaps for arctic ARs, summarised as guiding questions. A limited number of sondes can cause deviations in the airborne representation of AR moisture transport variability if the sounding spacing becomes too coarse to reflect the spatial variability of $IVT$. Such deviations in $IVT$ variability come with misinterpretation of the $IVT$ divergence. For mid-latitude ARs, Guan et al. (2018) compared sonde-based Total Integrated Water Vapour Transport ($TIVT$), the integral of $IVT$ along an AR cross-section, with reanalysis-based $TIVT$ and found an agreement up to 3 % based on a mean sounding spacing of 80 km. Accurate airborne $TIVT$, juxtaposed for two separate AR cross-section legs, gives an initial estimate of $IVT$ divergence in between both legs. However, Ralph et al. (2017) found considerable sensitivity of sonde-based $TIVT$ to the spacing between the sondes. When doubling the initial sonde spacing, which averaged about 80 km, by reducing the number of sondes included, they found a mean deviation of at least 5 % for $TIVT$. Since given sensitivity studies refer purely to mid-latitude cases where we do expect higher $TIVT$ (Guan et al., 2018), it remains as open question: *What is the maximum distance between sondes to determine the total moisture transport in arctic AR cross-sections (Q1)?*

When assessing spatial $IVT$ variability in arctic ARs, it is crucial how moisture and wind fields coincide in the AR cross-section or whether they contribute independently to $IVT$ variability. For instance, in a polar AR case study, Terpstra et al. (2021) identified incoherent patterns of moisture and wind forming the moisture transport pattern, that are less aligned than in mid-latitude ARs. Unravelling moisture transport into wind and moisture can improve observational strategies of airborne moisture transport divergence in arctic ARs. Especially, if the moisture transport variability (and divergence) were e.g. mainly controlled by the moisture field, supplementary remote-sensing should be involved in the airborne representation of AR moisture. For this reason, it is important to determine whether moisture and wind are aligned in AR cross-sections and to ascertain: *How correlated are moisture and winds in arctic ARs and do coherent patterns contribute significantly to IVT (Q2)?*

Knowing the spatial structure of $IVT$ is a prerequisite for flight planning and reveals insights into the moisture transport divergence pattern in arctic AR cross-sections. Since ARs primarily occur at the interface of the cold front and warm conveyor belt in extratropical cyclones Dacre et al. (2019), different dynamic and thermodynamic processes act on the moisture transport and its divergence across the embedded front (Cobb et al., 2021a). For mid-latitude ARs, Cobb et al. (2021a) found significant differences in vertical moisture and wind for different sectors before and behind the front, which are reflected in gradients in the $IVT$ divergence across the front (Guan et al., 2020). Using reanalysis data, Guan et al. (2020) specified lateral differences of moisture transport divergence across centres of ARs. In the AR centre with maximum $IVT$, they identified the dynamical convergence of moisture as the most prominent component regulating moisture amount and precipitation. The Arctic is more affected by exit regions of ARs rather than over-passed by AR centres and exhibits weaker $IVT$ from ARs compared to the mid-latitudes (Guan and Waliser, 2019). ARs here commonly start dissipating and terminating (Guan et al., 2023). For such

conditions, we lack knowledge of predominant $IVT$ divergence. Thus, we examine: *How can the divergence of moisture transport be characterised along cross-sections of arctic ARs (Q3)?*

Comparing two reanalyses, Guan et al. (2020) found differences in $IVT$ divergence that reach up to 30 % the magnitude of

$IVT$ divergence itself in the AR centres. Norris et al. (2020) determined $IVT$ divergence in a mid-latitude AR from dropsondes that allows interpreting the discrepancies of $IVT$ divergence in ARs found by Guan et al. (2020). Norris et al. (2020) point to the large variability of $IVT$ divergence at spatial scales of 50 km. This also has implications for sonde-based sampling best practices. For arctic AR conditions, we lack equivalent estimates. Moreover, besides sonde-based undersampling of $IVT$ variability and divergence, we hypothesise that airborne results are impaired by the flight duration: Over the duration required

to enclose an AR area, the atmospheric state changes, i.e. there is relevant temporal evolution of the AR (its life cycle and spatial displacement) that causes the sonde-based values to deviate from the instantaneous $IVT$ divergence. Hence, before dropsondes are used to interpret the actual $IVT$ divergence in arctic ARs, we have to disentangle sonde-based errors that arise from undersampling by discrete sounding and from non-instantaneous sampling over the flight duration. We quantify: *To what extent can non-instantaneous sondes reproduce $IVT$ divergence in the light of AR evolution during flight (Q4)?*

To pursue Q1-Q4, we focus on ARs occurring over the Arctic Ocean (i.e. Fram Strait and Greenland Sea) in the vicinity of the sea-ice edge due to the above-mentioned AR impacts on the sea-ice. We restrict our analysis to ARs during spring, when maximum sea-ice extent starts melting. We look at arctic ARs within the novel C3S Arctic Regional Reanalysis (CARRA). Introducing a flight strategy to analyse moisture transport divergence in arctic AR flight corridors, we consider arctic AR events along synthetic flight tracks that transect an area of the AR. We emulate synthetic dropsondes along the tracks by de-

picting single vertical profiles. This study compares actual $IVT$ variability and divergence along the flight tracks with the emulated sonde-based representation of $IVT$ in order to assess how adequately such airborne perspectives reproduce predominant AR-$IVT$ characteristics. In a nutshell, our synthetic assessment provides benchmarks of sonde-based uncertainties in their representation of $IVT$ divergence in arctic ARs to facilitate future mission planning.

The manuscript is structured as follows: After introducing our AR cases, Section 2 describes the methods of emulating

dedicated flight patterns and synthetic soundings, and how we derive moisture transport divergence. For this framework, Section 3 deals with the $IVT$ variability. This entails the total moisture transport and $IVT$ variability along cross-sections in arctic ARs, their sonde-based representation (Q1) and the coherence of moisture and winds (Q2). Section 4 specifies the moisture transport divergence in arctic ARs (Q3) and compares its continuous representation to that by sporadic sondes. Section 5 quantifies airborne deviations arising from AR evolution over flight duration that is mostly idealised as stationary (Q4).

**2    Airborne derivation of moisture transport divergence in arctic ARs**

The central quantity of our study is the Integrated Water Vapour Transport ($IVT$) that represents the AR intensity as:

$$IVT = -\frac{1}{g} \cdot \int\limits_{p_{\mathrm{sfc}}}^{p_{\mathrm{top}}} q\mathbf{V} \, dp \tag{1}$$

**Table 1.** Specifications of used reanalyses for AR analysis

| Reanalysis Dataset | Horizontal Resolution | Vertical resolution up to 10 km ($\approx 250\,\mathrm{hPa}$) | Time resolution |
|---|---|---|---|
| ERA5 | 0.25 x 0.25 deg | 21 levels | hourly |
| CARRA | 2.5 x 2.5 km | 15 levels | hourly |

where $\mathbf{V}$ is the horizontal wind vector and $q$ the specific humidity. The divergence of $IVT$ represents a key component contributing to the overall atmospheric moisture budget. Following Seager and Henderson (2013), the vertically integrated moisture budget components consist of:

$$\underbrace{\frac{\delta IWV}{\delta t}}_{\text{local change in Integrated Water Vapour}} = \underbrace{E}_{\text{Evaporation}} - \underbrace{P}_{\text{Precipitation}} - \underbrace{\nabla IVT}_{\text{Divergence of Integrated Water Vapour Transport}}, \qquad (2)$$

with all components in kilogram per metre squared per second. Precipitation and evaporation refer to surface values, while the integrated water vapour $IWV$ and $IVT$ (Eq. 1) represent the vertically integrated quantities of moisture and moisture transport. Note that Eq. 2 neglects the moisture flux through a tilted bottom pressure surface that is included in Seager and Henderson (2013).

Given the relevance of $IVT$ to the AR moisture budget, this feasibility study targets the overall observability of $IVT$ and its divergence ($\nabla IVT$) in arctic ARs by airborne sondes in a synthetic way. For this purpose, this section introduces the reanalysis framework we use to investigate a presented selection of arctic ARs. In addition, our airborne flight strategy to derive $\nabla IVT$ in arctic ARs is specified and how we emulate the synthetic sondes in the reanalyses. Lastly, we describe the sonde-based derivation of $\nabla IVT$ and how we categorise different sectors across the AR front to examine the divergence.

## 2.1 Reanalysis framework

This study investigates arctic ARs in a reanalysis framework (Tab. 1). We use ECMWF Reanalysis v5 (ERA5) (Hersbach et al., 2020) to identify the AR events of our interest. ERA5 outperforms other global reanalyses with respect to AR characteristics (Graham et al., 2019; Cobb et al., 2021b). Thus, recent studies consider ERA5 to investigate AR conditions specifically in the Arctic (e.g. Fearon et al., 2021; Zhang et al., 2022). At the Fram Strait and Greenland Sea, the lat-lon grid from ERA5 yields approximately $30\,\mathrm{km}$ distances. Given the flight performance of long-range research aircraft (see Stevens et al., 2019), the spacing of airborne soundings in such a resolution would require releases every 2 minutes, which are more frequent than conducted in recent campaigns (e.g. Ralph et al., 2017). Still, Skamarock et al. (2014) emphasise that the effective model resolution is much greater than the model grid spacing. Since our study aims to assess the sub-grid scale variability of moisture transport between sonde releases from reanalyses, the effective resolution should be of the order of $\sim 10\,\mathrm{km}$ rather than of $\sim 100\,\mathrm{km}$.

Therefore, we further include the C3S Arctic Regional Reanalysis (CARRA). CARRA has a 2.5 km horizontal resolution over the entire domain and is accessible by Schyberg et al. (2021). Driven by lateral boundary conditions from ERA5, CARRA includes more observations and hourly forecasts by the HARMONIE-AROME model (Bengtsson et al., 2017). Køltzow et al. (2022) verified the improved representation of arctic surface-near meteorological conditions in CARRA, with decorrelation lengths of wind speed in better agreement to reference observations than ERA5.

Both reanalyses are provided on pressure levels by the Copernicus Climate Data Store (CDS). While ERA5 contains $IVT$ as output, we calculate $IVT$ in CARRA by the trapezoidal integral of moisture transport along the pressure levels (Tab. 1). In the following, we declare the high spatial resolution representation in CARRA as our synthetic reality of AR features.

## 2.2 Selection of Atmospheric River cases

The transformation of arctic air masses over changing surface types (open ocean and sea-ice) along large-scale meridional circulations is part of current research and investigated by research aircraft over the Arctic Ocean (Wendisch et al., 2021). For this reason, our study selects ARs causing air masses to overshoot the sea ice edge in this region. The principal identification of arctic AR events is based on the $IVT$-based AR detection catalogue by Guan (2022) applied to ERA5 (Lauer et al., 2023a). Among these ARs, we focus on spring, when maximum sea-ice extent in the Arctic Ocean starts to break-up and is more vulnerable to the intrusion of warm and moist air (Rostosky and Spreen, 2023). We restrict to selected events from the last decade, as the arctic climate has witnessed rapid and intense changes over the last decades (Wendisch et al., 2023). Our AR pathways originate from the North Atlantic and Barents Sea, that Papritz et al. (2021) spot as dominant regions for arctic moisture intrusions. The selection constrains on ARs whose lateral width is purely situated over open ocean or sea-ice. This ensures that we do not encounter orographic induced effects on $\nabla IVT$ which are out of the scope of this study. Moreover, airborne sonde releases over land are more complicated to conduct. Given the criteria above, we selected ARs from nine spring days between 2011 and 2020 (Fig. 1).

Greenland troughs are synoptic situations where low-pressure systems force large-scale meridional transport, with ARs potentially evolving on the eastern flank of the cyclone and reaching into the Arctic (Papritz and Dunn-Sigouin, 2020). Similarly, blocking situations over Eurasia can favour meridional circulation. For our nine ARs (Fig. 1), we confirm a large case-to-case synoptic variability. While some ARs (AR2, AR3, AR4, AR9) have evolved along the eastern flank of large-scale troughs over Greenland, AR5 and AR6 are more steered by blocking high pressure over the Barents Sea. AR1 and AR7 are, in turn, reinforced by a mesoscale cyclone situated over the Fram Strait and reach very close into the cyclone centre.

The synoptic compositions distribute the ARs over the North Atlantic and Arctic Ocean (Fig. 1), which correspond to the typical arctic moisture transport pathways identified in Papritz et al. (2021). Some ARs exhibit straight meridional moisture transport north of Iceland and approach or exceed Svalbard (AR1, AR2, AR3 in Fig. 1). AR4 and AR7 show more elongated filaments along the Norwegian coast, but still reach far north. We consider eight independent AR events, wherein AR5 is also considered for the consecutive day (AR6). At this stage, the centre of AR6 reaches close to the North Pole. AR8 originates from Siberia that, according to Komatsu et al. (2018), represents another significant roadway for arctic moisture intrusions favouring ARs. The last events in 2020 (AR8, AR9) are accompanied by a warm air intrusion period observed by the Multidisciplinary

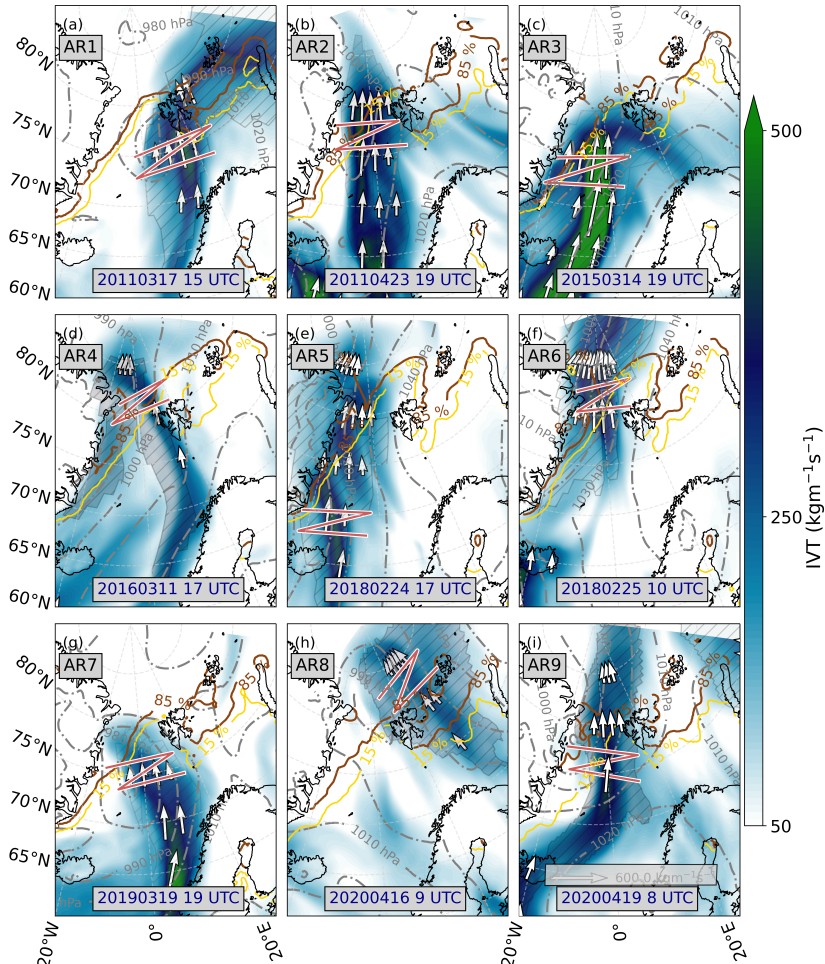

**Figure 1.** $IVT$ contours of investigated AR events from ERA5. Grey lines indicate surface isobars, while brown (orange) contour lines specify sea ice cover thresholds, given in %. The arrows depict the magnitude and orientation of $IVT$. Hatches represent the AR boundaries defined in Guan (2022). Red lines represent the zigzag flight pattern to investigate the moisture budget in AR flight corridors. Background maps were made with Natural Earth.

drifting Observatory for the Study of Arctic Climate (MOSAiC) expedition (Shupe et al., 2022), studied in Kirbus et al. (2023). Inspecting the vertical curtains of AR cross-sections, that are based on the southern red transects in Fig. 1, it becomes evident that the specific humidity exceeds $4\,\mathrm{g\,kg^{-1}}$ in almost every cross-section (Fig. 2). This indicates that our events are rather moist for arctic AR conditions (e.g. Viceto et al., 2022), but still much drier than mid-latitude ARs where $q$ easily exceeds $8\,\mathrm{g\,kg^{-1}}$ (Cobb et al., 2021a).

Figure 2 shows some of the features that we know about mid-latitude ARs. Over the course of various research flight campaigns over the eastern Pacific, Ralph et al. (2017) have developed a conceptual scheme for the cross-sections of ARs. This

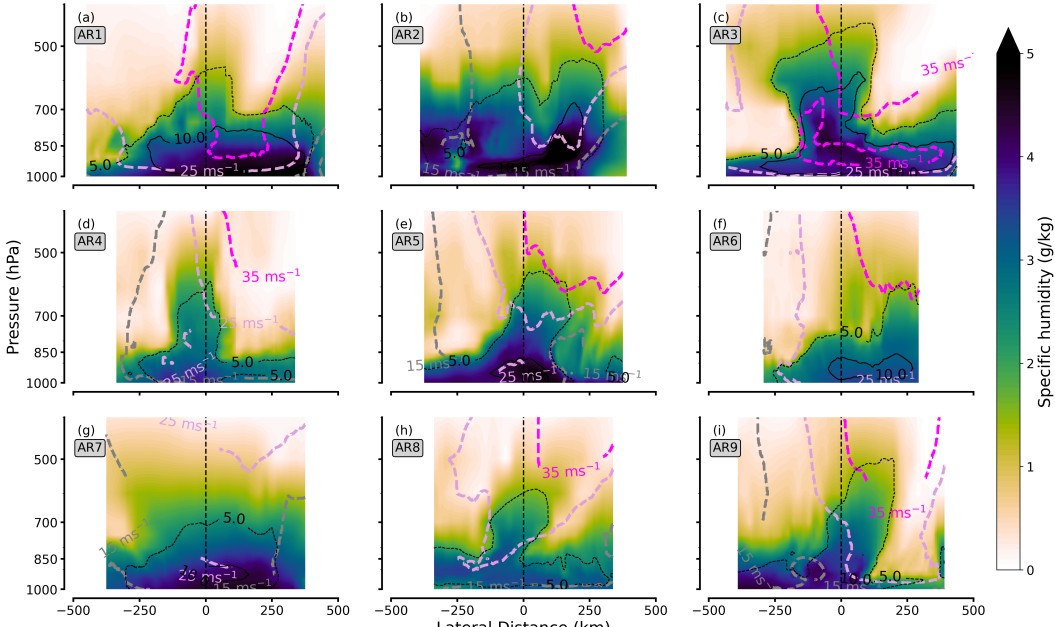

**Figure 2.** CARRA-based cross-sections of AR inflow legs for moisture (colour-coded contours), as well as contour lines of wind speed (pinkish) and moisture transport (black contour lines). The x-axes are oriented in flight direction, and values are given from eastern (negative) to western (positive) distances to the maximum $IVT$. Shown moisture transport values have the unit $\mathrm{g\,kg^{-1}\,m\,s^{-1}}$. As visible in Fig. 1, some ends of the cross-sections already reach out of the ARs, but this constitutes less than 10 % of the cross-section lengths.

scheme includes, for example, a low-level jet (LLJ), which is a strong low-level wind corridor (Ralph et al., 2004; Demirdjian et al., 2020). For the windy arctic AR events, e.g. AR3 and AR5, we detect the presence of LLJs stronger than $25\,\mathrm{m\,s^{-1}}$. The LLJ is located at a height of about 900 hPa, slightly lower than the mean height reported by Cobb et al. (2021a) for mid-latitude ARs. As another feature of the mid-latitude based AR cross-section scheme, Ralph et al. (2004) and Cordeira et al. (2013) found a horizontally slanted vertical structure of moisture transport from dropsondes and reanalyses. Ralph et al. (2017) confirmed the vertical interaction between the upper-level jet and the LLJ as a dominant effect for AR moisture transport. In Fig. 2, this is particular evident for AR5. Here, moist air masses residing in the cyclonic warm conveyor belt are lifted over the cold front sector. The downward intrusion of air from upper-level jets on the western flank causes the slanted structure of moisture transport.

In arctic ARs other than AR5, we find less agreement with the conceptual AR schemes. This is the case for the vertical structure of moisture, or the presence and the intensity of the LLJ, which is only strongly distinctive in AR1, AR3, AR5, AR7, but absent in all other cases. In some cases, e.g. AR9, the upper-level jet intrusion is accompanied by strong dry air subsidence that reinforces the slanting of the moisture transport pattern in the mid and lower levels. The variety of characteristics of winds, moisture, and its transport comes with the different synoptic patterns (such as troughs, ridges, smaller cyclones embedded in

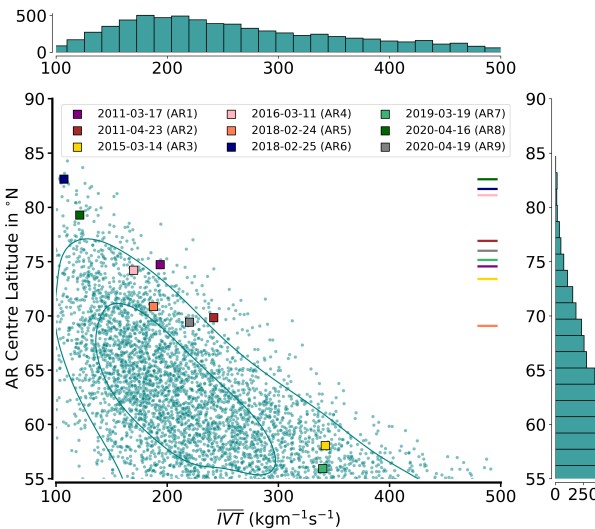

**Figure 3.** Comparison of selected AR events to long-term statistics (1979–2019) regarding mean $IVT$ and AR centre latitude at given reanalysis time step (ERA-Interim) based on the AR catalogue of Guan (2022). Isolines represent 25th and 75th percentiles of the kernel density estimate. Coloured lines on the right indicate the centre of the respective flight pattern. Histograms show absolute numbers of AR cases per bin.

a meridional, but weaker flow) that cause the arctic ARs. For example, we find the slanting most effective for ARs close to Eastern Greenland (AR2) or when the backside of the embedded cyclone strongly advects the dry Greenland air masses (AR9).

A caveat of our AR selection for making general statements about moisture transport variability in arctic ARs is the small sample size (nine events). Therefore, we place our events in the climatology for arctic ARs in spring. Using the entirety of spring ARs along the Atlantic pathway from the catalogue of Guan (2022), we compare the latitude of the AR centres and mean $IVT$ of our ARs with the long-term distribution (1979–2019) for spring ARs (Fig. 3). The climatological distribution in Fig. 3 indicates the decrease of mean $IVT$ with meridional location of the AR centre. Further towards the Arctic, ARs become weaker and when centred north of 65°N, the mean $IVT$ remains below $300\,\mathrm{kg\,m^{-1}\,s^{-1}}$. This is also the case for our AR events. However, for their specific latitude, the ARs are often characterized by higher $IVT$ compared to the long-term mean (Fig. 3). Only AR3 and AR7 are centred below 60°N, aligned with mean $IVT$ values around $350\,\mathrm{kg\,m^{-1}\,s^{-1}}$. Still, despite their southern centre, they reach far north with $IVT > 250\,\mathrm{kg\,m^{-1}\,s^{-1}}$ inside the Fram Strait (Fig. 1), so that we declare them as arctic ARs. We conclude from Fig. 3 that our cases represent the stronger AR events occurring in the Arctic.

## 2.3 Emulating flight patterns to sample ARs

To evaluate the airborne observability of $\nabla IVT$ within arctic ARs, we search for a suitable flight pattern. Such a pattern must capture certain areas of the AR well. Flight tracks that enclose such areas, like circles, best allow divergence calculations and are often used for such purposes (e.g. Bony and Stevens, 2019). However, ARs exhibit cross-frontal heterogeneity in

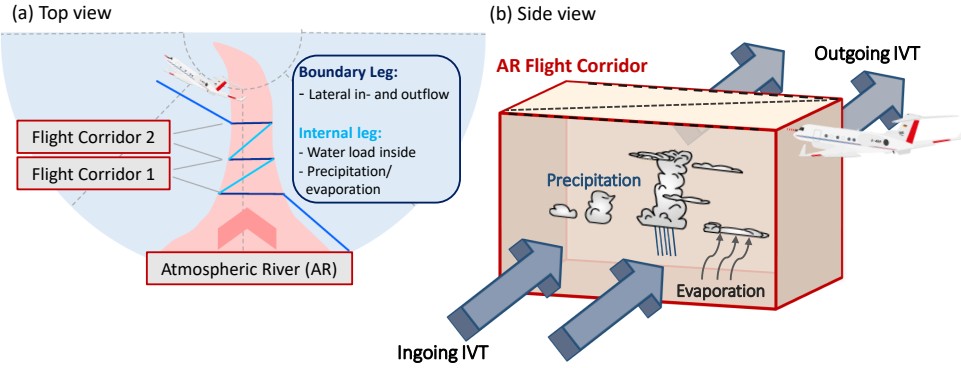

**Figure 4.** Top (a) and side (b) view on envisioned zigzag flight pattern to derive the moisture budget components inside AR flight corridors.

moisture and wind fields (Cobb et al., 2021a) that would smooth out in single circles. Instead, the high lateral variability in AR
moisture transport characteristics requires long flight legs across the AR filament to better capture divergence heterogeneity.
Two parallel cross-sections can be connected via an internal flight leg, resulting in a zigzag flight pattern (Fig. 4a). The zigzag
pattern observes the AR in flight corridors across its transport direction. For the sake of brevity, we speak of an AR flight
corridor in the following, when we mean the area of the AR that is captured by the flight pattern. The boundary cross-section
legs perpendicular to the major flow focus on quantifying the AR flight corridor moisture in- and outflow, i.e. in- and outgoing
$IVT$ over the entire lateral AR extension and enable simplified divergence calculations. Note that the diagonal internal legs
can focus on assessing precipitation rate, evaporation, or water load inside the AR flight corridor so that this pattern allows
quantifying the remaining moisture budget components of the budget box, i.e. the AR flight corridor (Fig. 4b).

We place the AR flight corridors close to the sea ice edge (Fig. 1). We orientate the cross-section legs orthogonal to the
major $IVT$ direction and extend the legs such that they transect the entire AR. One requirement is that the transect, and thus
the lateral AR extension, is completely over open ocean or sea ice, so that we neglect landfalling regions of ARs. To obtain
the spatial extension of the AR of a given case, we consider the shapes of the AR defined in the AR catalogue of Guan (2022).
The meridional distance between both cross-section legs is assured to be larger than 200 km, but closer than half the lateral AR
width. The decision for the placement and the meridional distance of the cross-sections is based on visual inspection of the
reanalysis taking in particular the movement of the movement of IVT filament into account. Due to their proximity to the sea
ice edge, the transects of AR flight corridors are mainly north of AR centres (horizontal lines in Fig. 3).

We mimic the airborne AR representation by the flight performance of state-of-the-art long-range research aircraft. We
refer to the High Altitude and LOng Range Research Aircraft (HALO), equipped with dropsondes and a remote sensing
configuration (Stevens et al., 2019; Konow et al., 2021); a similar aircraft like the one examining pacific ARs, specified in
Cobb et al. (2022). Both aircraft allow along-track observations during flight at common cruising levels above 10 km and a
ground speed of around 250 m/s. Accordingly, we idealize flights with a constant ground speed of 250 m s$^{-1}$ but neglect the
duration for turns and choose a constant flight altitude of 10 km. Based on the aforementioned flight performance, the aircraft

location is represented in a 1 Hz resolution, as common remote-sensing products on research aircraft (e.g. Mech et al., 2014; Konow et al., 2019), that can complement dropsonde data, have a comparable time- resolution and require the information of the aircraft location. The zigzag patterns (Fig. 1) require roughly 2–3 h to be flown, and up to 1 h for single AR cross-sections.

Using this idealised flight performance, up to three reanalysis time steps represent atmospheric conditions during the flights. We upsample the reanalysis data to minutely frequency by linear time interpolation. For our 1 Hz representation of flight location, we depict the reanalysis values from the nearest minute and spatially interpolate them along the flight using haversine distances. This spatio-temporally interpolated representation of meteorological values and AR characteristics along the flight will from now on be referred to as "continuous AR representation". We declare the interpolation as a suitable estimate of

airborne atmospheric observations in dynamic systems like ARs that are subject to significant spatial displacement.

## 2.4   Divergence derived from synthetic sondes

We synthetically refer to the measurement principle of dropsondes (Sect. 1). Along the continuous airborne AR representation of the cross-sections (Sect. 2.3), we depict profiles as synthetic soundings for which we neglect any vertical drift or fall time. We also neglect any measurement uncertainties. Such effects are out of the scope of this study, and assumed to cause lower

deviations than our considered effects. The synthetic sondes profile the atmosphere fully-vertically from their release location. $IVT$ is thus defined as the integral of the fixed vertical column from the respective reanalysis cells. Instead, we focus on the spatial representativeness of sporadic sonde-based $IVT$ and evaluate the uncertainties in the lateral variability of moisture transport, and how these uncertainties affect the airborne non-instantaneous perspective on $IVT$ divergence in arctic ARs.

    To derive $\nabla IVT$ in AR flight corridors from sondes, we compare two approaches. The first one is a simplified approximation

based on the derivation of the Total Integrated Water Vapour Transport ($TIVT$) of the two cross-section legs in Fig. 4. Ralph et al. (2017) defines $TIVT$ of a cross-section as:

$$TIVT = \int IVT \, dx, \tag{3}$$

representing the lateral integral of $IVT$ over the flight distance $x$ in a respective cross-section flight leg. Neglecting the moisture flux apart from perpendicular to the flight track, i.e. missing fluxes across the eastern and western boundaries, we can

approximate $\nabla IVT$ within an AR flight corridor by the difference of out- minus ingoing $TIVT$ of the cross-sections. Only if the lateral flow can confidently be neglected, we can obtain the divergence appropriately. Given this limitation, Lenschow et al. (2007) alternatively suggests the regression method, which marks our second approach. Under linear variations, a meteorological quantity $\Phi$ (e.g. wind speed) that is stationary in time can be inferred as:

$$\Phi = \Phi_o + \frac{\delta \Phi}{\delta x} \cdot \Delta x + \frac{\delta \Phi}{\delta y} \cdot \Delta y, \tag{4}$$

with the area mean value $\Phi_o$ and $\Delta x$ and $\Delta y$ being zonal, meridional displacements from the area centre point. Using the values of $\Phi$ at sounding locations and minimising the least-squared errors in the linear regression fit of Eq. 4, we obtain a linear estimate of zonal (x) and meridional (y) gradients, along with the mean mesoscale value for $\Phi$. Adding up both gradients, we calculate the divergence. Bony and Stevens (2019) and George et al. (2021) proved the feasibility of this method by comparing its divergence values with the Gaussian-based line integral over flown circles.

Having the mathematical expression, we view on the impact of $IVT$ divergence. The divergence of moisture transport can be split up into two components:

$$\nabla IVT = -\frac{1}{g \cdot \rho_w} \cdot \int_{p_{\text{sfc}}}^{p_{\text{top}}} \nabla(q\mathbf{V})\,dp = \underbrace{\frac{1}{g \cdot \rho_w} \cdot \int_{p_{\text{sfc}}}^{p_{\text{top}}} q(-\nabla\mathbf{V})\,dp}_{\text{dynamical mass convergence (CONV)}} + \underbrace{\frac{1}{g \cdot \rho_w} \cdot \int_{p_{\text{sfc}}}^{p_{\text{top}}} \mathbf{V}(-\nabla q)\,dp}_{\text{integral of horizontal moisture advection (ADV)}} \quad . \tag{5}$$

The first term represents the dynamical mass convergence, being the product of the moisture mass and divergence. The mass convergence term can be related to vertical velocity via the continuity equation and itself is closely linked to precipitation (Wong et al., 2016; Norris et al., 2020). The second term represents the horizontal advection of moisture that Guan et al. (2020) shows to be little correlated to precipitation formation. Instead, it locally affects the amount of water vapour. To calculate $ADV$ and $CONV$, we use the regression method. Finally, all terms in Eq. 5 are divided by the density of water $\rho_w$ to provide their contributions to the moisture budget (Eq. 2) in $\text{mm}\,\text{h}^{-1}$.

## 2.5 AR sectors and sonde locations

Research considering $IVT$ divergence in ARs suggests distinguishing between different sectors along the lateral AR cross-sections. Guan et al. (2020) highlight that different dynamics take place across the cold-frontal structures that are commonly embedded in the AR, which itself is situated at the western end of warm conveyor belts (Dacre et al., 2019). Hence, Guan et al. (2020) separate $IVT$ divergence calculations across the major AR axis and the AR embedded front. Similarly, Cobb et al. (2021a) classified different sectors in ARs based on the position of the AR embedded cold front and on the $IVT$ shape of airborne observations of a large set of pacific AR cross-sections. Both approaches distinguish between frontal sectors, namely a pre-frontal (warm) sector, the AR core with highest $IVT$, near which the cold front is expected (Ralph et al., 2017), and the post-frontal (cold) sector behind the cold front. Since there exist significant differences in moisture transport divergence between the sectors (Guan et al., 2020), a large part of the variability is smoothed out when calculating $IVT$ divergence for entire cross-sections.

Accordingly, we conduct a similar sector-based decomposition of $IVT$ divergence for our arctic AR events in CARRA. As in Guan et al. (2020) and Cobb et al. (2021a), our decomposition relies on the $IVT$ characteristics along the cross-section, which we depict for an example AR cross-section in Fig. 5. The central AR core represents the region of strongest $IVT$, which is more than 80 % of maximum $IVT$ ($IVT_{\text{max}}$). Following Ralph et al. (2017), we expect the cold front in the vicinity of the AR core. We denote the region east of the core as the pre-frontal sector containing warm air masses, and west of the core as the post-frontal sector that reaches out of the cold front in colder air masses. Since we focus on the AR relevant regions with high $IVT$, we restrict the outer extents of both extra-frontal sectors. For both sectors, we assign the outer edges where $IVT \leq 0.33 \cdot IVT_{\text{max}}$ to account for case-specific relative values (Fig. 5). As a secondary absolute threshold, we declare a moisture transport with $IVT \leq 100\,\text{kg}\,\text{m}^{-1}\,\text{s}^{-1}$ as too weak to be assigned as AR. This threshold to define the AR edges follows the approach of Cobb et al. (2021a). However, we lower their mid-latitude based $IVT$ threshold from 250 to 100

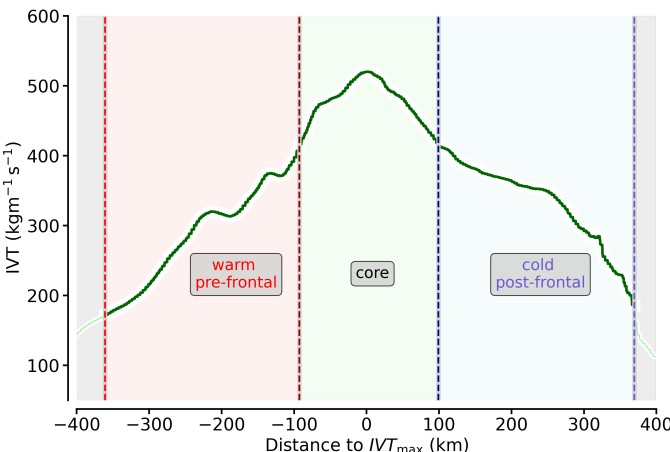

**Figure 5.** Sector decomposition for an example $IVT$ cross-section (AR1) using the criteria described in Sect. 2.5. The coloured shadings and text boxes indicate each frontal sector. The grey shading on the left represents moisture transport (i.e. $IVT$) that is not considered as AR because it is too weak. The orientation of the x-axis is in flight direction, from east to west.

kg m$^{-1}$ s$^{-1}$. By this, we refer to common polar moisture transport magnitudes that exceed the 95th percentile of climatology and are declared as ARs in the detection of Guan and Waliser (2015). Otherwise, we would either exclude most ARs north of 70°N or would shrink the AR cross-section so much that most transport is ignored, as the statistics in Fig. 3 indicate. Both thresholds form the outer boundaries of the AR and of the pre-frontal and post-frontal sector. Note that the sector terminologies are only a generalised categorisation of the surrounding air masses in the vicinity of the cold front inside ARs, but should not
be viewed as a synoptic cold front identification. All threshold criteria are applied to the continuous AR representation along the flight time, as in a realistic post-analysis from real research flights.

Using seven synthetic sondes per cross-section of the AR, we locate the sondes along the flight time in a way that three sondes each in the in- and outflow cross-section span one out of the three pre-defined frontal sectors (Fig. 6), and calculate its $IVT$ divergence, respectively. Note there is probably more variation in the actual release position in real flights due to forecast
uncertainties, even when the releases are planned using the threshold criteria based on forecasted $IVT$. However, we here stick to these pre-defined locations for comparability among the AR cases. Given varying sector lengths, the sonde spacing in Fig. 6 is not equidistant. Additionally, the comparison to the $IVT$ contours in Fig. 6 reveals that the sondes do not lie at the sector boundaries for the intermediate reanalysis time output. Since our $IVT$-based AR sectors and sonde positions are defined for each airborne cross-section representation individually, they do not refer to $IVT$ conditions at an instantaneous time step. In
fact, there is a north-eastward displacement of the AR filament over the course of the 2.5 h synthetic flight pattern. For this reason, the sectors along the flight track in Fig. 6 tilt while the internal $IVT$ has a northward orientation. Returning to this, Sect. 5 examines the extent to which the sonde-based $IVT$ divergence is affected by the flight duration, as opposed to an actual view on the AR in an instantaneous snapshot.

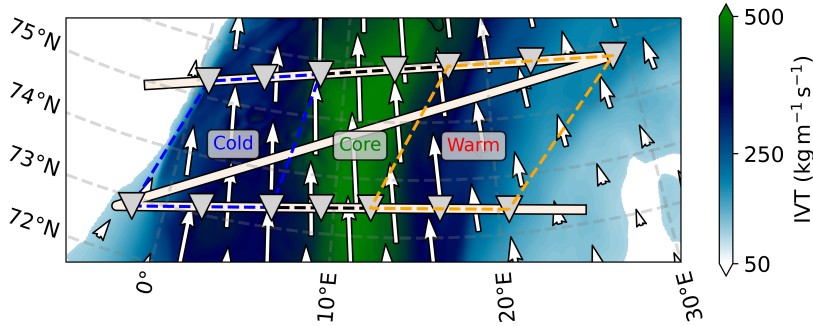

**Figure 6.** Illustration of AR cross-section sectors and placed sondes to calculate the divergence for AR1. $IVT$ contours refer to CARRA at the mid-hour of the flight corridor. Dashed lines connect the sonde sectors. Background map made with Natural Earth.

## 3 Moisture transport in arctic AR cross-sections and its variability

To examine the moisture transport variability in arctic ARs, we follow a two-fold approach. First, we stick to the vertically-integrated perspective and determine the maximum distance between sondes, needed to derive the total $IVT$ in AR cross-sections accurately (Q1). The synthetic soundings assess the observability of total moisture transport by discrete sondes. Since we lack real observations, we declare the spatio-temporally interpolated AR representation in CARRA as the truth. Second, we analyse to what extent coherent patterns in moisture and wind speed anomalies contribute to moisture transport and its variability (Q2). It is crucial to link the results to the large case-to-case variability with respect to $IVT$ magnitude (Fig. 1) and the vertical moisture and wind fields (Fig. 2).

### 3.1 $IVT$ shape across ARs

We investigate whether arctic ARs feature the same bell-shaped structure of $IVT$ along their cross-sections as observed in mid-latitudes (Ralph et al., 2017), and to what extent sondes can reproduce this structure. For AR1, Figure 7 illustrates the cross-section $IVT$ along the inflow flight leg. We recognise the bell-shaped $IVT$ from both, CARRA and ERA5. Within the cross-section centre which we declare as the AR core in Sect. 2.5, CARRA shows stronger moisture transport with a more pronounced $IVT$ maximum ($IVT_{\mathrm{max}}$) $> 500\,\mathrm{kg\,m^{-1}s^{-1}}$ than ERA5. Moreover, CARRA resolves more small-scale structures of the AR moisture transport. In particular, CARRA increases the cross-section variability for this case.

Most of the arctic AR cross-sections show this typical bell-shaped $IVT$ curve over widths of roughly 400–800 km and indicate pronounced $IVT$ maxima of 300–600 kg m$^{-1}$s$^{-1}$ in the core (not shown). We find that the arctic ARs are not substantially narrower than the AR widths of global climatology (Guan and Waliser, 2015) and observed mid-latitudes events (Cobb et al., 2021a). Flight planning should therefore consider cross-section distances of about 500-1000 km, similar to mid-latitude ARs. However, this only applies if the legs are not restricted to regions with $IVT > 250\,\mathrm{kg\,m^{-1}s^{-1}}$, which is a widely used threshold for mid-latitude ARs (e. g. Ralph et al., 2019). In contrast, the maximum $IVT$ for the arctic events is roughly half as high as the majority of mid-latitude ARs from airborne studies in Cobb et al. (2021a). Moreover, the $IVT$ magnitudes strongly differ

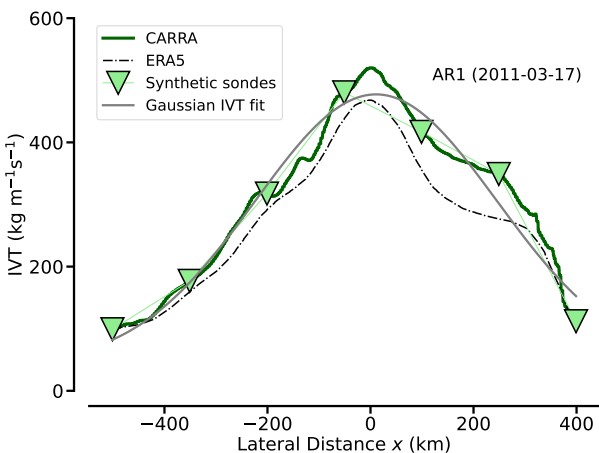

**Figure 7.** Inflow $IVT$ cross-section from AR1 (2011-03-17), comparing CARRA and ERA5 with seven synthetic soundings placed along the track. A Gaussian fit based on the sounding $IVT$ representation is calculated (grey).

between our cases and synoptic conditions. The strongest ARs, with $IVT_{max}$ exceeding $500\,\mathrm{kg\,m^{-1}s^{-1}}$ are found for intense Greenland troughs, while the ARs are weaker along the Siberian pathway (see also Fig. 1). In comparison to other arctic case studies (e. g. Viceto et al., 2022), we consider rather strong ARs.

Viceto et al. (2022) documented the improved representation of arctic AR characteristics in ERA5 against coarser reanalysis 350 data. In our comparison of CARRA with ERA5, the location and horizontal patterns of the ARs match quite well (not shown). For all cross-sections, we ascertain plausible $IVT$ values from CARRA with respect to ERA5. We highlight that maximum (mean) values of $IVT$ per cross-section increase by roughly 9 % (8 %) from ERA5 to CARRA on average. CARRA thus increases $IVT$ variability by about 11 %. We attribute this to the higher horizontal resolution of CARRA. The increased $IVT$ variability supports our treatment of CARRA as ground truth, before dedicated observations will be conducted.

When the observational focus is on the $IVT$ variability, in general, not on sector-based characteristics, we can simply place the seven sondes equidistantly. From this, a Gaussian fit can reproduce the bell-shaped AR-$IVT$ cross-section (Fig. 7). However, the Gaussian fit is very sensitive to the actual positions of dropsonde releases. While the centred sonde in Fig. 7 is positioned close to $IVT_{max}$, a slight shift of this sounding, which easily occurs in real observations, can quickly lead to an underestimation of the moisture transport in the AR core. Flight planning should thus imply a sonde release in the vicinity of 360 the predicted $IVT$ maximum and place additional sondes symmetrically around the core.

### 3.2 Sonde-based total cross-section moisture transport

The accuracy in sonde-based $TIVT$ of an AR cross-section depends on the number of sondes across the AR, i.e. their spacing (Ralph et al., 2017). Larger spacing of sondes affects the derived total moisture transport of AR cross-sections, whereby the sonde location becomes increasingly relevant. For example, the sondes shown in Fig. 7 for AR1, underestimates $TIVT$ by 365 more than 5 % against the continuous $IVT$ representation. Using CARRA for all of our ARs, we assess the sounding spacing,

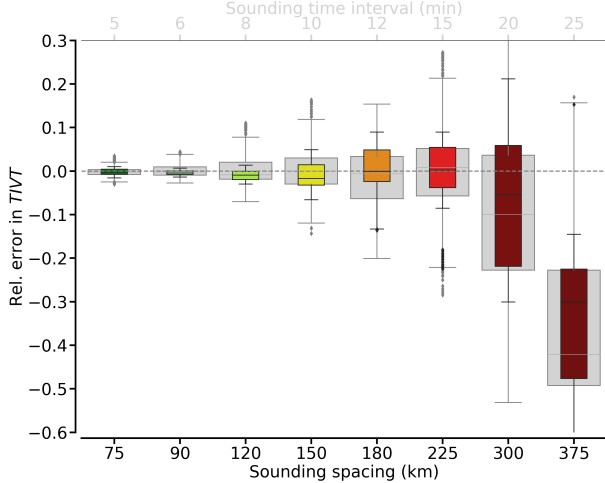

**Figure 8.** Relative error in $TIVT$ as a function of sounding spacing in km for all AR cross-section representations (grey) and those including highest (75th percentile) IVT maxima (coloured). Statistics rely on the boots-trapping approach, containing of 100 cross-section sonde representations per AR. The boxes show the quartiles, while the whiskers show the rest of the distribution, except for outliers (markers). For an assumed aircraft speed of $250\,\mathrm{m\,s^{-1}}$, equivalent release intervals are given on the top x-axis.

needed to gain adequate $TIVT$ estimates, by varying the density of synthetic sondes and by comparing their $TIVT$ values to a control case which is based on the continuous representation of the AR cross-sections along the flight (Sect. 2.3). To account for the dependency on sounding location, we conduct a bootstrapping approach in which we sample the cross-sections with varying release spacing and varying sounding locations. From this, the grey box-whiskers in Fig. 8, showing the distribution
of sonde-based errors of $TIVT$, reveal that the relative error of $TIVT$ against the continuous AR representation increases significantly for sounding spacing $\geq 150\,\mathrm{km}$. This corresponds to roughly five sondes for the given cross-section lengths, and release intervals above $10\,\mathrm{min}$ at a cruising speed of $250\,\mathrm{m\,s^{-1}}$. For sonde spacing $\geq 200\,\mathrm{km}$, sonde-based $TIVT$ can substantially deviate.

The $TIVT$ uncertainty in Fig. 8 increases less rapidly with sonde spacing than derived for mid-latitude AR cases (see Ralph
et al., 2017 and Guan et al., 2018). Total moisture transport in the arctic cases is, in turn, much smaller than in mid-latitude cases. Our arctic $TIVT$ values are roughly half as high as the sonde-based mean $TIVT$ of $5 \cdot 10^8\,\mathrm{kg\,s^{-1}}$ ascertained by Ralph et al. (2017) from 21 mid-latitude ARs. Our ARs have roughly two third the width of the ARs in Ralph et al. (2017) and Guan et al. (2018). Still, we remind that our $IVT$ threshold, scaling the outer edges of the AR, is much lower. Using mid-latitude based thresholds (Sect. 2.5), mean AR widths would be in the range of a few hundred kilometres and $TIVT$s even lower.
With increasing spacing, the spread in $TIVT$ errors in Fig. 8 increases, mainly due to the rising relevance of the sonde position. Too large sonde spacing enhances the likelihood that the sampling will not capture the region of strongest $IVT$. Especially with the occurrence of a LLJ, Guan and Waliser (2017) confirm that the AR core, alone, accounts for $\approx 50\,\%$ of the entire moisture transport. Yet, we also attribute the spread in relative $TIVT$ errors to the large AR case-to-case variability in

maximum $IVT$. In particular, the strong correlation of $IVT_{\max}$ to $IVT$ variability (correlation coefficient $r = 0.91$) may cause
sonde errors in derived $TIVT$. Hence, we expect that the smallest sonde spacing is required for the strongest AR events. The
coloured box-whiskers in Fig. 8 show that the distribution of the relative $TIVT$ error behaves similarly, when we only include
the cross-section representations with $IVT_{\max}$ larger than the 75th percentile from the bootstrapping sample. The mean relative
error increases more rapidly with spacing, while the inter-case spread is slightly lower than in the entire sample.

We conclude that highest $TIVT$ errors thus do not originate from the strongest events when having very few sondes. Still,
we emphasise that a minimum sounding spacing of 100-150 km has to be targeted for arctic ARs, which is less than mean
sonde spacing of $\approx 80$ km as conducted for mid-latitude ARs in Ralph et al. (2017).

### 3.3 Variability of moisture and wind in arctic ARs

To address (Q2), Fig. 9 a,b) summarise the cross-section variability of $v$ and $q$ over the vertical profile for all arctic ARs.
Moisture transport in the lowest levels up to 850 hPa is strongest and accounts for 50 % of the total $IVT$ (Fig. 9c). Up to this
height, both high moisture and wind speeds are predominant, with a local maximum of wind speeds around 900 hPa. Further
upwards, wind speed accelerates up to 20–40 m s$^{-1}$, while moisture decreases. The decrease of moisture with altitude leads to
a decline of moisture transport. Through the entire troposphere, $q$ is always below 5 g kg$^{-1}$ in our arctic ARs.

The vertical moisture characteristics (Fig. 9b) agree with the study by Viceto et al. (2022) who documented $q$ values up
to 5 g kg$^{-1}$ in soundings of arctic early summer ARs at Ny-Alesund. However, the winds in our AR cross-sections (Fig. 9a)
are roughly twice as strong as in their case study, which reports an orographic deceleration by Svalbard. For our arctic ARs,
the open ocean enables stronger winds, rather comparable to the wind conditions in the mid-latitude ARs. There, Ralph et al.
(2004) and Cobb et al. (2021b) report on mean low-level wind speeds from 10–25 m s$^{-1}$ for a large set of ARs over North-East
Pacific. The slight local wind maximum at 900 hPa (Fig. 9a) arises from the presence of LLJs (see also Fig. 2). Above the local
wind maximum, the increase of wind speed with height is less than found in sub-tropic/mid-latitude cases. Ralph et al. (2005)
and Cobb et al. (2021a) report on a stronger increase with height due to a more intense upper level jet.

The cross-section variability of both moisture and winds strongly affects moisture transport variability. The shadings in Fig.
9c) indicate that the horizontal standard deviation of moisture transport resembles the standard deviation of the winds for the
lower levels up to 850 hPa. At higher levels, moisture transport variability is driven by the standard deviation of moisture, and
the intrusion of dry air masses on the western flank (see Fig. 2), even though the wind standard deviation becomes highest
above 500 hPa. For example, in the strongest AR (AR3; Fig. 9), the LLJ exhibits high wind speeds above 30 m s$^{-1}$ that cause
strong moisture transport, whereas moisture is more or less average. While strong moisture transport in AR3 originates from
overall strong winds, moisture varies strongly and seemingly dominates the moisture transport variability (Fig. 9b). Hence, we
hypothesise that in strong arctic ARs with intense winds, moisture variability primarily steers $IVT$ variability and leads to the
bell-shaped $IVT$ cross-section pattern depicted in Sect. 3.1.

The identification of the more variable quantity can improve measurement strategies. Specifically, in case of a moisture
dominance, the ability of supplementary remote sensing devices from the research aircraft to derive moisture fields could
be explored. To this end, we quantify the relative standard deviations of wind and moisture ($s_q$ and $s_v$), normalised by the

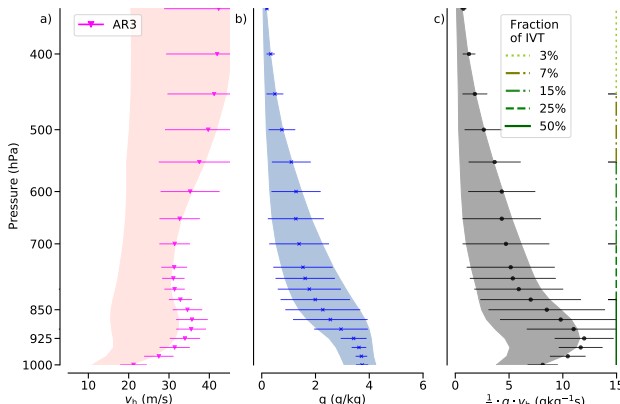

**Figure 9.** Vertical statistics of wind speed (a), specific humidity (b) and moisture transport (c) from the inflow cross-sections of the nine ARs analysed. Shaded areas represent the overall mean values $\pm$ the standard deviation. The error bars depict this distribution for the strongest AR (AR3). Vertical lines in c) specify the cumulative contribution of moisture transport to $IVT$ down to the given levels.

horizontal mean. We investigate $s_q$ and $s_v$ as a function of height (Fig. 10). Especially for the winds, the relative standard deviation in Fig. 10 remains rather constant throughout the troposphere and has a small magnitude of 20 to 35 %. Besides a

weak local maximum in the vicinity of the LLJ, the variability increases slightly near the upper-level polar jet, but with minor impact on the moisture transport (variability) due to dry air masses. Correspondingly, the wind contours of Fig. 2 indicate stronger horizontal gradients above 500 hPa, while moisture transport is already minor.

The variability of moisture behaves differently. In the boundary layer, moisture variability is negligible, similar to wind ($s_q, s_v < 20$ %). Yet, the decline of mean moisture with height is opposed by an increase of its relative variability. Between 600

and 850 hPa, high moisture variability ($s_q$ up to 50 % and more) contributes significantly to mean moisture transport variability. Based on our AR cross-sections, we conclude that moisture represents the more variable quantity in arctic ARs, which was already indicated in Fig. 2 by the prominent moisture plumes.

### 3.4 Coherence of moisture and wind

For the moisture transport, it is not merely important whether moisture and wind anomalies exist separately (Sect. 3.3), but

also how correlated they evolve along the AR cross-sections and whether such coherent patterns contribute significantly to AR-$IVT$ (Q2). If both patterns are coherent, carefully collocated observations are essential to determine $TIVT$, otherwise independent estimates of mean moisture and wind are sufficient. Therefore, we decompose the overall moisture transport $\overline{q \cdot v}$ that basically is a combination of transport by the mean quantities $\overline{q}$ and $\overline{v}$ and their cross-section variability, i.e. the spatial fluctuations $q'$ and $v'$, according to:

$$\overline{q \cdot v} = \overline{(\overline{q} + q')(\overline{v} + v')} = \overline{q} \cdot \overline{v} + \underbrace{\overline{q'\overline{v}}}_{=0} + \underbrace{\overline{\overline{q}v'}}_{=0} + \underbrace{\overline{q' \cdot v'}}_{\mathrm{cov}(q,v)} .$$
(6)

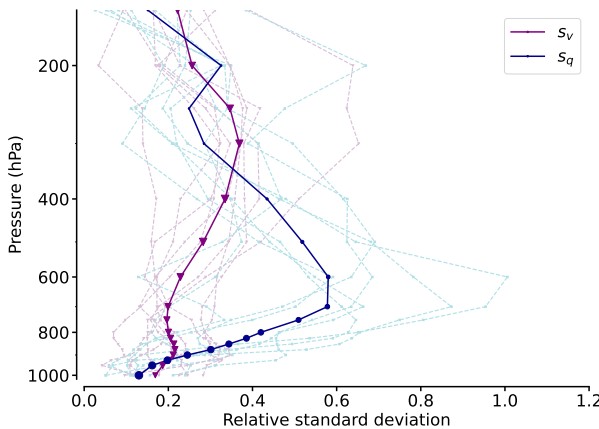

**Figure 10.** Vertical profile of relative standard deviation of wind ($s_v$) and moisture ($s_q$) for the AR cross-sections of each flight. The bold lines indicate the mean value over all ARs for both components. The sizes of the dots are scaled by the mean value at this height normed by the maximum mean value for the entire profile.

While the second and third summand equal zero, the last term represents the covariance cov() between $q$ and $v$. Using the relation between correlation coefficient $r_{corr}$ and cov(), we obtain:

$$\mathrm{cov}(q,v) = r_{corr}(q,v) \cdot \mathrm{std}(q) \cdot \mathrm{std(v)}, \tag{7}$$

and expanding by $\overline{q}$ and $\overline{v}$, we can reformulate Eq. 6 as:

$$\overline{q \cdot v} = \overline{q} \cdot \overline{v} \cdot \big( r_{corr}(q,v) \cdot \underbrace{\frac{\mathrm{std}(q)}{\overline{q}}}_{s_q} \cdot \underbrace{\frac{\mathrm{std}(v)}{\overline{v}}}_{s_v} + 1 \big). \tag{8}$$
$$\underbrace{\phantom{r_{corr}(q,v) \cdot \frac{\mathrm{std}(q)}{\overline{q}} \cdot \frac{\mathrm{std}(v)}{\overline{v}}}}_{cov_{norm}}$$

The normalised covariance $cov_{norm}$ (left summand in Eq. 8) weighs the coherent transport by fluctuations relative to the transport of mean quantities $\overline{q} \cdot \overline{v}$. Using $cov_{norm}$, Fig. 11 finds rather little coherence between moisture and wind in arctic AR cross-sections. The magnitude of the contribution of moisture transport variability to the overall moisture transport is below $\pm 10\,\%$ for each height. Assuming that CARRA resolves the scales of dominant fluctuations, the coherence is of minor influ-
ence for the entire $IVT$ variability. Main reasons for the low contribution of coherent patterns are the relatively low standard deviations compared to their mean (see $s_q$ and especially $s_v$). Even considerable correlation ($r_{corr} \geq 0.5$, Fig. 11) cannot generate relevant moisture transport contributions. Over the vertical extension, $cov_{norm}$ mostly remains below $5\,\%$. Even for low levels that contain most of the moisture transport ($\geq 700\,\mathrm{hPa}$), where considerable correlation of $q$ and $v$ is predominant, the cross-sections reveal a contribution of moisture transport variability to Eq. 8 in the range of 5 to $10\,\%$. Here, $q$ and $v$ prevail
with high values, but weak horizontal gradients (see Fig. 2). The strongest covariance ($> \pm 10\,\%$ for a few ARs) mainly occurs in higher levels above $500\,\mathrm{hPa}$, where moisture transport is weak (Fig. 9 and 11).

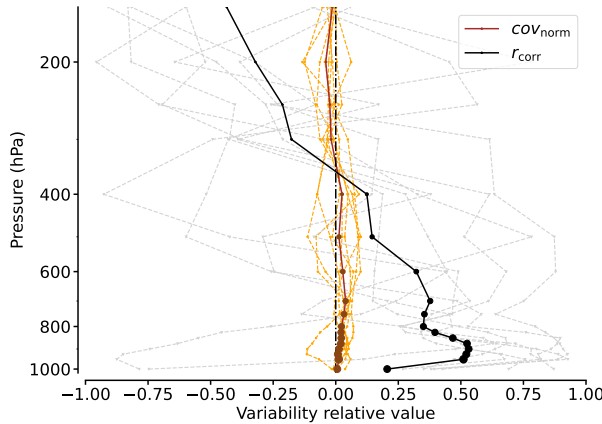

**Figure 11.** Vertical profile of AR moisture and wind normalised covariance ($cov_{\mathrm{norm}}$) and their correlation $r_{\mathrm{corr}}$ along the cross-sections. The bold lines indicate the mean value over all AR for each of the components. The sizes of the dots scales with the mean moisture transport value at this height normed by the maximum mean value of the entire profile.

In contrast to $cov_{\mathrm{norm}}$, the correlation between moisture and wind $r_{\mathrm{corr}}$ shows a large spread between the single ARs (grey lines in Fig. 11). Apart from AR5, other ARs exhibit less coherent patterns where wind and moisture do not necessarily correlate with each other (see also Fig. 2). Valid for most of the ARs, the correlation between moisture and wind peaks in the LLJ
height. The low-level negative correlation in Fig. 11 refers to AR9, indicating a clear horizontal displacement of the wind and moisture fields (Fig. 2). Here, subsiding dry air masses in the cold sector counteract the westward increase of wind speeds. Summarizing Fig. 10 and Fig. 11 above the marine boundary layer, the spatial distributions of moisture transport more resembles the moisture distribution than that of the winds, as there is more horizontal overlap between fields of moisture and moisture transport as against the wind fields (Fig. 2). Especially, AR1 and AR3 exhibit small horizontal variability in the wind
field, as winds are almost constant along the entire cross-section ($> 25\,\mathrm{m\,s^{-1}}$). ARs, being variable in moisture, consist of an elevated moist plume only residing in the AR core that is surrounded by dry air. The extension and intensity of the moisture plume in the mid-levels mainly controls the moisture transport variability (Fig. 10), while the location controls the case-to-case variability in the correlation of moisture and winds. The case-to-case variability in wind and moisture fields is also aligned with partly different regions of the respective ARs that we consider. While AR5 was mainly observed in the AR centre, other
ARs such as AR2-4 and AR7 are sampled in the exit region of the AR (see Fig.1 and Fig. 3). Terpstra et al. (2021) detected that moisture and wind patterns and their coherence in polar ARs strongly change along the AR direction. In their case study, the pronounced AR pattern, as depicted in Ralph et al. (2017), vanishes out more towards the Poles and the AR exit region. Moreover, Terpstra et al. (2021) identified decreasing coincidence between moisture and wind in the polar AR exit region. Their LLJ resides in rather dry regimes below the local moisture maximum. We detect this vertical separation of moisture and
wind patterns mainly for AR4, but less uplifting of maximum moisture. In turn, we emphasise the horizontal displacement of moisture and wind sectors in AR exit corridors, such as for AR9, that causes the decorrelated low-level profile in Fig. 11.

In conclusion, the mean moisture and wind account for 95 % of overall moisture transport in arctic ARs. Moisture and wind patterns exhibit little coherence, especially in arctic AR exit corridors, and show high case-to-case variability in their correlation. We find that strong ARs ($IVT \geq 400\,\mathrm{kg\,m^{-1}s^{-1}}$) tend to feature strong, but rather constant winds. Instead, narrow and high-reaching moisture plumes in the core control the AR moisture transport variability, and the intensity of dry subsidence on the western AR flank further modulates moisture transport variability. An improvement for observing the moisture transport variability should thus be built upon supplementary moisture measurements rather than those of winds.

## 4   Moisture transport divergence from sondes

The incoherent cross-section patterns of moisture and wind fields (Sect. 3) suggest lateral differences in the moisture transport divergence components (Eq. 5) and motivate investigating the divergence in separate sectors across the front embedded in the AR. Showing the limits of a $TIVT$-based divergence, we investigate whether high moisture advection occurs more frequently in strong moisture-dominated AR sectors and whether mass convergence dominates in windy AR sectors. By categorising our results based on the AR sectors (Sect. 2.5), we examine how the divergence of moisture transport is characterised along cross-sections of arctic ARs (Q3), and evaluate how the sondes reproduce the features of the continuous cross-section representation.

### 4.1   In- and outflow $IVT$

The $IVT$ cross-sections show maximum $IVT$ varying between 200–650 $\mathrm{kg\,m^{-1}s^{-1}}$ for all ARs, while the outflow $IVT$ generally has a similar intensity as in the inflow leg (Fig. 12). The strongest AR in terms of maximum $IVT$ has the highest total transport in both flight legs. Overall, $TIVT$ ranges from 100–300$\cdot 10^6\,\mathrm{kg\,s^{-1}}$. Recall that this is approximately one third to one half of the $TIVT$ magnitude found in mid-latitude ARs (Ralph et al., 2017). Although the AR cores are 200–300 km wide, they provide more than half of $TIVT$. This contribution of the AR core agrees with findings from Cobb et al. (2021a) in mid-latitude ARs. Except for AR2 and AR7, the weaker gradients of $IVT$ are generally in the cold sector and not in the warm sector. The steep post-frontal $IVT$ decline in AR2 and AR7 results from weak low-level winds west of the AR (Fig. 2).

By contrasting the in- and outflow cross-section legs (Fig. 12), it can be estimated whether convergence or divergence of moisture transport inside the AR flight corridor exists, under the idealisation that no lateral entrainment into the AR flight corridor occurs (Sect. 2.4). However, we emphasize that a $TIVT$-based interpretation of predominant moisture transport divergence underlies strong idealisation. It neither considers moisture flow being non-perpendicular to the flight and from the western and eastern boundaries, nor does it separate contributions of moisture advection and mass convergence.

Looking more closely into different aspects of $IVT$ and $TIVT$ values along the ARs, several effects become more apparent that may cancel each other out. In some cases, $IVT$ and $TIVT$ values across the AR decrease downstream, suggesting convergence. Yet, we likewise identify cases with weak downstream tendencies in total moisture transport or with slight increases, suggesting divergence (AR5 and AR6). Furthermore, the downstream difference of $TIVT$ is unevenly distributed along the cross-section $IVT$, and occurs mainly in the core of the AR (e.g. AR3, AR9), suggesting internal convergence. However, the

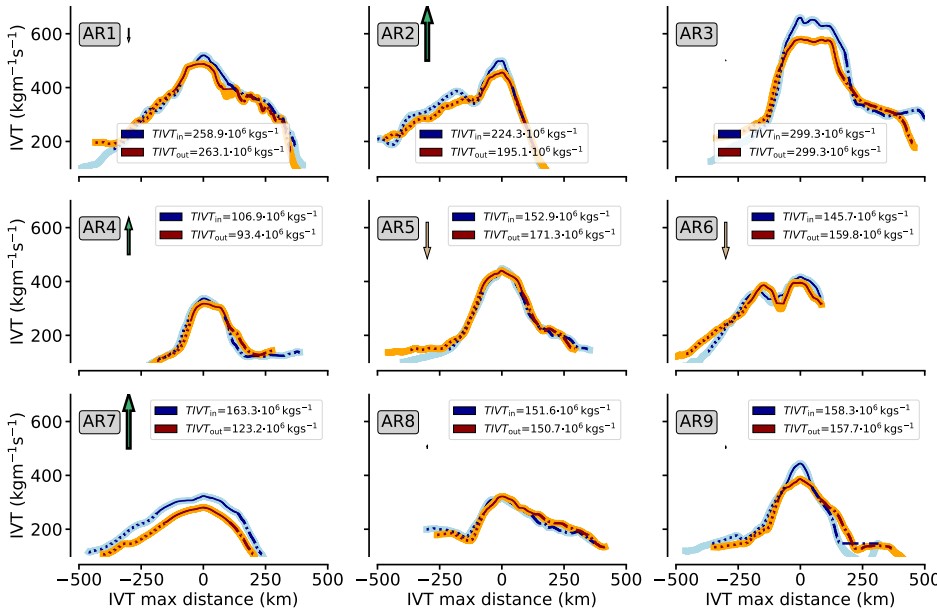

**Figure 12.** $IVT$ along inflow (outflow) legs in blue (orange) for all nine ARs (Fig. 1). Changes in line styles denote the AR sector classification (Sect. 2.5): Dotted lines represent cross-section parts attributed to pre-frontal (warm) sectors, while dashed lines refer to post-frontal (cold) sectors. Legend values depict $TIVT$ for the in- and outflow cross-section parts within the AR. They include $IVT$ purely internal of determined AR borders (Sect. 2.5). Arrows indicate the $TIVT$ difference between in- and outflow leg, scaled in length and width. The differences can be viewed as simple estimates of $IVT$ divergence in between both legs, according to Sect. 2.4. Upward (downward) arrow scales represent estimated convergence (divergence) magnitudes. Note that the along-flight x-axis orientation is from east (left) to west (right).

opposite behaviour in the frontal sectors partially compensates for the core and the downstream decrease of $IVT$ (e.g. sug-
505 gesting pre-frontal divergence in AR5 and AR6). This potential divergence is in contrast to the findings in Guan et al. (2020),
where the pre-frontal sector is denoted as a region of moisture transport convergence. Therefore, we choose the regression
approach from Sect. 2.4 to diagnose moisture transport divergence in each frontal sector of the arctic ARs, and to avoid strong
idealisation of the exclusively $TIVT$-based interpretation.

### 4.2 Sonde-based divergence and its representativeness

To derive the $IVT$ divergence ($\nabla IVT$), we thus use the regression-based approach (Sect. 2.4) for moisture advection $ADV$
and mass convergence $CONV$ (Eq. 5). The results from the continuous cross-sections are compared to results based on the
synthetic sondes that sample the cross-sections (as illustrated in Fig. 6).

In the continuous representation, $ADV$ and $CONV$ exhibit different vertical profiles throughout the cross-section AR
sectors. For the intense AR3, moisture transport divergence values in the sectors range from $-3 \cdot 10^{-4}$ to $+1 \cdot 10^{-4}\,\mathrm{g\,kg^{-1}s^{-1}}$
(Fig. 13). While moisture advection ($ADV$) does not rise above $\pm 1 \cdot 10^{-4}\,\mathrm{g\,kg^{-1}\,s^{-1}}$, mass divergence ($CONV$) decreases

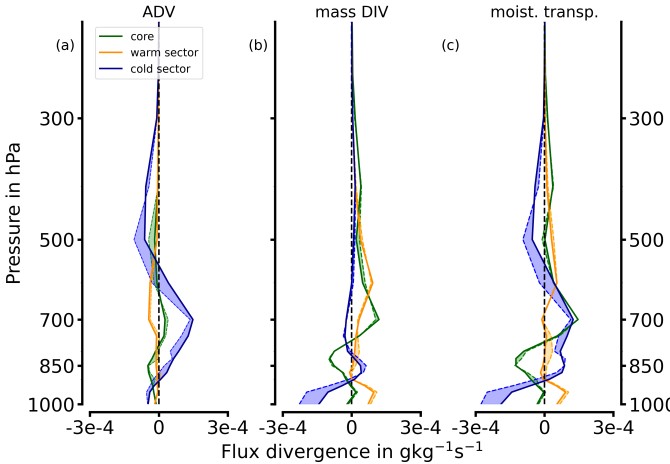

**Figure 13.** Vertical contributions from $ADV$ (a) and $CONV$ (b) to moisture transport divergence (c) for the frontal sectors in AR3. Bold lines represent the continuous AR representation, while dashed lines depict the sonde-based representation with the deviations as shadings.

below $-2 \cdot 10^{-4}\,\mathrm{g\,kg^{-1}s^{-1}}$. The moisture transport divergence is strongest in the post-frontal sector with both signs. In detail, substantial advection occurs in the post-frontal cold sector, whereas the warm pre-frontal sector and the core exhibit weak advection (Fig. 13 a). Similarly, the strongest mass convergence is found within the post-frontal sector (Fig. 13). Not only $ADV$ and $CONV$ act differently between the frontal sectors, they also dominate in different vertical levels. The maxima of $ADV$ and $CONV$ are not located at the heights where moisture and wind each dominate (Fig. 2). Instead, while advection is predominant at mid-levels above 800 hPa up to 500 hPa, the mass divergence primarily acts below this level (Fig. 13 a, b). In the vicinity of the LLJ in the AR core at around 850 hPa, predominant mass convergence (negative values) prevails, although the vertical column is slightly divergent in total. The advection of moisture in the prefrontal sector and core is too weak (Fig. 13 a) to compensate the more prominent mass divergence (Fig. 13 b). The vertically integrated moisture transport convergence (divergence) is highest in the cold post-frontal (warm pre-frontal) sector of AR3 (Fig. 13c), while the post-frontal sector shows the strongest variations with height. We attribute the drying in mid-levels of the cold sector in AR3 to the dry cold air masses overrunning the AR behind the cold front (as also visible in Fig. 2 c). The change in $IVT$ direction (not shown) behind the cold front accounts for the low-level mass convergence in the post-frontal sector.

The fact that the moisture transport divergence components differ across the frontal axis is in line with mid-latitude AR based statistics of Guan et al. (2020). In detail, the characteristics in AR3, described above, differ from the AR case observed by Norris et al. (2020). In their airborne study of a mid-latitude AR, they found moisture transport convergence to be strongest close to the AR core and found rather opposite signs for the pre- and post-frontal regions to us. It is worth mentioning the weakness of pre-frontal moisture advection in AR3, while advection is more enhanced in mid-latitude AR statistics (Guan et al., 2020). In contrast to both Norris et al. (2020) and Guan et al. (2020), we do not identify a dominance of dynamical convergence over advection. The magnitudes of moisture transport divergence in AR3 are lower. Nonetheless, we remind that

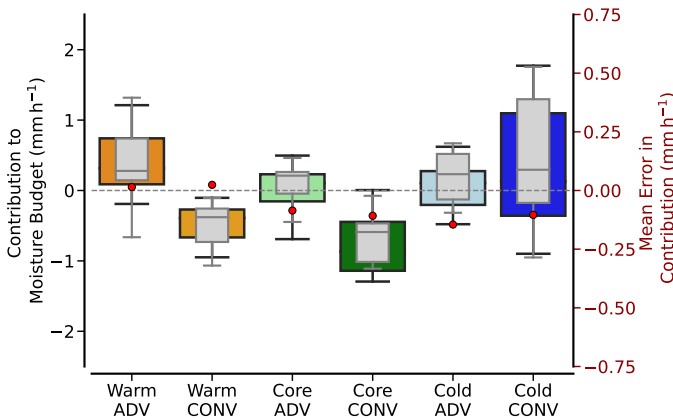

**Figure 14.** Box plot of moisture transport divergence contributions to daily moisture budget for all nine ARs. Values specify both components ($ADV$, $CONV$) for all frontal AR sectors (colour-coded). They compare the continuous AR representation (coloured box-whiskers) with the sonde-based values (grey). The boxes refer to the quartiles, and horizontal lines inside specify the respective mean.

Norris et al. (2020) and Guan et al. (2020) consider more intense mid-latitude ARs near its centre. While AR3 is exceptionally strong for arctic conditions (Fig. 3), it is rather moderate for mid-latitude scales (Ralph et al., 2019).

Three synthetic sondes per each frontal sector leg (located as for AR1 in Fig. 6) generally reproduce the divergence characteristics of the continuous reference for each frontal sector (Fig. 13). The highest deviations occur for the cold sector, arising
from mid-level moisture advection and low-level mass convergence overestimation. We note that slight cross-sectoral displacements of the sonde positions deviate the divergence characteristics, but they all maintain the principal vertical features for each component and sector that is shown in Fig. 13. Comparing our synthetic results with the mid-latitude based airborne study of Norris et al. (2020), which used real dropsondes, we see the strength of real sondes in their high vertical resolution. Real dropsondes provide much larger vertical variability than CARRA. Thus, the rather low divergence in Fig. 13 is probably not
only the result of truly lower divergence that prevails in arctic ARs compared to mid-latitude ARs. It may also result from the lower vertical variability and the coarser vertical resolution in CARRA, which leads to spatial aliasing in narrow mass convergent low levels.

To pinpoint the commonalities and differences of the moisture transport divergence characteristics compared to mid-latitude ARs, and to what extent the airborne sondes can reproduce them (Q3, Q4), we summarise all our arctic cases. We derive the
550 vertical integral $\nabla IVT$ (Eq. 5) and quantify the contribution of moisture transport divergence to the moisture budget (Eq. 2) in $mm\,h^{-1}$. According to Fig. 14, the warm pre-frontal sector overall supplies moisture via advection that overcompensates weak mass divergence. In contrast, the core and post-frontal cold sector advection have an inter-case variability in the advection of either dry or moist air masses. An overall large mass convergence in the post-frontal sector balances or even superimposes the advection. Surprisingly, the mass divergence in the core shows a robust negative contribution to the moisture budget (Fig. 14).
The advection profile of AR3 (Fig. 13) is consistent with the mean pre-frontal moisture advection in Fig. 14. For most cases, pre-frontal moisture advection primarily occurs in the mid-levels. The overall mass divergence in the core is surprising

as most arctic ARs contain LLJs (Fig. 2) which are associated with high mass convergence in mid-latitude AR. However, the low-level mass convergence below 800 hPa, found in many of our AR cases like AR3 (Fig. 13), is often superimposed by mid- and upper-level mass divergence above the LLJ (e.g. Fig. 13), where the AR widens causing directional divergence. The mass convergence in the post-frontal sector marks the highest inter-case variability. The high values of mass convergence, mostly originating from low-levels, as in Fig. 13, mainly arise from two cases (AR3, AR7). Here, we find changes in wind direction, as visible from the surface isobars in Fig. 1, inducing the confluence of moist air masses in the marine boundary layer. The sign of post-frontal advection is mainly determined by the intensity of dry air subsidence overrunning the western AR edge in mid-levels (see Fig. 2 and 13).

The range of budget contributions from -1.5 to +1.5 mm h$^{-1}$ of the arctic AR frontal sectors in Fig. 14 is smaller than mid-latitude AR magnitudes. Statistics of mid-latitude ARs in Guan et al. (2020) summarise budget contributions in the range -2 to +2 mm h$^{-1}$. Especially, the moistening in the pre-frontal sector due to advection is similarly identified for arctic (Fig. 14) and mid-latitude ARs (Guan et al., 2020). In turn, the fact that $CONV$ is found to be divergent in the pre-frontal sector and core in arctic ARs contradicts the findings of Guan et al. (2020), who emphasised a dominant convergence of mass in and ahead of the AR-embedded front for mid-latitude ARs. Unlike the mid-latitudes, the upper-level dominating mass divergence in the core of arctic ARs weakens the triggering of precipitation by convection. Instead, major precipitation fields are often shifted towards higher reaching convergence west of the $IVT$ maximum (not shown).

With six soundings per sector, the sondes reproduce the divergence characteristics (grey box-whiskers; Fig. 14), like the weak mass convergence that is ubiquitous for our arctic ARs. Overall, the sondes derive similar median values as the continuous AR representation and prove the fundamental observability of moisture transport divergence by discrete dropsondes. Six sondes per sector are basically capable to reproduce the general structure of moisture transport divergence across the AR and the vertically integrated contribution to the moisture budget. However, the percentiles between sondes and the continuous representation deviate. For individual events, sondes can misinterpret the magnitude of sector-specific divergence components considerably. Since this deviation is unbiased, though, the sonde mean errors remain below 0.1 mm h$^{-1}$ (Fig. 14).

Nonetheless, the precedent comparison of our sector-based values of moisture transport divergence in arctic ARs to those in Guan et al. (2020) has to consider additional aspects. First, our arctic corridors along the sea ice edge are primarily attributed to the AR exit region, as the centre is located more southwards (see Fig. 3). For this exit region, we expect stronger divergence than convergence when the outflows of the ARs spread out. Guan et al. (2020) refer to the conditions across the AR centres. Second, our frontal sectors are larger than those classified by single reanalysis pixel-based values used in Guan et al. (2020). Our sectors are, however, more comparable among the AR events as they are relative to AR strength and restrict to our defined boundaries of the AR. Note that the post-frontal sector in Guan et al. (2020) is also more distant and exclusive from the actual AR, where we would already consider dissimilar flow patterns, such as southerly flow backside of the cyclone (AR7; Fig. 1). Furthermore, Norris et al. (2020) highlight in their airborne case study that even higher values of moisture transport divergence occur at smaller scales than the reanalysis resolution used in Guan et al. (2020). Hence, we assume that the sector-based values of moisture transport divergence (Fig. 14) will also increase for smaller AR domains.

Another point not yet addressed is the fact that we mimic all observations of moisture transport divergence in terms of flight

duration. This is a major difference from Guan et al. (2020), who derived divergence components for individual reanalysis time steps. Our airborne continuous realisation (Sect. 2.3) is non-instantaneous. This realisation can cause considerable deterioration in our understanding of the AR flight corridors due to the temporal AR evolution meanwhile. This issue is also addressed in the

595 study of Norris et al. (2020), where they correct for the AR displacement over the flight duration by a time-to-space adjustment. However, they were not able to account for local temporal changes. Since our flight pattern cover larger AR flight corridors than in Norris et al. (2020), it is worth investigating to what extent the temporal AR evolution during flight may distort the airborne moisture transport divergence results.

## 5 Distortion by non-instantaneous soundings

This section examines the extent to which the temporal AR evolution during flight affects the sonde-based representation of $IVT$ divergence (Q4). During the up to 3 hours it takes to fly over AR flight corridors to observe the in- and outflow (Sect. 2.3), ARs evolve, resulting in $IVT$ divergence based on non-instantaneous observations. To quantify the deviations in $IVT$ divergence due to non-instantaneous observations, we contrast the spatially continuous representation, which is the spatially interpolated CARRA data at the aircraft location for each point, in two temporal perspectives. This is done by establishing an

instantaneous reference. The instantaneous reference is based on the spatially continuous airborne representation, but only for the CARRA output at the central hour of the flight, without interpolation in time. This can be thought of as a continuously sampling and infinitely fast aircraft, so we refer to it as the *optimum airborne representation*. We contrast the sector-based $\nabla IVT$ of this reference with the one of the non-instantaneous continuous airborne representation defined in Sect. 2.3 and analysed in the previous sections, which takes the flight time into account.

Summarising all arctic ARs, Fig. 15 demonstrates that the characteristics of $\nabla IVT$ in the different sectors are more or less reasonably reproduced by the non-instantaneous representation. However, we note that the mean deviations for the non-instantaneous divergence caused by the flight duration (red dots; Fig. 15) are much greater than those by discrete sounding we showed in Fig. 14). In detail, the evolution of the ARs changes the airborne estimates of prevailing moisture transport divergence by up to 25 % and even stronger in the post-frontal sector of the AR. In the post-frontal sector, the mean deviation

exceeds $0.5\,\mathrm{mm\,h^{-1}}$, whereas sonde undersampling (Fig. 14) only induces mean deviations $\leq 0.1\,\mathrm{mm\,h^{-1}}$. The temporal evolution of the AR throughout the flight can thus strongly change the divergence estimates for individual sectors. It is $ADV$ in the post-frontal sector where we find the highest deviations compared to the instantaneous snapshot (Fig. 15). The deviations show up in the mean, median and standard deviation. The optimum airborne representation, that is instantaneous and spatially continuous, not only shows more robust post-frontal dry advection than the non-instantaneous perspective, but there is also

much greater case-to-case variability in its magnitude (0 to -2 mm h$^{-1}$) than seen over the flight duration. Considering airborne (non-instantaneous) deviations for single ARs, we find cases with deviations in $ADV$ that exceed more than $2\,\mathrm{mm\,h^{-1}}$. In the prefrontal sector and core, the median of $CONV$ is barely affected. $CONV$ has higher mean deviations in relative terms, but these have less influence on absolute deviations of moisture transport divergence.

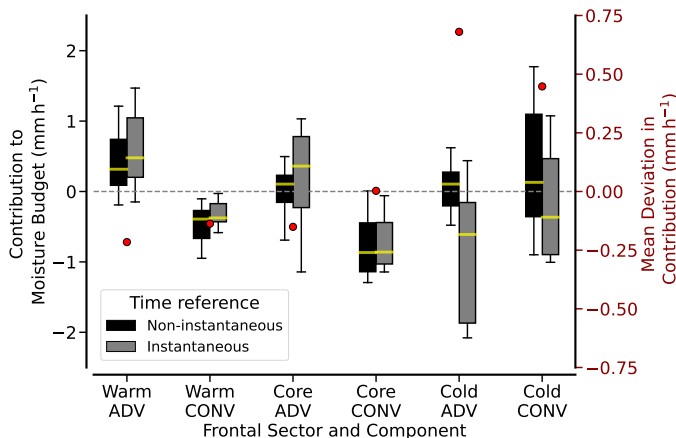

**Figure 15.** Comparison of divergence component contributions to daily moisture budget from spatially continuous AR representation, refer-ring to either evolving flight values (non-instantaneous) or to the values for the centred hour (instantaneous). Values are given for each frontal sector. Black error bars are identical to the coloured boxes in Fig. 14. Grey values represent centred hour-based values. Red dots quantify the mean divergence deviations between both time references.

For the optimum representation of $\nabla IVT$ in arctic AR cross-sections, the divergence characteristics in the AR sectors (Fig. 15) agree more with the mid-latitude statistics in Guan et al. (2020). In particular, we identify larger gradients in the divergence components across the front embedded in the AR, with overall moistening in the pre-frontal sector and drying in the post-frontal sector. In the arctic ARs, this is mainly driven by advection. Mass convergence being predominant in mid-latitude ARs is, however, missing or is at least superimposed by mid-level mass divergence.

Having confined ourselves to the spatially continuous representations of moisture transport divergence, we now contrast these with the sonde-based divergence resulting from the combination of non-instantaneous, and discrete spatial sampling (Sect. 4). To purely attribute the non-instantaneous effect on the divergence estimates at specific sonde locations, we hold the sonde positions fixed in both time perspectives. Thus, we do not relocate sondes once the sector-based $IVT$ thresholds are exceeded at different locations in the instantaneous representation. Using the root-mean-square error (RMSE), Fig. 16 compares the different spatial samplings (continuous and discrete) in both time perspectives, with the optimum airborne representation being our reference. According to Fig. 16, the deviations caused by discrete subsampling are minor compared to those induced by the temporal AR evolution and cannot compensate the latter, although they occasionally act in the opposite directions. The RMSEs for non-instantaneous sampling are higher than the ones only induced by discrete sampling. Figure 16 underlines that the largest deviations occur in the cold post-frontal sector. While the RMSE for the combination of non-instantaneous and discrete subsampling in the warm pre-frontal sector and core is around or slightly below $0.5\,\mathrm{mm\,h^{-1}}$, the RMSE reaches up to $1.5\,\mathrm{mm\,h^{-1}}$ in the post-frontal sector. Non-instantaneous discrete sampling by sondes misrepresents the divergence components by more than 50 % of the actual values (compare with Fig. 15). In turn, the RMSE resulting from the discrete subsampling only (green bars in Fig. 16) remains robustly below $1\,\mathrm{mm\,h^{-1}}$ throughout all frontal sectors.

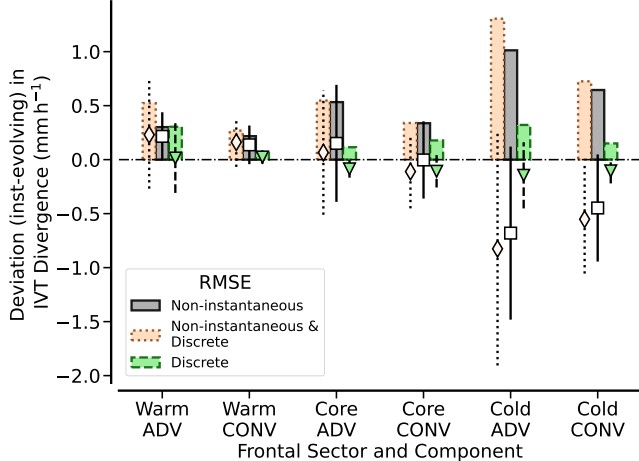

**Figure 16.** Total sonde deviation (orange) and individual deviation by only discrete sondes (green) and by non-instantaneous sampling (grey) for daily $IVT$ divergence in each frontal sector and divergence component (Eq. 5). For all ARs, positive bars indicate the root-mean-square error (RMSE), while error markers and lines depict mean deviations in combination with their standard deviations for all ARs.

The values of moisture advection are more sensitive to airborne sampling than mass divergence values. In contrast to discrete sampling, errors induced by instationarity to $ADV$ act in a more consistent direction (error bars in Fig. 16). In the cold sector,
AR instationarity mainly leads to an underestimation of dry advection. These tendencies result from the oblique movement of the ARs which are not necessarily aligned with the moisture transport direction. In fact, our ARs move more or less to the northeast, while our flight pattern aims for cross-sections being orthogonal to the transport and are more zonally orientated. For the flight centred hour (instantaneous reference), drier air masses are already more embedded in the post-frontal sector of the inflow cross-section leg, but less in the outflow leg. This changes the moisture gradient between both flight legs, increasing
the overall dry advection. The winds exhibit less horizontal variability (Sect. 3.4). Temporal displacement thus varies $CONV$ less than $ADV$, so that $CONV$ remain less sensitive to non-instantaneous sampling.

We deduce that our flight pattern is subject to the strongest sonde-based misrepresentation in $\nabla IVT$ from advection in the cold post-frontal sector. Flight planning should involve weather forecasts to adapt the flight legs for AR evolution. Forecasts can estimate the mean propagation speed and direction of the AR flight corridor centre so that in- and outflow legs can be shifted
adequately. Similar to the error magnitudes, the inter-case variability in the divergence misrepresentation is much higher for advection (Fig. 16). The large inter-AR variability in the results highlights that more ARs should be considered in order to generalise the errors to arctic ARs. This includes differentiating errors not only by the AR strength, but by their AR area type (entry, centre, exit). If flights are located in intense AR centres, we expect mass convergence to contribute more and errors due to sounding spacing to increase. The mass convergence may be more similar to the mid-latitude values in Guan et al. (2020).
However, AR centres are rarely encountered near the sea ice in spring (Fig. 3). The occurrence of AR exit corridors over arctic sea ice is significantly higher. This underpins the usefulness of our error estimates for research flights in arctic ARs.

## 6 Conclusions and perspectives

This study investigated the characteristics of the moisture transport divergence in arctic Atmospheric Rivers (ARs). We conducted our analysis from an airborne perspective to assess the dropsonde-based observability of moisture transport divergence of arctic ARs. We characterised the uncertainties in sonde-based estimates of the AR moisture transport divergence, focusing on two limitations: subsampling by a limited number of sondes and the instationarity of the AR over the flight duration. For this, we followed a synthetic approach using reanalysis data as virtual truth. CARRA reanalysis data were interpolated on synthetic flight patterns that consist of two cross-sections covering frontal sectors over the entire AR transect. Single vertical profiles emulate dropsondes. We considered nine arctic AR events over the Atlantic pathway to the Arctic Ocean in the vicinity of the sea-ice edge from last decade. The values of Integrated Water Vapour Transport ($IVT$) in the AR cores range from $300$–$600\,\mathrm{kg\,m^{-1}\,s^{-1}}$, although the ARs are primarily examined north of their centre. We thus classify these ARs as overall strong for arctic conditions. Still, the bell shape of $IVT$ across the AR varies strongly in between the AR cases. The cases cover a large variability and consist of various synoptic patterns (extended troughs, blocking situations, single cyclones) in which the ARs are embedded. This study delivers benchmarks of uncertainties in the airborne representation of sonde-based AR moisture transport divergence. We conclude the four pursued questions (Q1-Q4) as:

**What is the maximum distance between sondes to determine the total moisture transport in arctic AR cross-sections? (Q1)**

- For the sonde-based determination of Total Integrated Water Vapour Transport ($TIVT$) in arctic AR cross-sections, sonde spacing below $100\,\mathrm{km}$ robustly keeps $TIVT$ errors below $10\,\%$ (Fig. 8). In strong ARs with $IVT$ exceeding $500\,\mathrm{kg\,m^{-1}\,s^{-1}}$, too coarse $IVT$ representation at the AR core leads to $TIVT$ underestimation. Gaussian fits help to reproduce the cross-section $IVT$ shape, but are sensitive to how sondes estimate the maximum $IVT$ and its location. Thus, precedent flight planning should aim for a sonde release at the forecasted $IVT$ maximum and place additional sondes symmetrically around. For arctic AR widths of $400$–$800\,\mathrm{km}$, we suggest a minimum of seven soundings per cross-section (roughly $100\pm20\,\mathrm{km}$ spacing, similar to Ralph et al., 2017) to derive $TIVT$ in both cross-section legs. The maximum $IVT$ is more correlated to $IVT$ variability than the AR width is. The planning of sonde releases should thus rely on the forecasted gradients of $IVT$ along the cross-section. We highlight that the differences of $TIVT$ between the in- and outflow cross-sections are in a range of $2$–$15\,\%$ (Fig. 12). If we want to reliably estimate moisture transport divergence based on $TIVT$ from both cross-sections, the sonde-based uncertainty of $TIVT$ for a single flight leg must be considerably lower.

**How correlated are moisture and winds in arctic ARs and do coherent patterns contribute significantly to $IVT$? (Q2)**

- Moisture and wind in arctic ARs along the flight transects are only moderately correlated, with a maximum mean correlation coefficient of $0.5$ at about $850\,\mathrm{hPa}$ height, but much less at other heights. At the same time, the standard deviation of both quantities is smaller than its mean, respectively. Moderate correlation and limited variability result in a small contribution of coherent patterns to $IVT$, which is smaller than $10\,\%$ of the moisture transport by the mean quantities.

We draw the conclusion that collocated sampling of wind and moisture is not a priority. It is notable that the moisture variability dominates the wind variability over most of the profile between 850 and 500 hPa. Only close to the surface, the wind variability peaks at about 850 hPa and plays an essential role. When faced with a limited number of dropsondes, we prioritise supplementary airborne measurements of moisture to better represent arctic AR-$IVT$ variability.

**How can the divergence of moisture transport be characterised along cross-sections of arctic ARs? (Q3)**

– Contrasting the ingoing and outgoing $TIVT$ of the AR flight corridor using two cross-sections suggests an overall divergence in $IVT$. However, the ARs show different characteristics of $IVT$ divergence ($\nabla IVT$) in specific sectors across the AR-embedded cold front, especially when we decompose $\nabla IVT$ into moisture advection ($ADV$) and mass convergence ($CONV$). The advection term contributes most to the entire moisture transport divergence across the AR, especially in the pre- and post-frontal sectors (Fig. 15). The pre-frontal AR sector contributes to the moisture budget via
moisture advection, while the post-frontal sector generally shows dry advection. Across the front, the total contribution of $IVT$ divergence to the moisture budget is up to +1 mm h$^{-1}$ (pre-frontal moisture advection) to -2 mm h$^{-1}$ (post-frontal dry advection). This is slightly less than the magnitudes in mid-latitude ARs. However, in contrast to mid-latitude ARs, mass convergence is much less dominant in the arctic ARs apart from the post-frontal sector. Although the convergence of mass is dominant below 850 hPa, upper-level divergence often superimposes it. The advection term dominates at levels
higher than 850 hPa. For the post-frontal sector, this is mostly dry advection of cold air masses from west of the AR.

**To what extent can non-instantaneous sondes reproduce IVT divergence in the light of AR evolution during flight? (Q4)**

   – For reproducing $IVT$ divergence, the undersampling by a limited number of sondes matters. We recommend a sequence of at least seven sondes per AR cross-section. Given the widths of arctic ARs, this corresponds to a maximum sonde spacing of 100 km. Symmetrically placed around the maximum $IVT$, three sondes per AR sector leg are capable of
reproducing the sector characteristics of moisture transport divergence components with similar magnitudes. The mean absolute deviations to a continuous AR representation along the flight reach up to 0.1 mm h$^{-1}$ (Fig. 14).

   However, the deviations for moisture transport divergence by undersampling are minor compared to the deviations induced by the flight duration, that mostly range from 25–50 % of the actual divergence values (Fig. 16). The AR instationarity over flight time, including the displacement of the AR not necessarily along the moisture transport direction,
deteriorates the results more than undersampling and leads to an underestimation of the sector-based gradients in moisture transport divergence. In fact, the pre-frontal moisture advection and the post-frontal sector divergence (from $ADV$ and $CONV$) are stronger than assumed by sondes. Sonde-based values deviate the most in the post-frontal cold sector, where the AR has stronger gradients in moisture and winds than in the pre-frontal sector. The eastward displacement of the AR during flight changes the post-frontal wind and moisture conditions seen from the sondes. Over flight time,
the northern cross-section becomes drier due to the dry intrusions of air masses from west of the AR. The emerging meridional negative moisture gradient between both cross-sections, that is then seen by the sondes, suggests a meridional advection of moisture that partially compensates the actual zonal dry advection. This misrepresents sonde-based

$ADV$ and frequently causes deviations higher than 50 % of the actual values. Although post-frontal mass confluence is relevant, it is overestimated by sondes during flight.

The synthetic sondes confirm the observability of moisture transport divergence in arctic AR flight corridors by releasing drop-sondes in zigzag flight patterns in the future. Notwithstanding that we could release more sondes, it is the temporal evolution of the AR over the flight duration that leads to higher deviations in the divergence components rather than sonde undersampling. Therefore, the dedicated planning of such sonde-based purposes should not only include the sonde positioning, but also the minimisation of the flight duration. The placement of cross-section legs and their spacing should carefully consider the AR

displacement during flight. Shorter distances between the cross-sections not only reduce the flight duration, but also the area enclosed by sondes. Given the widths of the arctic ARs sectors, both cross-sections should be no more than 200 km apart. For several of our cases, the meridional separation is higher, and we have to expect considerable subgrid scale variability. The optimal and practically realisable strategy is to have collocated flights by two aircraft, with both cross-sections not far apart and sampled simultaneously. In addition to the sondes, complementary measurements of moisture should be prioritised due to

its higher variability, and its advection dominating $\nabla IVT$.

Additional limitations of our study need to be discussed. All our conclusions of the AR characteristics are drawn from simulations and should be verified with observations, as the extent to which the simulations reproduce the small-scale variability is uncertain. Furthermore, as our results are mainly based on corridors in the AR exit region, we strongly recommend extending our uncertainty assessment to other AR regions and expect the role of winds and mass convergence to increase in strong AR

centres. This becomes an even more important issue with respect to the tendency of arctic ARs to shift more northward and intensify under climate change (O'Brien et al., 2022). Furthermore, as we include a large variability of synoptic AR patterns but a small sample, we propose follow-up statistics with a larger number of AR events. The statistics can improve our understanding of the moisture transport divergence pattern in arctic ARs and attribute it to the dynamic and thermodynamic atmospheric conditions. For this purpose, CARRA represents a very suitable reanalysis framework. We encourage the use of the higher

vertical resolution of the model levels rather than our chosen pressure levels, although they were sufficient for initial estimates. For real sondes, we emphasise the added value of their high vertical resolution. Sondes provide more accurate information on the vertical composition of $ADV$ and $CONV$. The sonde-based approach is limited to regression-based divergence, where we only consider large areas and open meridional boundaries. Even with continuous lateral sampling, the meridional gradients are only coarsely sampled.

Therefore, a follow-up study should investigate how the arctic AR moisture transport divergence acts inside the flight corridor at grid-cell scales. This allows two additional topics to be addressed: First, the internal variability between both cross-sections can be derived more precisely to improve the flight patterns, second, the actual scales at which the moisture transport divergence varies significantly can be evaluated. This may increase the divergence magnitudes, similarly to Norris et al. (2020) who found larger values of the divergence components. They considered smaller AR flight corridors than the ERA-Interim pixels

referred to in Guan et al. (2020).

Despite the above limitations, the orders of magnitude for $IVT$ variability and divergence that we provide are representative for arctic ARs and quantify benchmarks for the sonde-based derivation. Emulated soundings assess possible airborne misrepre-

sentation in moisture transport divergence and will improve the interpretation of future real soundings aiming for the airborne closure of the moisture budget in ARs. The benchmarks are not only useful for improving flight strategies, but also indicate

deviations in corresponding model-observation comparisons. With the quantified uncertainties in the sonde-based AR representation of $IVT$ divergence, future airborne observations can better assist modellers in terms of the resolution and complexity required to represent predominant moisture transformation processes in arctic ARs.

*Code and data availability.*   The code created by HD analysing the downloaded reanalyses and creates the figures can be accessed via github and is made available under: https://github.com/hdorff94/Synthetic_Airborne_Arctic_ARs. The reanalysis data from CARRA (Schyberg

et al., 2021) and ERA5 (Hersbach et al., 2018) were accessed from the Copernicus Climate Change Service(C3S) Climate Data Store (CDS). The AR catalogue (Guan, 2022) used to pre-identify AR events of interest is provided by Bin Guan via https://ucla.box.com/ARcatalog.

*Author contributions.*   HD, FA and HK were main initiators for the work in the scope of this manuscript. FA, HK and VS helped to conceptualise the manuscript. HD conducted the analysis presented and drafted the manuscript under scientific supervision of FA, HK and VS. All authors contributed to revising the manuscript.

*Competing interests.*   The authors declare that they have no conflict of interest.

*Acknowledgements.*   This study was supported by the Deutsche Forschungsgemeinschaft (DFG; German Research Foundation) under the HALO SPP 1294. We thank the two anonymous reviewers, for providing us with helpful comments that improved the quality of the paper, and the associate editor Geraint Vaughan. We explicitly acknowledge the Copernicus Climate Change Service (C3S) Climate Data Store (CDS) for providing access to CARRA and ERA5 data. We thank Bin Guan for making the AR catalogue publically available via https:

//ucla.box.com/ARcatalog. Furthermore, we want to thank Melanie Lauer and Geet George for various helpful discussions. Henning Dorff is thankful to Jochem Marotzke and Dallas Murphy for providing fruitful comments on the writing style and structure of the manuscript. Thanks also go towards Norbert Noreiks for delivering sketches of the research aircraft.

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
