# Peer review of "Observability of Moisture Transport Divergence in Arctic Atmospheric Rivers by Dropsondes"

_EGUsphere, 2023_

## Author Comment (AC1)

**Response to the comments from Anonymous Referee 1 for the submitted ACP paper: ´Dorff, H. et al. 2023: Observability of Moisture Transport Divergence in Arctic Atmospheric Rivers by Dropsondes**

**Superior Erratum:**

With the aid of the reviewer's remarks concerning the frontal gradients in moisture transport divergence and the emerging revision of our manuscript, we identified erroneous results in our divergence calculation. In specific, we accidently did not calculate the wind divergence from using both u, v components but considered the absolute values of wind speed. By that our divergence results were direction-independent. In order to conduct a component-wise divergence calculation, we now had to rotate the u, v components in CARRA, as they are oriented along the local grid rotation and not the zonal/meridional direction. In doing so, the results in chapter 5 and chapter 6 (Figure 12-15) have changed moderately. In the response sections that refer to the respective manuscript sections, you will find the updated figures, alongside a specification of differences to the preprint results. In the remainder of this response, our updated results are already included when we present corresponding snippets of the updated manuscript paragraphs.

**Prefaces:**

We thank the ACP associating editor, Geraint Vaughan, as well as the Anonymous Referee #1, for this enlightening review. Please find our responses (in standard font) to the remarks from the Anonymous Referee #1 (in *italics*) below. We structured this response in such a way that comments on the most relevant text blocks being improved (e.g. introduction and discussion of results) are distinguishable from each other.

We reserve the right to apply minor changes to the here modified text snippets for the final revised manuscript in order to achieve even more concise phrasing and to guarantee grammatical correctness.

**Responses to Reviewer 1:**

*The article is a comprehensive piece of work and presents nice and illustrative figures. I think the presented approach is valid and the content is certainly worth for publication. However, in its current form, the study is not sufficiently motivated and the results are not properly discussed in view of related work. Hence, the novelty in terms of applied methods and results and the added knowledge about Arctic ARs and their observation strategy remains unclear. The presentation quality suffers from a confusing writing style. I recommend major revisions and encourage the authors to carefully rewrite their work to improve the readability of the manuscript.*

**Response:** First of all, we want to expressly thank you for the very detailed and well-specified feedback. We are certainly confident that the consideration of your remarks enables a significant improvement of the manuscript. The remarks help to transfer our scientific content and knowledge, that is considered as worth publishing, to the reader in a more logical and precise way. Accordingly, in our revision, we focus on improving the readability by more elaborated clarity and structure.

**Writing/Grammar**

*The grammar is a bit awkward and the article misses coherence and logical order within the paragraphs and sentences. The writing of this paper would benefit from a grammatical editing and language check.*

**Response:** We will invest more focus on coherence and the logical order for the paragraphs (individually and in its entirety). We will confirm additional cross-reading of the revised manuscript by either well-experienced or native English speakers.

**Terminologies**

- *the authors skim over many aspects simultaneously and it is up to the reader to guess potential relationships. Terms like "significant impacts", "pathway of ARs" (in a Lagrangian sense or AR displacement?), "moisture transformation processes", "moisture budget", "precipitation efficiency", "divergence of IVT", "IVT variability", "horizontal corridors", "dynamical and thermodynamical processes", "AR moisture budget components", "AR evolution" (many more examples in the other sections) are not defined or described, which makes it hard to understand the context of this work.*
- *Formulations like "are widely assessed over the mid-latitudes" (L33) or "manifold understanding" (L35) are without substance and should be avoided.*

**Response:**

- We agree that the introduction in particular was overloaded with terms defined/ described little or even not at all. You will find more details in our responses concerning the introduction (see below):

  *"significant impacts"* → The impact of ARs in the Arctic is now the key topic of the first paragraph and thus not unspecified.

  *"pathway of ARs"* → the ambiguous formulation has been modified to specify the long-distance displacement where air mass transformations occur.

  *"moisture transformation processes"* → here we decided to keep them more or less unspecified. Transformation processes, such as airmasses starting to precipitate, are described above (although indeed not linked to the term "transformation process"). Now we refer to according literature (You et al., 2022) and more clearly highlight the relevance of moisture transport to understand air mass transformation processes. We see a risk of distracting the reader more from our major research focus, the moisture transport and its observability (which the following paragraph is about) if we specify the air mass transformation much more at this early stage.

  *"moisture budget", "precipitation efficiency", "divergence of IVT"* → In order to not confuse the reader by too much details, we will erase the term "moisture budget" and focus on the fact that Seager and Henderson et al. (2013) finds the link between moisture transport divergence/convergence to local tendencies of moisture amount and how efficiently precipitation is induced. We are confident that this reformulation achieves more clarity. Nonetheless, in our opinion, defining each of the terms once again can be neglected in some cases. In particular, we consider the conceptional definition of divergence (convergence) to be known. A concise definition of the moisture budget follows in Section 2 using Eq. 1.

  *"IVT variability"*

  We agree that this wording was imprecise. We will specify that we mean the spatial variability and add respective findings from Guan et al (2015) describing how spatial IVT variability is composed in atmospheric rivers.

  *"horizontal corridors"*

  We rephrased this to "horizontally extended areas". Later, the term "AR corridor" will be defined using Figure 3.

  *"dynamical and thermodynamic processes",*

  We carefully checked the respective phrase, but argued that too much specification here might distract from the key message of this phrase and paragraph, namely the frontal gradients in moisture transport divergence.

  *"AR moisture budget components"*

we will reformulate the sentence without explicitly mentioning the moisture budget components and solely refer to IVT divergence. By that, the term "AR moisture budget component" is not needed before it is introduced in Sec 2.3.

*"AR evolution"*
We will specify that we mean the evolution in a temporal sense, meaning the life cycle of an AR and the AR displacement, that cause Eulerian differences over time.
We are confident that stronger interconnection between the above-mentioned terms delivers a substantial improvement for the clarity and logical order of the terms. This is explained in more detail in the answers to the introduction.

- We will delete any formulations like "widely assessed" (L33) or "manifold understanding" (L35) that impair the argumentation in the introduction.

**Introduction:**

*The motivation […] remains unclear. I understand that a limited number of dropsondes might affect IVT estimates and […] that Arctic ARs may be not well characterized, but is that all? The authors remain very vague about related work. Although references are given, only rarely a relevant result is described. […] this is needed to understand the motivation of the study.*

**Response:** We suppose that this results from an interplay between insufficient clear structuring in specifying the motivation and identifying the research gaps that emerge from so far broadly studied airborne observations of ARs (at least in the mid-latitudes). When referring to literature, we mostly miss to pinpoint clear findings relevant for our motivation.
Therefore, we carefully rewrite the introduction to clearly identify research gaps and include the following modifications, addressing several reviewer remarks:

- We rephrase the first sentence of the manuscript that actually intended to highlight the presence of ARs. Following Referee 2, a *"clear and concise description (definition, shape, evolution, region of occurrence) "*of ARs is now added.
- The purpose of the first paragraph which is about the importance of the presence of ARs in the Arctic, is described in more detail.
- We conduct a more thorough literature review regarding arctic ARs. We will describe findings from the literature precisely, in particular, in the first paragraph that focusses on the importance of ARs in the Arctic on short-term scale, but also at many other places throughout the manuscript.
- The research focus and overarching motivation of our feasibility study are introduced earlier (4th paragraph). This facilitates the understanding of our conceptual perspective (assess the observability of AR to improve flight campaigns) in the following.
- Built on this, our four research questions will not anymore be introduced at the end of the introduction. They are motivated with a respective paragraph one after another. Each paragraph faces a relevant problem/requirement of airborne moisture transport sounding and refers to knowledge gained from ARs in mid-latitudes. By this, we aim to more thoroughly cover the questions arising for the reviewer:
  *Why should ARs and their observation strategy be different in the Arctic? What is known about Arctic ARs in general and dropsonde-based IVT estimates?* Current studies so far mainly focused on IVT variability within mid-latitude AR centers. For the Arctic, where we do expect less IVT, the sea-ice region is more affected by the outflow region of long-elongated ARs residing over the North Atlantic than by the center of an AR overpassing this region. One may argue, that if IVT and its variability are supposed to be weaker in the Arctic, then the mid-latitude AR based requirements will, or even more so, hold for arctic ARs. However, the deployment of dropsondes is cost-intensive and should be always optimized in a cost-loss ratio. This should be quantified in such a way: *How few dropsondes are sufficient to characterize IVT in Arctic ARs and what is their uncertainty?*
  Regarding the frontal-specific characteristics of divergence in arctic AR, Guan et al. (2020) and Norris et al. (2020) determined AR frontal differences of moisture transport

divergence for mid-latitude ARs. However, we do not know how the divergence/convergence of moisture transport takes place in arctic AR regions. Guan et al. (2023) found that the Arctic is more affected by mature ARs that commonly start to dissipate and by the outflow regions of ARs. Correspondingly, such facts are elaborated more thoroughly in the introduction to better motivate Q3.

*"How has the problem been addressed (methods)?".* We understand this interest. We give more relevant information from the literature (e.g in how the sensitivity to sonde spacing has been assessed already). In our opinion, too many details on specific methods to derive IVT divergence should be postponed for the sake of readability. When picturing the remainder of the manuscript (last paragraph of the introduction), we try to more explicitly mention that an entire section introduces how divergence can be calculated from airborne sonde profiles and how, in detail, it is done in this study.

- *How and why were the particular nine cases selected (unclear: L68 "predefined in ERA5", L119ff "picking (…) from catalogue") and why was only the Atlantic region considered? Please explain the purpose of placing the legs at the sea ice edge (L105ff is unclear). Why is only spring considered?* The selection is related to the AR impact on the sea-ice melting when sea-ice reached its maximum extent in spring (paragraph 1). We specify the Arctic ocean as our region of interest due to the fact that the North Atlantic represents one of the most prominent pathways for ARs (Guan et al., 2023) into the Arctic as the moisture transport is undisturbed by any orographic barriers.
We revisit the explanation of the case selection are revisited in the respective section introducing the AR cases (Sec 2.2). We there add more details as given above. We remove the distracting and unprecise information about ERA5.

- ***Observation strategy:*** *How was the simulated observation strategy defined? How do aircraft limitations (flight duration, number of dropsondes) affect the strategy?*
**Response:** In the introduction, we now highlight that long-range research aircrafts are in any case needed for a strategy to derive IVT divergence in ARs. The different characteristics across the embedded AR front require a large area of interest and the flight duration consists of a couple of hours, in contrast to single circles (being performed to derive divergence in trade-wind regions, Bony and Stevens, 2019). We add more information of the width of ARs responsible for the final flight duration and the importance of two cross-section legs. In turn, such detailed specifications will be given in Section 2.3 that deals entirely with the reason for our envisioned flight strategy.

**Specific remarks in the introduction:**

Unusual language:
*L31:*
**Response:** as mentioned above we rephrased the sentence: "Seager and Henderson (2013) point to the divergence (convergence) of IVT as the link of the temporal evolution of moisture amount to its efficiency to induce precipitation."

*L51 (deteriorate … representation?)*:
**Response:** We rephrased the sentence to clarify that too few sondes affect the airborne representation of IVT negatively: "A limited number of dropsondes may deteriorate the airborne representation of AR moisture transport variability if the sounding frequency is too low to reproduce the spatial variability of IVT."

*L57 (monitors transport (…) seen from research aircraft?):*
**Response:** As a consequence of our restructuring we deleted this sentence.

*In the introduction the reader is distracted by details about CARRA regional reanalyses. I suggest adding more details about CARRA, but in the methods section.*
**Response:** We delete the corresponding sentence and extended the specifications of CARRA in the respective data section

*Q2 addresses correlations of wind and moisture, which has not been motivated by the introduction. What is (un)known? For understanding Q3, the relevance of IVT divergence needs to be explained more carefully.*

**Response:** The restructuring of the introduction considers these remarks in the paragraphs. Q2 now has an individual motivation like all other research questions. We motivate more thoroughly the correlations of wind and moisture and their potential interest for the measurement strategies and also for different spatial patterns along the AR cross-section. This is done after highlighting the sensitivity to sounding frequency.

For Q3, we describe the relevance of IVT divergence more thoroughly with respect to steering the local amount of moisture or to precipitation triggering.

**Section 2 and 3 (structure of the article):**

*The structure of the method sections is confusing and I suggest that sections 2 and 3 are merged. Section 2.1 (description of dropsonde data that is actually not used) can be deleted. Sections 2.2-2.4 can be summarized in a data and methods section. The TIVT definition (now in Sec. 4) should be moved to the Sec. 2-3.*

**Response:** We agree that a merging of both sections improves the structure of the manuscript. Furthermore, we deleted Section 2.1 which is indeed redundant. Hence, we come up with a modified structure of Section 2, which now consists of:

**2 Airborne derivation of moisture transport divergence in arctic ARs**
2.1 Reanalysis framework
2.2 Selection of Atmospheric River cases
2.3 Flight pattern and emulated observations
2.3.1 Zig-zag flight tracks observing AR corridors
2.3.2 Synthetic dropsondes
2.4 Sonde-based divergence derivation
2.5 Decomposition in AR frontal sectors

Since this section becomes rather long, we will provide some guidance for the reader at the beginning of Section 2, before then coming to the subsections.

In the following, the responses relevant to the modified Section 2 are specified. All section numbers refer to the modified outline. From Section 2.3 on, this means that the first number of each original major section decreases by one.

**Section 2.2**

*L105ff is unclear. How and why were the particular nine cases selected (unclear: L68 "predefined in ERA5 […] and why was only the Atlantic region considered? Please explain the purpose of placing the legs at the sea ice edge (L105ff is unclear). Why is only spring considered?*

**Response:** We will carefully rewrite the first paragraphs of the original Sect 2.2. We will move the definition of IVT to Sect. 2 as it is a basic concept to define ARs. Before, it distracted in Sect. 2.2 when giving details about our individual AR selection criteria. To better explain our selection, we orientate to your question and recapture information given in the introduction as: "The transformation of arctic airmasses moving over changing surface types (open ocean and sea-ice) along large-scale meridional circulations is part of current research and investigated by research aircraft over the Arctic ocean (Wendisch et al., 2021). In this context, our study selects ARs causing air masses to overshoot the sea ice edge in the Arctic ocean. The principle identification of relevant arctic AR events is based on the IVT-based AR detection catalogue by Guan et al. (2022). Among these ARs, we focus on spring season, when maximum sea-ice

extent in the Arctic ocean starts to break-up and reacts very prone to the intrusion of warm and moist air (Mattingly et al., 2018). We remain to conditions and AR events only from last decade, as the arctic climate has been changed rapidly and intensively over the last decades (Wendisch et al., 2023). Our selection constrains on ARs, whose lateral width is purely situated over open-ocean or sea-ice. This ensures that we do not encounter additional land effects on IVT (e.g. orographic-induced convergence) which are out of the scope of this study. Moreover, airborne observations and sonde releases over land are more complex to be conducted. Given these criteria, our study selected ARs from nine spring days between 2011 and 2020."

*Can you explain the relation of ARs and warm air intrusions (L116)?*
**Response:** Arctic events designated as warm or moist air intrusions can often be classified as atmospheric river, as the intrusions are in conjunction with strong transport of moist airmasses. Due to their subpolar origin, the airmasses are preferably warm compared to predominant arctic conditions. Accordingly, we will add an overarching description in the manuscript.

*Regarding CARRA, the authors should "[…] clarify the extent to which km-scale variability of moisture transport can be assessed. The grid spacing (how determined?) and effective resolution of such gridded data are certainly different."*
**Response:** The documentation of CARRA specifies an equidistant 2.5 km grid spacing over the entire model domain. Indeed, the grid spacing differs to the effective resolution of moisture transport. At least, Koltzow et al. (2022) illustrates the significant improvement of the decorrelation length for surface-near wind speed compared to ERA5. According to observations, the correlation decreases rapidly below 0.6 for distances longer than 50 km, roughly the ERA5 resolution in the Arctic. Nonetheless, we are aware that surface near wind is much more affected than upper levels, especially also when over complex terrain. In the manuscript, we will hence add: "Koltzow et al. (2022) verified the improved representation of CARRA in arctic surface-near meteorological conditions by decorrelation lengths of wind speed approaching observations more than ERA5."
It is obvious that the resolution of CARRA is certainly different to the effective resolution. If we consider the results from Skamarowk et al. (2014, https://doi.org/10.1175/JAS-D-14-0114.1) designating an effective model resolution of approximately six times the grid spacing, we can still assume that moisture transport can be resolved in the order of several 20 kilometers in CARRA. Using ERA5, we would thus remain in the range of ~100 kilometers and would be in the order of magnitude of envisioned sonde resolution, and cannot make robust statements.
Nevertheless, further investigation of the added value of CARRA in representing ARs is definitely very interesting. For our study, however, we see a risk to overload its content and hope that our study instead motivates further research of ARs using the novel reanalysis CARRA. We come back to this in the conclusions.

*Why do you use pressure level data only and how might the rather low number of vertical levels influence the results (L374)? What is the separation of the levels in the lower troposphere?*
The advantage and our reason for the usage the pressure-level CARRA data results from the consistency of pressure-levels allowing an easy calculation of the vertically integrated moisture transport (IVT) following Eq. 1. Moreover, when deriving the moisture transport divergence, the values at unique pressure values do not require any further interpolation. The separation of the levels can be depicted from the dots in Fig7-9.
In an exemplary case (not shown), we have used the model-level data and basically find higher variability in the vertical profile for wind and moisture, but the effect on IVT and IVT divergence is minor, and in particular does not change our overall results significantly (not shown). Still, we recommend using the higher resolution model data in the conclusion for follow-up studies.

L152f (why are radiometer/radar relevant?)
**Response:** We will rephrase the sentence to:
"Our 1 Hz representation of the aircraft location is in line with the operational resolution of common airborne remote-sensing products (e.g. Mech et al., 2014; Konow et al., 2019) that can complement dropsonde-based moisture data."

*Arctic ARs: I recommend adding a more detailed discussion about the determined characteristics of Arctic ARs (e.g., L315ff, results of Fig.1 and 10). This should involve a discussion of the communalities and differences of the presented nine cases. The large case to case variability should be better discussed.*

**Response:** We will extend the description of our selected AR events. For that, we restructure the second paragraph of the modified Section 2.2, that introduces our AR cases and split it into two subparagraphs. First, we emphasize the inter-case variability with respect to communalities/differences in the synoptic situation. Second, we describe the actual AR pathways seen for our selection (aligned to the preprint version), as follows: "Low-pressure systems forcing large-scale meridional transport represent a common synoptic composition where ARs can evolve on the eastern cyclone flank and reach into the Arctic (Papritz et al., 2020). Similarly, blocking situations can favor meridional circulation. For our nine ARs (Fig. 1), we confirm a large case-to-case variability regarding the synoptic situation. While some ARs (AR2, AR3, AR4, AR9) have evolved along the eastern flank of large-scale troughs over Greenland, AR5 and AR6 are more steered by blocking high pressure over the Barent Sea. AR1 and AR7 are, in turn, reinforced by a mesoscale cyclone situated over the Fram Strait and reach very close into the cyclone center.

The synoptic compositions cause the ARs in Fig. 1 to extend over the North Atlantic and Arctic Ocean; the typical arctic moisture transport pathways (Papritz et al., 2021). Some ARs exhibit straight meridional moisture transport north of Iceland and approach or exceed Svalbard (AR1, AR2, AR3). AR4 and AR7 show more elongated filaments along the Norwegian coast but still reach far north. We consider eight independent AR events wherein AR5 is also considered for the consecutive day (AR6). At this stage, the centre of AR6 reaches close to the North Pole. AR8 originates from Siberia that represents another significant roadway for arctic moisture intrusions causing ARs (Komatsu et al., 2018). The last events in 2020 (AR8, AR9) are accompanied by a warm air intrusion period observed by the Multidisciplinary drifting Observatory for the Study of Arctic Climate (MOSAiC) expedition (Shupe et al., 2022), studied in (Kirbus et al., 2023)."

*Unclear statement", L119ff "picking (…) from catalogue")*

**Response:** we will modify the beginning of the paragraph as follows:

"A caveat of our selection is that a number of nine AR cases is rather small to make general statements about IVT variability in arctic atmospheric rivers. Therefore, we place our cases in the context of the climatology of arctic ARs in spring. Using the entirety of spring ARs along the Atlantic pathway from the catalogue of Guan (2022), […]"

Furthermore, we put our following statements in a more logical order to clarify that our AR sample is representative for the rather strong AR cases.

**2.3 Flight pattern and emulated observations**

*Removal of initial Section 2.1:*

**Response:** Due to the removal, slightly more description of flight performance and dropsonde characteristics to be emulated are given in Section 2.3. In the following our remarks regarding the flight strategy will be responded.

*I did not get how the flight tracks were defined. Isn't the zig-zag pattern only the consequence of sufficiently long cross-frontal legs at two latitudes that are required to capture the lateral heterogeneity and to be able to derive divergence?*

**Response:** For the divergence purposes, the cross-frontal legs are of relevance and actually sufficient. We discussed to keep our term "zig-zag pattern" due to the fact that a single aircraft has to perform an internal flight leg in order to connect both cross-sections (a relevant time constraint also for our analysis). Nonetheless, we agree to put more emphasis on the cross-section legs themselves when introducing our flight pattern. Accordingly, we will reformulate: "Instead, the high lateral variability in AR transport characteristics requires long flight legs across the AR front to better capture divergence heterogeneity. Such cross-sections can be

connected via an internal flight leg in a zig-zag flight pattern (Fig. 3). The zig-zag pattern thus observes AR corridors, along its transport direction. The boundary cross-section legs perpendicular to the major flow quantify the corridor in- and outflow, i.e. in- and outgoing IVT over the entire lateral AR extension and enable simplified divergence calculations." Information about diagonal legs and the moisture budget closure will still be given, but as a site note.

*Are all terminologies for the flight pattern (AR corridor, boundaries, boxes, sectors etc.) needed or would it be enough to describe two cross-sections at separate latitudes that are then classified in sectors? What defined the latitudinal spacing?*
**Response:** We will reduce the terminologies accordingly and speak of cross-section legs rather than "zig-zag" whenever sufficient. The latitudinal spacing was adapted in a way that no landmasses reduce the cross-section length in the outgoing cross-section and that the northern leg is at least 100 km away from the ingoing cross-section.

*It is sometimes confusing what data is used. I actually thought that the flight duration was not considered for the "continuous" (L394, 426). "Continuous" was also used earlier (see e.g., Fig. 14 caption), however, I think it referred to the high-resolution cross section profiles. I suggest a clear structure and description.*
**Response:** We apologize for imprecise terms as they are very essential for the comprehension of our work. Therefore, in our restructuring of Section 2 and 3, we will add the definition of the "continuous representation" at the end of Sect. 2.3.1 to make clear that this represents the "ideal" sampling of moisture transport from the moving aircraft.

*L233ff should be moved to the method section.'*
**Response:** We will move the definition of the TIVT to the Section 2.4 dealing with the sonde-based divergence derivation and are convinced that now this section is more compact.

*The advantages and limitations of the applied methods should be considered in view of other approaches.*
**Response:** We will extend our description of the applied methods and contrast more the advantages and limitations of our cross-section pattern for divergence calculations. One obvious limitation are the open boundaries that the cross-sections leave. The major advantage we see in the ability of the cross-sections to derive the divergence in different sectors across the AR embedded front more or less simultaneously. Similar as in Norris et al. 2020, that investigated the airborne divergence pattern and subdivide the examined AR corridors, the sensitivity to different spatial scales can be assessed.

*Unclear L182:*
**Response:** We assume that the connection to the precedent sentence was unclear, as well as the vague statement of "two impacts". We will rephrase the sentence to:
"The convergence/divergence of moisture transport thus affects the moisture transformation via two composites that we can attribute when splitting $\nabla IVT$ as follows:"

*I do not understand the sector classification: Please specify the "requirements" in Cobb et al. (L198ff). In L194 the prefrontal, core and postfrontal are differentiated. Then you come up with a threshold definition for the AR edges. How does this all fit together and how are the sectors defined? Please move relevant information about Arctic ARs to the introduction.*
**Response:** We will rephrase the description of our sector classification and explain the requirements of Cobb et al. in more detail, especially how we adapt those requirements to arctic conditions we found in Fig. 2.
With the term "AR edges", we mean the outer boundaries of the frontal sectors. At some lateral distance, the moisture transport (IVT) becomes too weak to be considered as atmospheric river. The requirements will be described as follows:
"Therefore, we conduct a similar sector-based decomposition of IVT divergence for our arctic AR events in CARRA. As in Guan et al. (2020) and Cobb et al. (2021a), our decomposition relies on the IVT characteristics along the cross-section (as depicted for an exemplary crosssection in Fig. 4). The central AR core represents the region of strongest IVT (> 80% of maximum IVT). East of the core we situate the pre-frontal sector and west the post-frontal sector. Yet, their outer edges are less trivial as ARs basically have open outer boundaries. To account for case-specific relative values, we assign frontal edges where IVT ≤ 0.33 $IVT_{max}$. As a secondary threshold, we declare a moisture transport with IVT ≤ 100 kg m$^{-1}$s$^{-1}$ as too weak to be assigned as AR-IVT. Both form the outer edges of the AR where the pre- and post-frontal sectors end (Fig. 4). Note that the latter threshold to define the AR edges follows the approach of Cobb et al. (2021a). However, we lower their mid-latitude based IVT threshold from 250 to 100 kg m$^{-1}$s$^{-1}$. By this, we refer to common polar moisture transport magnitudes that exceed the 95th percentile of climatology and are declared as ARs in the detection of Guan and Waliser (2015). Otherwise, as statistics in Fig. 2 indicate, we would either exclude most ARs north of 70∘N, or would shrink the AR cross-section that strong that most transport is ignored."
To facilitate the connection of our terminologies, we will provide a Figure (listed as Fig. 4) illustrating the IVT-based frontal sector classification along AR cross-sections. Afterwards, we display how the sondes are located correspondingly in both cross-sections (then Fig. 5).

[Figure]

*Figure 4 (in manuscript): Frontal sector decomposition for an exemplary AR IVT cross-section using the criteria described in Sect. 2.5. The colored shadings and text boxes indicate each frontal sector. The grey shading on the left represents moisture transport (i.e. IVT) that is not considered as AR because it is too weak.*

*Comment: In the following the Figure labels still rely on the original numbering.*

**Section 2 specific comments:**

*L202: I cannot see the three dropsondes that calculate IVT.*
**Response:** We will specify our misleading explanation: […], six synthetic sondes (three from the in- and outflow leg each) calculate the IVT divergence for each frontal sector respectively.

*L204 (putative? inconsistency?)*
**Response:** we delete both words as they do not provide any added values.

**Section 3: Moisture transport in Arctic AR cross-sections from soundings**

**General remarks:**

*The paper lacks a thorough discussion of the results, either within the result section or in a separated section at the end. [..] a few references within the result section, however, not detailed enough (see above) so that the added value of the paper becomes clear.*
**Response:** We agree that the discussion of results is worth improving. We decided to manifold and strengthen the discussions in the respective result sections rather than merging them in a

separated "discussion" section. We restructure concerning paragraphs in order to still unravel the discussions of results more stringently. We strengthen the interpretation of our results in a more connected comparison to findings from literature (mostly based on mid-latitude ARs).

In the following, you will find specific responses for the relevant sections, whereby many reviewer remarks are applicable for several paragraphs throughout our results sections.

*Add more references to figure panels within the text whenever appropriate.*
**Response:** Yes, this improves readability. We will add them especially in our result sections.

**Sect. 3.1: Shape of IVT across arctic ARs**

*I recommend adding more detailed discussion about determined characteristics of Arctic ARs.*
**Response:** We will take up this point for Sec 3.1, as the IVT shape of ARs in the Arctic is here first presented in more detail, and we find that comparisons are helpful here to categorize our cases. Accordingly, we added some more discussions about the IVT strength for our AR cases with respect to mid-latitude cases (Cobb et al, 2021) and arctic cases studies (Viceto et al, 2022). This is will be done as:

"Summarizing all cross-sections of our ARs from Sect. 2.2, most arctic AR cross-sections show this typical bell-shaped IVT curve over widths of roughly 400 -800 km and exhibit pronounced IVT maxima in the core of 300-600 kg m$^{-1}$ s$^{-1}$ (not shown). Only for the weak AR8, this structure is less pronounced. We find that our arctic AR are not substantially narrower than the AR widths of global climatology (Guan et al., 2015) or observed mid-latitudes events (Cobb et al., 2021). The flight planning should thus consider cross-section lengths around 500-1000 km similar to the mid-latitudes, but not only restrict to regions with IVT>250 kg m$^{-1}$ s$^{-1}$, that is broadly used threshold for mid-latitude ARs (e.g. in Ralph et al., 2019). The maximum IVT for the arctic events, is roughly half as high as the majority of mid-latitude AR from airborne studies in Cobb et al., 2021. Moreover, the IVT magnitudes strongly differ between our cases and synoptic conditions. The strongest ARs with maximum IVT (IVT$_{max}$) exceeding 500 kg m$^{-1}$ s$^{-1}$ are found for intense Greenland troughs, while weaker ARs along the Siberian pathway (see Fig. 1). Compared to other arctic cases, e.g. Viceto et al. (2022), we include stronger ARs."

**Additional Response:** We will specify the comparison between ERA5 and CARRA:
"Viceto et al. (2022) documented the improved representation of arctic AR characteristics in ERA5 against coarser reanalysis data. In our comparison of CARRA and ERA5, the location and horizontal pattern of the ARs agree quite well (not shown). For all cross-sections, we ascertain plausible IVT values from CARRA with respect to ERA5. In particular, we highlight that maximum (mean) values of IVT per cross-section increase by roughly 9 % (8 %) from ERA5 to CARRA on average. CARRA further increases the IVT variability by roughly 11 %. We attribute this to horizontal resolution being higher than in ERA5."

*Should there be a strategy to place one dropsonde at a simulated maximum IVT (L223)?*
**Response:** The restructured discussion of results will follow this suggestion as:
"The Gaussian fit to reproduce the IVT shape (Fig.4) is very sensitive to the actual positions of dropsondes. While the centered sonde is positioned close to IVT$_{max}$, a slight shift of this sounding, which easily occurs in real observations, can quickly lead to an underestimation of the moisture transport in the AR core. Flight planning should thus imply a sonde release in the vicinity of predicted IVT$_{max}$ and place additional sondes symmetrically around the core. While sonde positions in Fig. 6 are suitable to represent the cross-section IVT, other AR evince more complexity in being accurately represented by this number of soundings. We need further inspections on how sounding intervals deteriorate the AR moisture transport observability.

**Sec. 3.2 Sonde-based total cross-section moisture transport**

*I suggest adding a recommendation for the spatial separation (L252, L425) instead of a number per flight which depends on the flight performance. Figure 6: Change "seconds" to "minutes".*

**Response:** Indeed, a recommendation for the spatial separation is more universal with respect to flight performance. We will change the descriptions in this way and we also changed the axis of Fig. 6 to distances (km). Yet, since measurement operators frequently rely on specified time intervals when performing sonde releases manually, we add a light secondary axis referring to the spacing time. It is true that the duration depends on the flight performance, however, the values are valid for a common groundspeed at cruising level above 10 km.

*Fig. 6: The median lines for the grey boxplots are hard to see. I guess that these distributions are calculated from the boot-strapping method (add information to caption). How many cross-sections? Please add what percentiles the box and whiskers represent.*

**Response:** We add the information about the median lines, that are now illustrated bolder. We change seconds to minutes. The qualitive meaning of the colour-coding is now specified in the caption. Yes, the statistics are based on the boot-strapping approach considering hundred positions of sondes per cross-section. In total, this includes 900 cross-sections. The boxes show the quartiles while the whiskers extend to show the rest of the distribution, except for outliers (depicted as markers).

*How sensitive are these results to the length of the flight pattern?*

**Response:** Indeed, the TIVT values are always dependent on the flight lengths. We also compared the TIVT values of the arctic ARs in more detail with mid-latitude observations, where we also point out the different AR widths between arctic and mid-latitude ARs, if one would restrict to the same thresholds defining the outer edges. Regardless the actual AR width on which we also align the flight length, we stick to our recommendation of seven sondes that should be envisioned to be released in order to derive IVT divergence in the three different frontal sectors (pre- and post-frontal, and the core). Since the stronger ARs (in terms of $IVT_{max}$) are also broader, Fig. 6 demonstrates that the minimum required sonde spacing is less sensitive to the actual AR width. However, we admit that robust conclusions in this sense should involve a much higher number of AR events.

**Sec 3.3: Variability of moisture and wind in arctic ARs**

**Response (according to a more detailed discussion) and vague relation to other studies (L266f):** In this section, we also put more emphasis on clearly disentangling results (e.g. Fig. 7) and on discussions that we manifold. We compare vertical profiles in Fig. 7 with those of radiosondes in an arctic early summer AR period studied in Viceto et al. (2022) and synthesize communalities and differences to mid-latitude AR soundings in more detail.

*L274f: How can you see this in Fig.7? Winds also strongly vary and the transport distribution (grey shading) resembles the wind distribution (red shading). The sentence in L275f contains redundant information.*

**Response:** We referred imprecisely to the strong AR case (AR3) represented by error bars and not by the shadings in Fig. 7. We will separate our descriptions for the entirety of ARs and AR3 more obviously and rephrase the discussion of moisture transport variability and the role of wind and moisture:

"The cross-section variability of both moisture and winds strongly affects IVT variability. The shadings in Fig. 7 indicate that the standard deviation of moisture transport resembles the standard deviation of the winds for the lower levels up to 850 hPa, before moisture transport variability is apparently driven by the standard deviation of moisture in upper levels above 500 hPa, although the wind standard deviation here becomes highest. For the most intense AR3, Figure 7 depicts the LLJ with high wind speeds above 30 m s$^{-1}$ that causes strong moisture transport whereas moisture is more or less average. While strong moisture transport in AR3 originates from overall strong winds, moisture varies strongly and seemingly dominates the moisture transport variability. Hence, we can hypothesize more specifically that in strong arctic ARs with intense winds, primarily moisture variability causes IVT variability and leads to to the bell-shaped IV T cross-section pattern (Sect. 3.1)."

*L285f: Do you have an explanation for the increased variability in the free troposphere? Fig. 10 shows that your cross-sections pick up dry post-frontal subsidence regions and also dry Arctic air eastward of the AR feature, which likely impacts this result. Or maybe this is what your last sentence wants to say? How much sense does it make to calculate horizontal means for such heterogeneous sections?*

**Response:** Indeed, the subsidence of dry airmasses is one of the major explanations for the increased variability in q. Since you also refer to Fig. 10, we would like to take up with this question again in Sec 4.4. Here, Fig. 10 enables an illustrative explanation.

Your last remark opens a very crucial discussion that results from the question of what do we consider as the "AR itself"? Here, the scientific perception strongly differs between the horizontal and vertical perspective. In the large majority (and as this study does), moisture transport with respect to vertical integrated quantities (IVT and/or IWV) is designated as AR where a certain threshold is exceeded. Even if the thresholds may change between AR detection algorithm for various reasons, they mostly have in common that they project the AR from 3D (horizontal and vertical) to 2D (horizontal). No matter if different air masses are entrained at certain vertical levels, the domain is still considered as AR, as long as moisture / or moisture transport are sufficiently high in the vertical integral.

The vertical atmosphere may thus still hold two airmasses (dry post-frontal subsidence regions and a moist air mass smaller than the 'plume' in the AR core). How both airmasses interact across this interface is a question for itself. The degree of mixing can have strong impact on cloud and precipitation formation (beyond the scope of this study). However, due to this fact, we pretend that is worth to consider such edges where we find a coexistence of air masses. Also, in the perspective of practical flight planning, forecasts of IWV and IVT represent the quick identification of the AR object to locate the flight tracks in.

*Wouldn't it be more interesting to focus on the AR itself and check how much the fluctuations at small scales contribute to IVT?*

We highlight that we applied our cross-sections with respect to the AR edges from the AR catalogue (Guan, 2022) and can confirm that less than 5% of the flight tracks reach out of moisture transport that is declared as AR. This can also be seen in Fig. 11, where the AR edges (outer edges of the frontal sectors) are partially even more restrictively defined than in Guan (2022). Accordingly, we assure our results are representative for AR internal variability. By our frontal decomposition, we also put more focus of the central AR core.

**Sect. 3.4: Coherence of moisture and wind**

**Response**: we will specify that we mean the correlation of both variables along the cross-section and here now just speak of "connected of pattern". We stated more explicitly that the non-coherent transport consists of the individual means of moisture and winds ($\bar{q} \cdot \bar{v}$).

*Like for Fig 10: This should involve a discussion of the communalities and differences of the presented nine cases. The large case to case variability should be better discussed.*

**Response:** We will restructure the paragraph describing Fig. 10 and insist in more detail on large case-to-case variability with more direct relations to the references given. We will extend the discussion to mid-latitude ARs and focus on the discrepancies to AR schematics as in Ralph et al., 2017 when arctic wind and moisture pattern do not coincide. We highlight on the differences in AR corridors that are closer located to the AR center (e.g. AR5) against those corridors situated in the outflow region (e.g. AR7/9). In this regard, we refer to Terpstra et al. 2021 that also detected missing coincidence in a polar AR outflow, but rather in the vertical axis than we do in the horizontal.

With the added comparison, we specify the role of dry subsiding airmasses that become more effective if there is upper-level advection from Greenland air masses. We note that our analysis from Fig. 10 can still be manifold in various perspectives and details. At a certain point,

however, we need to refocus on our research question and investigate the correlation between moisture and wind for the given cases and how the patterns contribute to IVT and its variability.

**Section 3 specific comments:**

*L225 (maintain?)*
**Response:** Changed to "show"

*L322f: If there is little information from small scale fluctuations, why should one care about supplementary q observations?*
**Response:** Yes, this is valid point. However, if one is still willing to improve the measurements, then one should focus on supplementary moisture observations. Accordingly, we will rephrase L322 to: "An improvement of observing the moisture transport variability should thus rely on supplementary moisture measurements rather than for the winds."

*L260 (why intuitive?),*
**Response:** rephrased to "simplified"

*L268 (behaves more homogeneous?),*
**Response:** changed to: "remains more homogeneous"

*L277 (How?)*
**Response:** rephrased to: "The identification of the more variable quantity out of q and v can be useful for the improvement of measurement strategies for moisture transport. Specifically, moisture can be derived from supplementary remote sensing devices on long-range research aircraft and thus complements sporadic sonde-based data."

*L278 ("long-term aircraft"?)*
**Response:** typo, changed to long-range

*L289-291 (e.g., "carefully correlated observations?", "cross-sectoral")*
**Response:** changed to "collocated" and "cross-section variability"

*L322 ("narrowed moisture columns here form"?)*
**Response:** Rephrased to: "Instead, narrow but high-reaching moisture plumes in the core control the moisture transport variability."

**Section 4: Moisture transport divergence from sondes**

**4.1 Sectoral in- and outgoing moisture transport**

*Fig. 11 "frontal specific AR sectors" is unclear. I […] wonder that the dotted lines at negative distances are warm pre-frontal areas? I do not understand the two sentences "Leg specific … (lines)" – please rephrase. What is "corridor IVT convergence"?*
**Response:** We will rephrase the figure caption: "Figure 11: AR-IVT of inflow (outflow) section in blue (orange) for all nine corridors in the AR. Changes in line styles denote the frontal sector classifications (Sect. 2.5). Dotted lines represent cross-section periods belonging to pre-frontal sectors, while dashed lines refer to post-frontal sectors. The legend specifies TIVT values for the in- and outflow cross-section for the parts situated within the actual AR. They include IVT internal of determined AR borders (Sect. 2.5). Arrows, scaled in length and width, indicate the TIVT difference between in- and outflow leg. As described in Sect. 3.4, they can be viewed as simple estimates for the IVT divergence in between both legs. Upward (downward) arrow scales represent estimated convergence (divergence) magnitudes. Note that the axis orientation has to be mirrored for west-east orientation. "

*It did not become clear to me how the cross-sectional IVT gradients (Fig. 11) are connected to the dynamical situation and the results are contrasting the impression that I got from Fig. 1.*

**Response:** We suppose that the axis orientation confuses the reader. Negative distance values refer to the eastern end, while positive values refer to the western end of our cross-section legs. Thus, the cross-section pattern should be mirrored when comparing with Figure 1. We will add a clarifying explanation in the Figure caption (see above). Then we cannot detect such inconsistencies to Figure 1. For example, the steep decline in the post-frontal sector of AR2 (Fig. 11b) is well in conjunction with IVT pattern shown in Fig. 1. Similarly, the gradients of in- and outflow legs for AR2, AR7 are consistent with the outflow IVT pattern in Fig. 1.

Still, later in the section, we will highlight that a comparison for estimating IVT divergence can be misleading in some cases. Note that the IVT cross-sections are (although continuous) based on flight duration and not on an instantaneous snapshot as we get from Fig 1.

*The TIVT discussion (Fig. 11) could focus more on the AR area: Why is the divergence dominating in what you call warm sector – isn't that surprising?*

**Response:** We will restructure the section: The first paragraph will cover TIVT as a whole, the second paragraph the cross-section differences, the last paragraph will investigate in more detail the TIVT gradients for the frontal sectors in the last paragraph. We remind that all three sectors do only consider moisture transport inside the AR area. We will highlight surprising findings more, but point out the limits in estimating divergence purely from IVT magnitudes. Thus, the two last paragraphs will be composed as:

"Figure 11 further separates the AR cross-sections in the three sectors (pre-frontal, core, post-frontal). Although the AR cores are roughly 200–300 km narrow (slim lines in Fig. 11), they provide more than half of the entire AR-TIVT. This contribution of the AR core agrees with Cobb et al. (2021a) in mid-latitude ARs. Except for AR2 and AR7, weaker slopes of IVT are generally in the cold sector as opposed to the warm sector. The steep post-frontal IVT decline in AR2 and AR7 suggests different evolution processes associated with a high pressure ridge, favored by anticyclonic Rossby wave breaking (Zavadoff and Kirtman, 2018).

Comparing both legs (Fig. 11), some arctic AR cross-section TIVT tend to decrease downstream. Higher IVT and higher TIVT in the inflow leg suggests potential total convergence in the AR corridor. Still, we detect cases with weak stream-ward tendencies in total moisture transport or with slight increases of TIVT. The downstream difference of TIVT is distributed unevenly over the cross-section IVT. We mainly find IVT decreasing towards the outflow leg within the AR core (e.g. AR3, AR9) and thus obtain an impression of convergence. Yet, we occasionally find different behavior in the frontal sectors that partially compensates the core. As in AR6, the overcome-pensating increase of warm sector IVT towards the outflow conveys a seeming divergence in the warm sector. This is in contrast to the findings in Guan et al. (2020), where the pre-frontal sector is denoted as rather converging. Although the IVT pattern of AR5 and AR6 (Fig. 1) may allow slight divergence in the pre-frontal sector, we emphasize that a TIVT-based interpretation of predominant moisture transport divergence underlies strong idealization. Neither it considers moisture flow being not flight perpendicular, nor it does separate contributions of moisture advection and mass convergence. We insist on the regression approach to diagnose moisture transport divergence in each sector of arctic ARs."

*It is sometimes confusing what data is used. I actually thought that the flight duration was not considered for the "continuous" (L394, 426). "Continuous" was also used earlier (see e.g., Fig. 14 caption), however, I think it referred to the high-resolution cross section profiles. I suggest a clear structure and description.*

**Response:** We apologize for the missing clarity of the term "continuous". We will add a clear statement in the method section (2.3.1) that underlines our definition of "continuous" cross-sections. We will refer to the Sect. when speaking of "continuous" in the remainder of the study.

**Section 4.2: Sonde-based divergence and its representativeness**

*missing detail about the related work: L352f, L355f, L358f:*
**Response:** We will specify our reference to related work and will state more precisely how we build on the precedent studies of Guan et al. (2020) and Norris et al. (2020). We will reformulate this as:

"This section depicts the regression-based (Sect. 2.4) IVT divergence *($\nabla$IVT)* in arctic ARs. Moisture transport divergence is specified for the frontal sectors (Sect. 2.5) using the decomposed terms, namely moisture advection *ADV* and mass convergence *CONV* (Eq. 5). We compare $\nabla$IV T in our arctic ARs with those based on statistics for mid-latitude ARs in Guan et al. (2020). The results we obtain from the continuous cross-section flight legs (Sect. 2.3) interpolated from CARRA represent our idealized reference. For them and seven synthetic sondes per cross-section (as in Fig. 4), we apply the regression method to derive *ADV* and *CONV*. In doing so, we build on the Norris et al. (2020), who pioneered the airborne derivation of all moisture budget components, including moisture transport divergence, by investigating a mid-latitude AR event. With our framework, we can assess uncertainties of sonde-based determination of $\nabla$IVT in arctic ARs."

Erratum modifications: As mentioned in the beginning, the divergence results have changed due to the correction in the vectorized divergence calculation code. The replaced figures will be illustrated below, with bullet points highlighting differences to the preprint version.

[Figure]

Figure 12 in manuscript): Vertical contributions from *ADV* (a) and *CONV* (b) to the moisture transport divergence (c) for the frontal sectors in AR3. Bold lines represent sonde-based values while filled areas denote the deviation to values based on continuous AR representation. **Comment: the additional tick-labels will be removed in the final version**

**Changes to preprint results:**

- Slightly smaller magnitudes in all components
- warm sector and core values became more positive (less moisture advection and mass convergence). In particular, the warm sector became rather divergent (Fig. 12c).
- Low-level mass convergence in the cold-sector (exhibited divergence before).

Explanations: Before, we did not consider the u and v components of the wind separately. This directionality, in turn, substantially affects the results. Since the winds are partly elongated with

respect to the moisture pattern, we find relevant contributions of cross-section parallel gradients (e.g. in $u \cdot \frac{\delta q}{\delta x}$ and $q \cdot \frac{\delta u}{\delta x}$), that were not pronounced in the previous approach. The current dominant post-frontal mass convergence results from strong changes in wind direction and superimposes rather weak changes in wind speed.

[Figure]

*Figure 13: Box plot of moisture transport divergence contributions to daily moisture budget for all nine ARs specifies both components (ADV, CONV) in each frontal AR sector. The continuous AR representation (coloured box-whiskers) is compared to sonde-based values (grey box-whiskers). Boxes refer to quartiles and horizontal lines specify the respective mean.*

**Changes to preprint results:**
- Smaller magnitudes in moisture budget contributions than before
- Less frontal gradient, meaning less positive contribution in the pre-frontal sector and less negative contribution in the post-frontal sector
- Mass convergence does not exhibit a clear gradient along the front. In particular, the post-frontal sector now shows the strongest mass convergence against all the other sectors (before divergent).

We will update the text in Sect. 5.2, accordingly. To account for the findings of related work in a more concise way, we will separate the paragraphs that describe our results from the respective discussions referring to mid-latitude statistics given in Guan et al. 2020. Surprising findings, such as the mass divergence in the core are carved out.

**Section 4 specific minor comments:**

*L333ff (What is the "simplified understanding of divergence"?, "benchmarks…"?)*
**Response:** We merge both expressions in a connected sentence as: "The comparison of *TIVT* in both legs reveals first simplified benchmarks of the prevailing divergence. Idealizing that no entrainment into the AR corridor (Sect. 3) takes place, Figure 11 contrasts TIVT of in- and outflow cross-section to estimate whether convergence or divergence of moisture transport exists inside the AR corridor.

*L358 ("behave differently"):*
**Response:** formulation rephrased to: "*ADV* and *CONV* exhibit different vertical profiles throughout the frontal cross-section"

*L360 ("lower atmosphere")*

**Response:** changed to lower troposphere

**Response:** The respective sentence will be removed because not valid for updated results

**Response:** We will reformulate the sentences to: "When we place our sonde results in the context of the airborne study by Norris et al. (2020) using real dropsondes, we recognize the strength of true sondes with a high vertical resolution, which provides much greater vertical variability. Thus, the quite low divergence displayed in Fig. 11 is likely not only due to less divergence prevailing in Arctic ARs compared to mid-latitude ARs, but may also result from the coarser vertical grid that average out larger values."

**Response:** changed to: "for our sample of arctic ARs"

**Section 5: Deterioration by non-instantaneous sounding**

Like in Section 4, the divergence results for the instantaneous perspective have changed, causing modified figures. Both figures are displayed here equivalently.

[Figure]

*Figure 1: Comparison of divergence component contributions to daily moisture budget from the continuous AR representation referring either on the time-propagating flight values or when using the values for the centered hour. Values are given for each frontal sector. Black error bars are identical to the coloured boxes in Fig. 2 (13 in manuscript). Grey values represent the centered hour based values.*

**Changes to preprint results:**

- Instantaneous whiskers follow the modified frontal tendencies (smaller magnitudes and weaker gradients along the front)
- Mean errors in contribution (red-dots are less affected) by updated divergence calculations the post-frontal sector
- Highest mean error in cold sector advection remains. Robust dry advection visible in instantaneous view on post-frontal sector

[Figure]

*Figure 2: Total (orange) and individual errors only by discrete sondes (green) and only by instationarity (grey) for daily IVT divergence in each frontal sector and divergence component (Eq.). For all AR cross-sections, positive bars indicate the root-mean square error while error markers and lines depict mean errors in combination with their standard deviations.*

**Changes to preprint results:**

- Minor changes in magnitudes
- Equivalent key messages: Non-instationarity counts more than sounding frequency for the sonde-based misrepresentation of moisture transport divergence

**Section 5: Specific minor comments**

*L403f (unclear)*
**Response:** changed to: **"**Within the AR corridor, the temporal AR evolution can distort the airborne representation of Eulerian IVT divergence."

*L485 (unclear)*
**Response:** changed to: "Contrasting in- and outflow TIVT through the AR transects, we overall expect divergence in moisture transport."

**Section 6: Summary and Conclusions**

*[…] should synopsize and synthesize the key results and identify the contribution to research on Arctic ARs. I think it will strongly profit from an improved discussion of the results.*
**Response:** In the following, we will list how we will improve our conclusions in terms of clarity in structure and the discussions of our key results. We refer to your more specific comments:
*So far, the first paragraph is a repetition of what was done. The second paragraph claims that higher resolution reanalyzes increase our understanding of arctic moisture transformation and precipitation efficiency, which I don't see is addressed.*
**Response:** Our reason for the summary of our synthetic framework is to remind which perspective this feasibility study has chosen. We are confident, that such a repetition of the synthetic approach facilitates the readability in later discussions, as well as the interpretation of our conclusions. However, we admit that we listed to many details and will erase to detailed information (e.g. the regression method used).
In accordance with the remarks of reviewer 2, who suggests to put more emphasis on the general arctic AR conditions rather than airborne perspective, we will restructure our conclusions. First, we will synopsize the basic AR IVT characteristics in the Arctic we found

before specifying the airborne perspective. By that, we can also get rid of the second paragraph which lacks clear structure and key messages. Instead, we summarize our considered IVT magnitudes in the Arctic. This comprises IVT shape, AR widths, maximum IVT, how they differ to mid-latitude ARs and specify how CARRA outperforms the IVT variability compared to ERA5. By that, we aim to achieve a more logical order, when approaching the concrete perspective of our study (the observability of arctic AR IVT divergence by dropsondes). Here, we then find appropriate to repeat how the synthetic soundings were established to answers of our research questions.

*The authors should try to better synthesize the central message of their results to each of the RQs in view of the gained knowledge. I suggest a separate discussion of how the obtained results may affect future flight planning and the deployment of dropsondes (Q1, Q4).*

**Response:** We will update the central messages to our research questions and conduct a stronger connection to individual flight planning which is achieved by an improved structure. We will extend the specifications in our recommendations for future dedicated flight planning. We here give a suggestion on how we plan to modify the bullet points for Q1: "For the sonde-based determination of Total Integrated Water Vapour Transport (TIVT) in arctic AR cross-sections, sonde spacings below 100 km have to be envisioned to certainly keep TIVT errors below 10 % (Fig. 6). In strong ARs with IVT exceeding 500 kg m$^{-1}$s$^{-1}$, too coarse IVT representation at the AR core leads to TIVT underestimation. Gaussian fits help reproduce the cross-sectoral IVT shape but are sensitive to how sondes estimate maximum IVT and its location. Precedent flight planning should thus aim for a sonde release at forecasted IVT maximum and place the additional sondes symmetrically around. For the arctic AR widths of 400-800 km we found, we suggest a minimum of seven soundings per cross-section (roughly 60 to 120 km spacing) to derive TIVT in both cross-section legs. Not necessarily, larger AR width is associated with higher IVT variability, maximum IVT is more correlated to IVT variability causing TIVT errors. The planning of sonde releases should also rely on the steepness of IVT along the cross-section. Furthermore, we highlight that the meridional differences of TIVT between the in- and outflow cross-sections remains at 2-15% of the TIVT magnitude (Fig. 11). Therefore, estimates of moisture transport divergence using TIVT from both cross-sections only become robust, if the TIVT uncertainty for a single leg is considerably lower."

We already gave some suggestions for future flight pattern in the preprint's conclusion, but in a poor structure. We will improve the structure of our concluding implications on flight planning emerging out of the conclusions from Q1-Q4 as:" Overall, we confirm the observability of moisture transport divergence in arctic AR corridors by releasing sondes in such dedicated flight patterns. A maximum sonde spacing of 100 km within the AR cross-section can principally characterise the divergence between both cross-sections at the given uncertainties of ≤ 10 %. For the duration needed to perform the flight pattern, we obtain the entire moisture transport divergence specified for the frontal sectors with an uncertainty lower than 25 %. For the frontal-sector investigation, we deduce that sonde undersampling matters and recommend a sequence of at least seven sondes per cross-section, given the widths of arctic ARs this represents a sonde spacing of maximum 100 km. Yet, notwithstanding that we could release a much higher number of sondes, it is the temporal AR evolution over flight duration that leads to higher deviations in divergence components rather than sonde undersampling. Thus, dedicated planning of such sonde-based observation purposes should not only involve dropsonde positioning but rather pursue minimizing the flight duration. The placement of the cross-section legs and their separation should carefully consider the AR displacement during flight. Lower meridional distances between the cross-sections do not only shorten the flight duration, they also reduce the area which is enclosed by the sondes. For the AR and frontal-sector widths found in the Arctic, the two cross-sections should not be more than 100-200 km apart. For several of our larger AR corridors, we have to expect substantial sub-grid scale variability in the flow parallel direction. Therefore, we postulate collocated flights by two aircraft where both cross-sections are not far apart and are sampled simultaneously as the optimum and still feasible strategy. When faced with a limited amount of dropsondes, supplementary measurements of moisture should be prioritized, as moisture represents the more varying

quantity in our AR and moisture advection is mostly dominating the moisture transport divergence in the arctic AR corridor."

After these implications we will add a description of the limitations of our study that covers several points mentioned in your remarks (vertical resolution of CARRA used, limits of regression approach) that needed to be discussed more thoroughly and we will mention specific suggestions for arctic AR follow-up studies using CARRA.

Finally, we summarize the necessity of assessing the sonde-based observability and of deriving uncertainties for model-observation intercomparison.

---

## Author Comment (AC2)

**Response to the comments from Anonymous Referee 2 for the submitted ACP paper: ´Dorff, H. et al. 2023: Observability of Moisture Transport Divergence in Arctic Atmospheric Rivers by Dropsondes**

**Superior Erratum:**

With the aid of the reviewer's remarks concerning the frontal gradients in moisture transport divergence and the emerging revision of our manuscript, we identified erroneous results in our divergence calculation. In specific, we accidently did not calculate the wind divergence from using both u, v components but considered the absolute values of wind speed. By that our divergence results were direction-independent. In order to conduct a component-wise divergence calculation, we now had to rotate the u, v components in CARRA, as they are oriented along the local grid rotation and not the zonal/meridional direction. In doing so, the results in chapter 5 and chapter 6 (Figure 12-15) have changed moderately. In the response sections that refer to the respective manuscript sections, you will find the updated figures, alongside a specification of differences to the preprint results. In the remainder of this response, our updated results are already included when we present corresponding snippets of the updated manuscript paragraphs.

**Prefaces:**

We thank the ACP associating editor, Geraint Vaughan, as well as the Anonymous Referee #2, for this inspiring review. Please find below our responses (in standard font) to the remarks from the Anonymous Referee #2 (in *italics*). We structured this response in such a way that comments on the most important text blocks for improvement (e.g. motivation) are bundled and distinguishable from each other. In this response, we occasionally refer to our Author's responses for Referee 1 (AC1), because several answers in AC1 also consider the remarks given from Referee 2 and we intend to avoid too much repetition.

We reserve the right to apply minor changes to the here modified text snippets for the final revised manuscript in order to achieve even more concise phrasing and to guarantee grammatical correctness.

**Responses to Reviewer 1:**

**General**

*This paper provides a contribution to advancing our understanding of arctic atmospheric rivers by presenting an analysis of them using different reanalysis products, and suggesting ideal targeting strategies for the purpose of understanding and closing the moisture budget.*

**Response:** We want to thank you for the detailed and inspiring feedback. Given your remarks, we realize that the perspective of our study, in particular, should be carved out more clearly. We are confident that such a specification enables to significantly improve the manuscript readability and clarity. Accordingly, in our revision, we focus on improving the readability by a more elaborated structure which provides more precise motivation for sampling of ARs from airborne dropsondes.

**Major**

*In general, I think it is important for this paper to provide some additional context and motivation for the exercise of synthetic sampling.*

**Response:** We agree that the preprint lacks a well-elaborated motivation for our sonde sampling approach with respect to arctic ARs. Therefore, we will carefully rewrite the introduction which, based on the reviewer remarks, we identified as one of the major weaknesses of our manuscript.

In detail, we specify our changes in structuring the motivation in the Author's Response for Referee 1 (from now on denoted as AC1). At this point, we summarize some of the changes that we declare as most relevant according to your review:

- The first paragraph will state more explicitly the presence and impact of Atmospheric Rivers (ARs) in the Arctic. Given your recommendation for more information about Arctic conditions, we will now refer to a broader collection of studies with respect to the Arctic and, in particular, concretize their findings relevant for our motivation, rather than only list relevant literature (as done before). The second paragraph characterizes the moisture transport (IVT) and its divergence in ARs, how this becomes relevant for the transformation of moist air masses, and how IVT shapes the regional moisture patterns in the Arctic (Nygard et al., 2020). Finally, it disentangles the issue and research gap, which is the lack of quantitative estimates for IVT divergence within Arctic ARs.

- "*Is there a possibility for in situ sampling of arctic ARs? Is the paper calling for this capability as a requirement for us to meaningfully further our understanding in this region?*" **Response:** As stated before, the second paragraph serves as a logical transition to the following paragraph describing the required measurement strategies to derive IVT and highlighting the need of in-situ sampling to obtain the moisture transport throughout the troposphere. The third paragraph will thus address your point in more detail: We mention that the observational radiosonde network in the Arctic (Dufour et al, 2016) allows the derivation of IVT divergence into the Arctic, but argue that this network is too coarse to resolve IVT variability and divergence within single AR events. This motivates the use of dropsondes from research aircraft specified in the following. We are confident that this argumentation better highlights the ability of in-situ sampling which serves as a prerequisite to be able to meaningfully expand our knowledge of moisture transport in arctic ARs.

- Both referee reviews demonstrate that the manuscript requires an improved motivation for our choice of a synthetic sampling approach. Therefore, the research focus and overarching motivation of our feasibility study are introduced earlier (beginning of 4th paragraph). In particular, we mention that there currently exist arctic airborne flight campaigns using a long-range research aircraft proposed by Wendisch et al. (2021). This fact motivates a pre-assessment of the sonde-based observability of moisture transport divergence in arctic ARs. Not only, this will improve the interpretation of sonde-based observations gained in the HALO-(AC)3 campaign (Wendisch et al., 2021; Walbröl et al., 2023) which are currently under processing, our feasibility study also aims at facilitating future flight mission planning that has a similar special focus on high-latitude ARs, e.g. like the NAWDIC campaign:
https://internal.wavestoweather.de/campaign/projects/nawdic/wiki)

- The changed order in which we will introduce our research questions sequentially, rather than at the end of the introduction as in the preprint, is intended to strengthen our argumentation. In doing so, we expect that this leads to more clarity why we have chosen a synthetic approach in investigating the arctic ARs.

*How would this papers' findings be different if the synthetic sampling wasn't a part of it? Does this framing potentially distract from the findings regarding the structure of arctic ARs?*
**Response:** We fully agree that a study purely dealing with the structure of arctic ARs using reanalyses or model simulations will represent a very fruitful scientific contribution. Nonetheless, we admit that we, the authors, are mostly situated in the observational scientific community and are confronted with the necessity of airborne data in arctic ARs. Since we do expect several studies to emerge from the previous HALO-(AC)³ flight campaign and upcoming campaigns with respect to arctic/ high-latitude ARs, we certainly see a benefit in our approach for future studies. For instance, in the HALO-(AC)³ special issue of ACP, we envision a contribution of the novel airborne derivation of all moisture budget components (including IVT divergence) in an arctic AR that was observed during HALO-(AC)³ (Walbröl et al., 2023). For this, our feasibility study can serve as a preparational study that quantifies the magnitude of airborne misrepresentation in sonde-based moisture transport divergence. Correspondingly,

our title was chosen in a way that it immediately becomes clear that our focus is in the observability.

Nonetheless, we will take your valid suggestion into account in the manuscript. Our conclusions will now emphasize the ability of investigating arctic ARs in a more general perspective using CARRA. We will promote follow-up studies that attribute dynamic and thermodynamic conditions to the AR characteristics: "[…] since we include a large variability of synoptic AR patterns but a small sample, we suggest statistical analyses involving a larger amount of AR events. Such statistics can foster our understanding of moisture transport divergence pattern in arctic ARs and attribute them to the dynamic and thermodynamic atmospheric conditions. For such follow-up purposes, CARRA represents a very suitable reanalysis framework […] "

*I suggest the authors consider strengthening their case for structuring the paper in this way and referring to more papers studying arctic ARs and their structure in addition to observational studies covering the midlatitudes if they would like to keep this framing.*

**Response:** We will restructure our introduction in this way (see above). We will include more literature findings from arctic ARs, not only in the introduction but also when comparing our results to polar AR characteristics investigated in other studies, such as Terpstra et al. (2021); Viceto et al., (2022); Lauer et al (2023). In this regard, please find detailed manuscript modifications in the AC1 (especially for Section 3 and 4). In compliance with the remarks from Referee 1, we elaborate on the results concerning the general structure of arctic ARs in more detail in the respective sections, before moving on to the sonde-based representation. This also applies to the presentation of our arctic AR cases, where we manifold the discussion of the synoptic conditions causing the ARs to reach the Arctic (see AC1, Sect. 2). In Sect 3.1 (originally Sect 4.1), we compare the arctic AR-IVT shapes in more detail with those from mid-latitude cases. We use this knowledge as a prerequisite for the following examination of the sonde-based AR-IVT representation. Not only for general AR characteristics, but also for the sonde-based representation, we will enlarge and concretize the comparison to mid-latitude based studies (e.g. in the relabeled Sect 2.2, 3.1, 3.4, 4.1, 4.2). You can find concrete examples given in AC1 that consider your comment of including more analysis of the phenomenology of arctic ARs in contrast to mid-latitude ARs. By this, we aim to consider your comment of including mid-latitude based AR knowledge. Yet, we would like to keep our original framing/scientific perspective in principle. Nonetheless, we already see a clear progress of the manuscript in providing more details of the general arctic AR characteristics. These details will additionally improve our argumentation and discussion in our sonde-based assessment.

*I suggest considering a reframing where the authors discuss what can be learned about arctic AR structure from appropriate reanalyses at different resolutions, and then recommend sampling strategies to verify/supplement this knowledge.*

**Response:** For our purposes, we see a certain risk in changing the whole structure in this direction, because then the main objective of this study (which is the assessment of sonde-based IVT observability and uncertainties in the sonde-based representation) would be underrepresented or the paper could become too long and overloaded. Instead, we will sketch the impact of the reanalysis resolution on the arctic AR structure in more detail in Sect. 3.1 (originally 4.1) and will refer to the current study of Viceto et al. (2022) in which they conducted a reanalysis comparison in a case study of arctic ARs. This will be included in two paragraphs as follows:

"[…] we recognize the bell-shaped IVT inside the AR from both, CARRA and forcing ERA5. Within the cross-section centre which we declare as the AR core (Sect. 2.5), CARRA, however, shows stronger moisture transport with a more pronounced IVT maximum > 500 kg m$^{-1}$s$^{-1}$. Moreover, CARRA with its higher horizontal resolution depicts more small-scale structures of the AR moisture transport. In particular, CARRA increases the cross-section IVT variability for this case."

"[…] Viceto et al. (2022) further documented the improved representation of arctic AR characteristics in ERA5 against coarser reanalysis data. In our comparison, the location and horizontal pattern of the ARs in CARRA agree quite well with ERA5 (not shown). For all crosssections, we ascertain plausible IVT values from CARRA with respect to ERA5. In particular, we highlight that maximum (mean) values of IVT per cross-section increase by roughly 9 % (8 %) from ERA5 to CARRA on average. CARRA further increases the IVT variability by roughly 11 %. We attribute this to the higher horizontal resolution than in ERA5."

*Do your results regarding non-instantaneous sampling change if you take into account the observations in time and space where and when they occur, as is possible in many assimilation systems now?*

**Response:** To prevent any misunderstanding, we remind that we interpolate the reanalysis data in time and space onto the flight track and compare this to the reanalysis output at centered hour (denoted as instantaneous snapshot).

Current methods to derive divergence from airborne soundings require atmospheric stationarity (e.g. Bony and Stevens, 2019), which in turn can only be idealized in observations. To circumvent this issue, Norris et al. (2020) conducted a time-to-space adjustment of their sonde profiles. In our study, we aim to address the impact of instationarity for our divergence calculations by research question Q4. Here, we clearly see the limitations in the sonde-based derivation of moisture transport divergence from our flight pattern.

Still, your point is a very useful recommendation for future steps in improving the regression methods to a multivariate regression involving the temporal component. For our purposes in which we refer to the state of the art in the calculation assumptions for the observations, assimilation methods remain out of scope of this study.

From the recent flight campaign HALO-(AC)3, we know that the dropsonde data has been integrated into the Global Telecommunications System and used for assimilation in ERA5. The upcoming investigation of the HALO-(AC)³ observations in arctic ARs should certainly take this into account. So far, we can only speculate about the outcome but see a very good agreement between ERA5 and the dropsondes due to the assimilation.

*I very much like how the authors identify key questions and then revisit them in the summary with their answers to synthesize the paper for the readers.*

**Response:** We are glad that the key questions are considered as suitable guideline throughout our manuscript. Under consideration of the remarks from Referee 1, the motivation improved unravelling and identifying our key questions. As mentioned in AC1, the synthesis of our answers now has a more precise focus on implications for future airborne measurement strategies.

**Minor**

*Line 24 – flood may not be the best word choice here, please revisit ("affect"?)*

**Response:** We will rephrase the first sentence and also include a concise definition of Atmospheric Rivers (ARs) as follows:

"Atmospheric rivers (ARs), which are elongated (>2000 km in length) but narrow (<1000 km in width) water vapour rich corridors causing high moisture transport, occasionally enter the Arctic."

*Figure 1, Figure 4: locate us in space with lat/lon*
**Response:** We will add a lat/lon grid accordingly

*Figure 3 – suggest including a box in (a) to illustrate where the box in (b) comes from.*
**Response:** We include a slight rectangle in top view illustration (a) and renamed the boxes in (b) from "sections" to "corridors" in order to guarantee consistency between both subfigures.

*Line 252 – does this suggestion of 5 sondes at minimum depend on the AR width?*
**Response:** Concerning both Referee remarks, we put more focus on actual sonde spacing (in distances) rather than number of sondes (see also AC1, Section 3.2). Therefore, we reformulate: "we emphasis that a minimum sounding spacing of 100-150 km has to be targeted for arctic ARs." This corresponds to 4-8 dropsondes for most of the AR widths in the range of

400-800 km. Like we will add in the respective discussions, the sounding spacing is not only affected by the AR width, but also by the steepness of IVT. For example, in wider but weaker ARs, a spacing of 150 km may be sufficient to accurately derive TIVT, but in a narrower but stronger AR (greater steepness of IVT) we recommend a spacing of at least 100 km.

*Figure 6 – suggest this would be better presented as spatial interval to not require so much information regarding assumptions about plane speed etc. Indicate what the colors represent in the caption.*
**Response:** The x-axis is changed to spatial intervals. Further information is given in AC1.

*Line 268 – isn't this larger difference in q expected given the colder air?*
**Response:** Yes, it is not surprising. Still, we want to emphasize the contrast to the winds that are in a comparable magnitude of order as in mid-latitude ARs. We will rephrase it as follows: "q is considerably lower due to the cooler air masses."

*If the AR is more moist or more windy, does that affect the spacing requirements to fully capture the structure?*
**Response:** So far, we cannot disentangle clear statements that a more windy AR requires more dropsondes than a more moist AR or vice-versa. But how we explain for our strongest AR event (AR3), the winds here were rather constant (and high) along the horizontal AR transect while the internal moisture plume was primarily responsible for the moisture transport variability. As an outlook, upcoming studies could investigate this in more detail in a larger sample.

*Line 323 – constant winds in time or in space? Can you refer to one of your figures here?*
**Response:** Here, we speak of the winds along the cross-sections, but will specify it correspondingly in the revised manuscript.

*Figure 13 – what is the purpose of the colors in the box-whiskers plot?*
**Response:** the colors refer to our colour-coded frontal sector composition as shown in Fig. 4. We will add this information in the Figure caption.

**Editorial**

*A general quick read for grammar/word choice (clarity)/readability is warranted although generally the paper is in good shape. A few suggested changes are below (non-exhaustive).*
**Response**: For its final form, we will conduct careful cross-reads to assure clarity and readability, and correctness in grammar and word choice.

*Line 214 – suggest changing "infer" to "investigate"*
**Response:** we will change the wording accordingly.

*Line 219 – suggest rephrase "arises the question, how" to "raises the question whether".*
**Response:** We will change it correspondingly.

*Line 264 – suggest rephrase "contributes to IVT with roughly 50%" to "contains roughly 50% of the IVT magnitude"*
**Response:** We will change it accordingly

*Line 265 – remove "even"*
**Response:** confirmed

*Line 321 – suggest rephrase "are little coherent" to "exhibit little coherence"*
**Response:** we will reformulate it accordingly

*Line 348 – suggest rephrase "neither it considers" to "it considers neither"*
**Response:** we will reformulate it accordingly

*Figure 11 – suggest removing "corridors in the" from the caption*
**Response:** We agree and will remove it from the caption

---

## Author Response (AR1)

**Point by Point Response for Manuscript, entitled 'Dorff, H. et al. 2023: Observability of Moisture Transport Divergence in Arctic Atmospheric Rivers by Dropsondes', for final publication**

**Table of Contents**

**I) Response and Changes to the Comments from Anonymous Referee 1 (AC1)**

We thank the ACP associating editor, Geraint Vaughan, as well as the Anonymous Referee #1, for this enlightening review. Please find below our responses (in standard font) to the remarks from the Anonymous Referee #1 (in *italics*), which were given very similarly in the letter of responses uploaded before. The changes/modifications in the manuscript are specified below. Frequently, the major changes consist of rewriting several paragraphs. Citing such paragraphs would partially be too long in this report. In these cases, we only refer to line numbers in the revised manuscript (**bold**). Preprint line numbers of minor comments (grey) now refer to the lines in the revised manuscript (**black, bold**).

**Responses to Reviewer 1:**

*The article is a comprehensive piece of work and presents nice and illustrative figures. I think the presented approach is valid and the content is certainly worth for publication. However, […] the study is not sufficiently motivated and the results are not properly discussed in view of related work. Hence, the novelty in terms of applied methods and results and the added knowledge about Arctic ARs and their observation strategy remains unclear. The presentation quality suffers from a confusing writing style. I recommend major revisions and encourage the authors to carefully rewrite their work to improve the readability of the manuscript.*

**Response:** First of all, we want to expressly thank you for the very detailed and well-specified feedback. We are certainly confident that the consideration of your remarks enables a significant improvement of the manuscript. Your remarks help to transfer our scientific content and knowledge, that is considered as worth publishing, to the reader in a more logical and precise way. Accordingly, in our revision, we focus on improving the readability by more elaborated clarity and structure.

**Writing/Grammar**

*The grammar is a bit awkward and the article misses coherence and logical order within the paragraphs and sentences. The writing of this paper would benefit from a grammatical editing and language check.*

**Response:** We will invest more focus on coherence and the logical order for the paragraphs (individually and in its entirety). We will confirm additional cross-reading of the revised manuscript by either well-experienced or native English speakers.

**Terminologies**

- *the authors skim over many aspects simultaneously and it is up to the reader to guess potential relationships. Terms like "significant impacts", "pathway of ARs" (in a Lagrangian sense or AR displacement?), "moisture transformation processes", "moisture budget", "precipitation efficiency", "divergence of IVT", "IVT variability", "horizontal corridors", "dynamical and thermodynamical processes", "AR moisture budget components", "AR evolution" (many more examples in the other sections) are not defined or described, which makes it hard to understand the context of this work.*
- *Formulations like "are widely assessed over the mid-latitudes" (L33) or "manifold understanding" (L35) are without substance and should be avoided.*

**Response:**

- We agree that the introduction, in particular, was overloaded with terms defined/ described little or even not at all. You will find more details in our responses concerning the introduction (see below):

*"significant impacts"* → The impact of ARs in the Arctic is now the key topic of the first paragraph and thus specified. **Modification: Term removed**

*"pathway of ARs"* → the ambiguous formulation has been modified to specify the long-distance displacement where air mass transformations occur.

**Modification (Line 33f): "Along the long-distance displacement, the AR embedded moist and warm air masses are subject to substantial air mass transformation (You et al.,2022)."**

*"moisture transformation processes"* → we decided to keep them more or less unspecified. Transformation processes, such as airmasses starting to precipitate, are described above (although not linked to the term "transformation process"). Now we refer to relevant literature (You et al., 2022) and more clearly highlight the relevance of moisture transport to understand air mass transformation processes. Otherwise, we see a risk of distracting the reader more from our major research focus being the moisture transport and its observability if we specify the air mass transformation much more at this early stage.

*"moisture budget", "precipitation efficiency", "divergence of IVT"* → In order to not confuse the reader by too much details, we erased the term "moisture budget" and focused on the fact that Seager and Henderson et al. (2013) finds the link between moisture transport divergence to local tendencies of moisture amount and how efficiently precipitation is triggered.

**Modification in Manuscript (Line 37f): "Seager and Henderson (2013) point out that the divergence of IVT links the local temporal evolution of moisture amount to the efficiency of precipitation induction."**

Nonetheless, in our opinion, defining each of the terms once again can be neglected in some cases. In particular, we consider the conceptional definition of divergence (convergence) to be known. A concise definition of the moisture budget follows in Section 2 using Eq. 1.

*"IVT variability"*

We agree that this wording was imprecise. We specified that we mean the spatial variability and added respective findings from Guan et al (2015) describing how spatial IVT variability is composed in atmospheric rivers.

**Modification (Line 39f): Guan and Waliser (2015) considered global ECMWF Interim reanalysis (ERA-Interim, Dee et al., 2011) to specify strong moisture transport gradients that exist along AR cross-sections, perpendicular to the major IVT orientation. Along such lateral cross-sections through the AR center, a bell-shaped IVT distribution, having the strongest moisture transport in the AR core.**

*"horizontal corridors"*

We rephrased this.

**Modification in Manuscript (Line 54f):** "horizontally extended areas".

*"AR corridor"*

Later, the term will be defined using Figure 3.

**Modification (Line 193, 198ff).**

*"dynamical and thermodynamic processes",*

We carefully checked the respective phrase, but argued that too much specification here might distract from the key message of this phrase and paragraph, namely the frontal gradients in moisture transport divergence.

*"AR moisture budget components"*

we reformulated the sentence without explicitly mentioning the moisture budget components and solely refer to IVT divergence. By that, the term "AR moisture budget component" is not needed before it is introduced in Sec 2.3.

**Modification in Manuscript (Line 47): "Furthermore, the observation of moisture transport requires simultaneous measurements of winds and moisture for the entire troposphere."**

*"AR evolution"*
We specified that we mean the evolution in a temporal sense, meaning the life cycle of an AR and the AR displacement, that cause Eulerian differences over time.
**Modification in Manuscript (Line 96f): "Over the flight duration required to enclose an AR corridor, the temporal evolution of the AR (its life cycle and spatial displacement) can affect the sonde-based results significantly."**

- **We deleted any formulations like "widely assessed"** *(L33)* **or "manifold understanding"** *(L35)* **that impair the argumentation in the introduction.**

**Introduction:**

*The motivation […] remains unclear. I understand that a limited number of dropsondes might affect IVT estimates and […] that Arctic ARs may be not well characterized, but is that all?*
*The authors remain very vague about related work. Although references are given, only rarely a relevant result is described. […] this is needed to understand the motivation of the study.*
**Response:** We suppose that the claims results from our insufficient clear structuring in specifying the motivation and in the identification of the research gaps that emerge from broadly studied airborne observations of ARs (at least in the mid-latitudes). When referring to literature, we mostly miss to pinpoint clear findings relevant for our motivation.
Therefore, we carefully rewrite the introduction to clearly identify research gaps and include the following modifications, addressing several reviewer remarks:

**Modifications Introduction:**

- We rephrased the first sentence of the manuscript that actually intended to highlight the presence of ARs. Following Referee 2, a *"clear and concise description (definition, shape, evolution, region of occurrence) "*of ARs is now added:
  **"Atmospheric Rivers (ARs), which are elongated (> 2000 km in length) but narrow (< 1000 km in width) water vapour rich corridors with high moisture transport, occasionally enter the Arctic."**
- The purpose of the first paragraph, which is about the importance of the presence of ARs in the Arctic, is described in more detail. **(Line 24ff)**
- We conducted a more thorough literature review regarding arctic ARs. We stated findings from the literature more precisely. In particular, in the first paragraph that focusses on the importance of ARs in the Arctic on short-term scale, but also at many other places throughout the manuscript. **(not shown here, as distributed throughout entire manuscript)**
- The research focus and overarching motivation of our feasibility study are introduced earlier (4th paragraph, line 56ff). This facilitates the understanding of our conceptual perspective (assess the observability of AR to improve flight campaigns) in the following.
- **Our four research questions are now motivated with respective paragraphs one after another. Each paragraph faces a relevant problem/requirement of airborne moisture transport sounding and refers to knowledge gained from ARs in mid-latitudes. (Line 62ff).**
- *Why should ARs and their observation strategy be different in the Arctic? What is known about Arctic ARs in general and dropsonde-based IVT estimates?*
  Current studies so far mainly focused on IVT variability within mid-latitude AR centers. For the Arctic, where we do expect less IVT, the sea-ice region is more affected by the outflow region of long-elongated ARs residing over the North Atlantic than by the center of an AR overpassing this region. It can be argued that if IVT and its variability are

supposed to be weaker in the Arctic, then the requirements based on mid-latitude ARs will apply, or even more so, to Arctic ARs. However, the deployment of dropsondes is cost-intensive and should be always optimized in a cost-loss ratio. This should be quantified in such a way: How few dropsondes are sufficient to characterize IVT in Arctic ARs and what is their uncertainty?

Regarding the frontal-specific characteristics of divergence in arctic AR, Guan et al. (2020) and Norris et al. (2020) determined AR frontal differences of moisture transport divergence for mid-latitude ARs. However, we do not know how the divergence/convergence of moisture transport takes place in arctic AR regions. Guan et al. (2023) found that the Arctic is more affected by mature ARs that commonly start to dissipate and by the outflow regions of ARs. Correspondingly, such facts are elaborated more thoroughly in the introduction to better motivate Q3.

**Modifications in manuscript: According to the restructuring such points are addressed in the respective paragraphs in Line 62ff and 81ff.**

- *How has the problem been addressed (methods)?".* We give more relevant information from the literature (e.g in how the sensitivity to sonde spacing has been assessed already). However, in our opinion, too many details on specific methods to derive IVT divergence should be postponed for the sake of readability. When picturing the remainder of the manuscript, we mentioned more explicitly that an entire section introduces how divergence can be calculated from airborne sonde profiles and how, in detail, it is done in this study.

  **Modifications in manuscript (Line 110f): "After introducing our AR cases, Section 2 describes the methods emulating dedicated flight pattern and synthetic soundings, and how we derive moisture transport divergence."**

- *How and why were the particular nine cases selected (unclear: L68 "predefined in ERA5", L119ff "picking (…) from catalogue") and why was only the Atlantic region considered? Please explain the purpose of placing the legs at the sea ice edge (L105ff is unclear). Why is only spring considered?* The selection is related to the AR impact on the sea-ice melting when sea-ice reached its maximum extent in spring (paragraph 1). We specify the Arctic ocean as our region of interest due to the fact that the North Atlantic represents one of the most prominent pathways for ARs (Guan et al., 2023) into the Arctic as the moisture transport is undisturbed by any orographic barriers.

  **We removed the unprecise information about ERA5 *(Line 68)*. Instead, we revisited the explanation of the case selection in the section introducing the AR cases (Sec 2.2). We added more details (Line 155ff).**

  *Observation strategy: How was the simulated observation strategy defined? How do aircraft limitations (flight duration, number of dropsondes) affect the strategy?*

  **Response:** The different characteristics across the embedded AR front require a large area of interest and the flight duration consists of a couple of hours, in contrast to single circles (being performed to derive divergence in trade-wind regions, Bony and Stevens, 2019). We highlighted that in any case long-range research aircrafts are needed for a strategy to derive IVT divergence in ARs.

  **Modifications (Line 54f): "[…], it is necessary to release the sondes at close spacing but over horizontally extended areas above the AR which can only be achieved by long-range research aircraft (Neiman et al., 2014)." We added more information of the width of ARs responsible for the flight duration and the importance of two cross-section legs. In turn, the detailed specifications were added in Sect 2.3 that specifies the reason for our flight strategy (Line 203ff).**

**Specific remarks in the introduction:**

Unusual language:
*L31:*
**Response:** we rephrased the sentence:

**Modifications in manuscript (Line, 37f): "Seager and Henderson (2013) point out that the divergence of IVT links the local temporal evolution of moisture amount to the efficiency of precipitation induction."**

*L51 (deteriorate … representation?)*:
**Response:** We rephrased the sentence to clarify that too few sondes affect the airborne representation of IVT negatively:
**Modification (Line 62f): "Deteriorations in the airborne representation of AR moisture transport variability can result from a limited number of sondes if the sounding spacing becomes too coarse to reflect the spatial variability of IVT."**

*L57 (monitors transport (…) seen from research aircraft?):*
**Response/Modifications:** As a consequence of our restructuring we deleted this sentence.

*In the introduction the reader is distracted by details about CARRA regional reanalyses. I suggest adding more details about CARRA, but in the methods section.*
**Response:** We delete the corresponding sentence and extended the specifications of CARRA in the respective data section:
**Modifications (Line 144ff): "Therefore, we further include the C3S Arctic Regional Reanalysis (CARRA). CARRA has a 2.5 km horizontal resolution over the entire domain and is accessible by Schyberg et al. (2021). Driven by lateral boundary conditions from ERA5, CARRA includes more observations and hourly forecasts by the HARMONIE-AROME model (Bengtsson et al., 2017). Køltzow et al. (2022) verified the improved representation of arctic surface-near meteorological conditions in CARRA, with decorrelation lengths of wind speed in better agreement to reference observations than ERA5.**

**Both reanalyses are provided on pressure levels by the Copernicus Climate Data Store (CDS). While ERA5 contains IVT as output, we calculate IVT in CARRA by the trapezoidal integral of moisture transport along the pressure levels (Tab. 1). In the following, we declare the high spatial resolution representation in CARRA as our idealised background reality of AR features."**

*Q2 addresses correlations of wind and moisture, which has not been motivated by the introduction. What is (un)known? For understanding Q3, the relevance of IVT divergence needs to be explained more carefully.*
**Response a):** The restructuring of the introduction considered these remarks in the paragraphs. Q2 now has an individual motivation like all other research questions. We motivated more thoroughly the correlations of wind and moisture and their potential interest for the measurement strategies and for different spatial patterns along the AR cross-section. This is done after highlighting the sensitivity to sounding frequency:
**Modifications (Line 72f): "When assessing spatial IVT variability in arctic ARs, it becomes crucial how moisture and wind fields coincide in the AR cross-section or whether they contribute independently to the IVT variability. For instance, in a polar AR case study, Terpstra et al. (2021) identified incoherent patterns of moisture and wind that form the moisture transport patterns but are less aligned than in mid-latitude ARs. The disentanglement of both quantities facilitates flight strategies in the observation of moisture transport divergence in arctic ARs. If the moisture transport variability (and divergence) were e.g. mainly controlled by the moisture field, more investment should be spent on airborne AR moisture representation by supplementary measurements. For this reason, it is important to determine whether moisture and wind are aligned in AR cross-sections and to ascertain: How correlated are moisture and winds in arctic ARs and do coherent patterns contribute significantly to IVT (Q2)?"**

**Response b):** For Q3, we describe the relevance of IVT divergence more thoroughly with respect to the steering of the local amount of moisture or to precipitation triggering.
**Modifications: see again Line 37f and Line 80ff**

**Section 2 and 3 (structure of the article):**

*The structure of the method sections is confusing and I suggest that sections 2 and 3 are merged. Section 2.1 (description of dropsonde data that is actually not used) can be deleted. Sections 2.2-2.4 can be summarized in a data and methods section. The TIVT definition (now in Sec. 4) should be moved to the Sec. 2-3.*
**Response/Modifications:** We agree that a merging improves the structure of the manuscript. **We deleted Section 2.1 which is indeed redundant and we come up with a modified structure of Section 2 (merging the original Sect. 2 and 3):**

**2 Airborne derivation of moisture transport divergence in arctic ARs**
**2.1 Reanalysis framework**
**2.2 Selection of Atmospheric River cases**
**2.3 Flight pattern and emulated observations**
**2.3.1 Zig-zag flight tracks observing AR corridors**
**2.3.2 Synthetic dropsondes**
**2.4 Sonde-based divergence derivation**
**2.5 Decomposition in AR frontal sectors**

Since this section becomes rather long, we provide some guidance for the reader at the beginning of Section 2, before then coming to the subsections **(Line 128):**
**"For this purpose, this section introduces the reanalysis framework we use to investigate a presented selection of arctic ARs. In addition, our airborne flight strategy to derive ∇IVT in arctic ARs is specified and how we emulate the synthetic sondes in the reanalyses. Lastly, we describe the sonde-based derivation of ∇IVT and how we categorise different sectors across the AR front to examine the divergence."**

In the following, the responses relevant to the modified Section 2 are specified. All section numbers refer to the modified outline. From Section 2.3 on, this means that the first number of each original major section decreases by one.

**Section 2.2**

*L105ff is unclear. How and why were the particular nine cases selected (unclear: L68 "predefined in ERA5 […] and why was only the Atlantic region considered? Please explain the purpose of placing the legs at the sea ice edge (L105ff is unclear). Why is only spring considered?*
**Response:** We carefully rewrote the first paragraphs of the original Sect 2.2. We moved the definition of IVT to Sect. 2 as it is a basic concept to define ARs. Before, it distracted in Sect. 2.2 when giving details about our individual AR selection criteria. To better explain our selection, we orientate to your question and recapture information given in the introduction as:
**Modifications 153ff:**
**"The transformation of arctic air masses moving over changing surface types (open ocean and sea-ice) along large-scale meridional circulations is part of current research and investigated by research aircraft over the Arctic ocean (Wendisch et al., 2021). For this reason, our study selects ARs causing air masses to overshoot the sea ice edge in**

the Arctic ocean. The principle identification of relevant arctic AR events is based on the IVT-based AR detection catalogue by Guan et al. (2022). Among these ARs, we focus on spring season, when maximum sea-ice extent in the Arctic ocean starts to break-up and reacts very prone to the intrusion of warm and moist air (Rostosky and Spreen, 2023). We restrict to conditions and AR events only from last decade, as the arctic climate has witnessed rapid and intense changes over the last decades (Wendisch et al., 2023). Our AR pathways originate from the North Atlantic and Barent Sea that Papritz et al. (2021) spot as dominant regions for arctic moisture intrusions. The selection constrains on ARs whose lateral width is purely situated over open-ocean or sea-ice. This ensures that we do not encounter orographic effects on IVT which are out of the scope of this study. Moreover, airborne observations and sonde releases over land are more complicated to be conducted. Given the criteria above, we selected ARs from nine spring days between 2011 and 2020."

*Can you explain the relation of ARs and warm air intrusions (L116)?*
**Response:** Arctic events designated as warm or moist air intrusions can often be classified as atmospheric river, as the intrusions are in conjunction with strong transport of moist airmasses. Due to their subpolar origin, the airmasses are preferably warm compared to predominant arctic conditions. Accordingly, we rephrased:
**Modifications (Line 175f): "AR8 originates from Siberia that, according to Komatsu et al. (2018), represents another significant roadway for arctic moisture intrusions favoring ARs."**

*Regarding CARRA, the authors should "[…] clarify the extent to which km-scale variability of moisture transport can be assessed. The grid spacing (how determined?) and effective resolution of such gridded data are certainly different."*
**Response:** The documentation of CARRA specifies an equidistant 2.5 km grid spacing over the entire model domain. Indeed, the grid spacing differs to the effective resolution of moisture transport. At least, Koltzow et al. (2022) illustrates the significant improvement of the decorrelation length for surface-near wind speed compared to ERA5. According to observations, the correlation decreases rapidly below 0.6 for distances longer than 50 km, roughly the ERA5 resolution in the Arctic. Nonetheless, we are aware that surface near wind is much more affected than upper levels, especially also when over complex terrain.
It is obvious that the resolution of CARRA is certainly different to the effective resolution. If we consider the results from Skamarowk et al. (2014, https://doi.org/10.1175/JAS-D-14-0114.1) designating an effective model resolution of approximately six times the grid spacing, we can still assume that moisture transport can be resolved in the order of several 20 kilometers in CARRA. Using ERA5, we would thus remain in the range of ~100 kilometers and would be in the order of magnitude of envisioned sonde resolution, and cannot make robust statements.
Nevertheless, further investigation of the added value of CARRA in representing ARs is definitely very interesting. For our study, however, we see a risk to overload its content and hope that our study instead motivates further research of ARs using the novel reanalysis CARRA. We come back to this in the conclusions.
In the manuscript, we added:
**Modifications (Line 140ff):**
**Still, Skamarock et al. (2014) emphasise that the effective resolution of processes in simulations is much greater than the model grid spacing. However, if our study aims to assess the sub-grid scale variability of moisture transport between sonde releases from reanalyses, the effective resolution should be of the order of ~ 10 km rather than of ~ 100 km. […] Koltzow et al. (2022) verified the improved representation of CARRA in arctic surface-near meteorological conditions by decorrelation lengths of wind speed in better agreement to reference observations than ERA5."**

*Why do you use pressure level data only and how might the rather low number of vertical levels influence the results (L374)? What is the separation of the levels in the lower troposphere?*

**Response:** The advantage and our reason for the usage the pressure-level CARRA data results from the consistency of pressure-levels allowing an easy calculation of the vertically integrated moisture transport (IVT) following Eq. 1. Moreover, when deriving the moisture transport divergence, the values at unique pressure values do not require any further interpolation. The separation of the levels can be depicted from the dots in Fig 9-10.

In an exemplary case (not shown), we have used the model-level data and basically find higher variability in the vertical profile for wind and moisture, but the effect on IVT and IVT divergence is minor, and in particular does not change our overall results significantly (not shown). Still, we recommend using the higher resolution model data in the conclusion for follow-up studies.
**Modifications: No changes made**

*L152f (why are radiometer/radar relevant?)*
**Response/Modifications:** We rephrased the sentence to:
**Line 211f: "We represent the aircraft location in a 1 Hz resolution, in line with the operational resolution of common airborne remote-sensing products (e.g. Mech et al., 2014; Konow et al., 2019) that can support dropsonde data."**

*Arctic ARs: I recommend adding a more detailed discussion about the determined characteristics of Arctic ARs (e.g., L315ff, results of Fig.1 and 10). This should involve a discussion of the communalities and differences of the presented nine cases. The large case to case variability should be better discussed.*
**Response:** We extended the description of our selected AR events. For that, we restructured the second paragraph of the modified Section 2.2, that introduces our AR cases and split it into two subparagraphs. First, we emphasize the inter-case variability with respect to communalities/differences in the synoptic situation. Second, we describe the actual AR pathways seen for our selection (aligned to the preprint version), as follows:
**Modifications (Line 165ff): "In the synoptic composition of a Greenland trough, low-pressure systems force large-scale meridional transport where ARs can evolve on the eastern cyclone flank and reach into the Arctic (Papritz et al., 2020). Similarly, blocking situations over Eurasia can favor meridional circulation. For our nine ARs (Fig. 1), we confirm a large case-to-case variability regarding the synoptic situation. While some ARs (AR2, AR3, AR4, AR9) have evolved along the eastern flank of large-scale troughs over Greenland, AR5 and AR6 are more steered by blocking high pressure over the Barents Sea. AR1 and AR7 are, in turn, reinforced by a mesoscale cyclone situated over the Fram Strait and reach very close into the cyclone center.**
**The synoptic compositions lead to AR dispersions over the North Atlantic and Arctic Ocean (Fig. 1), which correspond to the typical arctic moisture transport pathways identified in Papritz et al. (2021). Some ARs exhibit straight meridional moisture transport north of Iceland and approach or exceed Svalbard (AR1, AR2, AR3). AR4 and AR7 show more elongated filaments along the Norwegian coast but still reach far north. We consider eight independent AR events wherein AR5 is also considered for the consecutive day (AR6). At this stage, the centre of AR6 reaches close to the North Pole. AR8 originates from Siberia that represents another significant roadway for arctic moisture intrusions causing ARs (Komatsu et al., 2018). The last events in 2020 (AR8, AR9) are accompanied by a warm air intrusion period observed by the Multidisciplinary drifting Observatory for the Study of Arctic Climate (MOSAiC) expedition (Shupe et al., 2022), studied in Kirbus et al. (2023)."**

*Unclear statement", L119ff "picking (…) from catalogue"*
**Response:** we modified the beginning of the paragraph as follows:
**Modification (Line 179ff): "A caveat of our selection for making general statements about IVT variability in arctic ARs is the small sample size (nine AR cases). Therefore, we place our cases in the climatology of arctic ARs in spring. Using the entirety of spring ARs along the Atlantic pathway from the catalogue of Guan (2022), […]"**
**Furthermore, we put our following statements in a more logical order to clarify that our AR sample is representative for the rather strong AR cases.**

**2.3 Flight pattern and emulated observations**

*Removal of initial Section 2.1:*
**Response:** Due to the removal, slightly more description of flight performance and dropsonde characteristics to be emulated are given in Section 2.3. In the following our remarks regarding the flight strategy will be responded.

*I did not get how the flight tracks were defined. Isn't the zig-zag pattern only the consequence of sufficiently long cross-frontal legs at two latitudes that are required to capture the lateral heterogeneity and to be able to derive divergence?*
**Response:** For the divergence purposes, the cross-frontal legs are of relevance and actually sufficient. We decided to keep our term "zig-zag pattern" due to the fact that a single aircraft has to perform an internal flight leg in order to connect both cross-sections (a relevant time constraint also for our analysis). Nonetheless, we agree to put more emphasis on the cross-section legs themselves when introducing our flight pattern. Accordingly, we will reformulate:
**Modifications (Line 196ff): "Instead, the high lateral variability in AR moisture transport characteristics requires long flight legs across the AR front to better capture divergence heterogeneity. Two parallel cross-sections can be connected via an internal flight leg in a zig-zag flight pattern (Fig. 3). The zig-zag pattern observes AR corridors across its transport direction. The boundary cross-section legs perpendicular to the major flow quantify the corridor in- and outflow, i.e. in- and outgoing IVT over the entire lateral AR extension and enable simplified divergence calculations."** Information about diagonal legs and the moisture budget closure will still be given, but as a site note.

*Are all terminologies for the flight pattern (AR corridor, boundaries, boxes, sectors etc.) needed or would it be enough to describe two cross-sections at separate latitudes that are then classified in sectors? What defined the latitudinal spacing?*
**Response:** We reduced the terminologies accordingly and speak of cross-section legs rather than "zig-zag" whenever sufficient. The latitudinal spacing was adapted in a way that no landmasses reduce the cross-section length in the outgoing cross-section and that the northern leg is at least 100 km away from the ingoing cross-section.
**Modifications: no changes made**

*It is sometimes confusing what data is used. I actually thought that the flight duration was not considered for the "continuous" (L394, 426). "Continuous" was also used earlier (see e.g., Fig. 14 caption), however, I think it referred to the high-resolution cross section profiles. I suggest a clear structure and description.*
**Response:** Precise terms are very essential for the comprehension of our work. Therefore, in our restructuring of Section 2 and 3, we added the definition of the "continuous representation" at the end of Sect. 2.3.1 to make clear that this represents the best-possible sampling of moisture transport from the moving aircraft.
**Modifications (Line 216f): "This representation of meteorological values and AR characteristics will from now on be referred to as "continuous AR representation"."**

*L233ff should be moved to the method section.'*
**Response:** We moved the definition of the TIVT to the Section 2.4 dealing with the sonde-based divergence derivation that this section is now more compact.
**Modifications (230ff): TIVT-Definition added.**

*The advantages and limitations of the applied methods should be considered in view of other approaches.*
**Response:** We extended our description of the applied methods and contrast more the advantages and limitations of our cross-section pattern for divergence calculations. One obvious limitation are the open boundaries that the cross-sections leave.

**Modifications (Line 233ff): "Neglecting the moisture flux that exists apart from perpendicular to the flight track, we can approximate $\nabla$IVT in an AR corridor by the difference of out- minus ingoing TIVT of the cross-sections. However, this excludes any divergence of the flow perpendicular to the cross-section.**

**The Gaussian Theorem sets the moisture flux over the perimeter of a closed surface equal to its divergence. However, our flight pattern (Fig. 3) has open boundaries at the outer sides. Only if lateral flow can be neglected, we can obtain the divergence by subtracting the inflow in the entrance leg from the streamward outflow. Given this limitation, Lenschow et al. (2007) alternatively suggests the regression method. […] Adding up both gradients, we calculate the divergence. Bony and Stevens (2019) and George et al. (2021) proved the feasibility of this method by comparing its divergence values with the Gaussian-based line integral over flown circles."**

The major advantage we see in the ability of the cross-sections to derive the divergence in different sectors across the AR embedded front more or less simultaneously. Similar as in Norris et al. 2020, that investigated the airborne divergence pattern and subdivide the examined AR corridors, the sensitivity to different spatial scales can be assessed. (see answers in Sect. 2.5)

*Unclear L182:*
**Response:** We assume that the connection to the precedent sentence was unclear, as well as the vague statement of "two impacts". We rephrasedy the sentence to:
**Modifications (Line 248f): "The divergence of moisture transport can be split up into two components"**

*I do not understand the sector classification: Please specify the "requirements" in Cobb et al. (L198ff). In L194 the prefrontal, core and postfrontal are differentiated. Then you come up with a threshold definition for the AR edges. How does this all fit together and how are the sectors defined? Please move relevant information about Arctic ARs to the introduction.*
**Response:** We rephrased the description of our sector classification and explained the requirements of Cobb et al. in more detail, especially how we adapt those requirements to arctic conditions we found in Fig. 2.
With the term "AR edges", we mean the outer boundaries of the frontal sectors. At some lateral distance, the moisture transport (IVT) becomes too weak to be considered as atmospheric river. To facilitate the connection of our terminologies, we provide a Figure (listed as Fig. 4) illustrating the IVT-based frontal sector classification along AR cross-sections. Afterwards, we display how the sondes are located correspondingly in both cross-sections (then Fig. 5).
The requirements are described as follows:
**Modifications (Line 267ff): "Therefore, we conduct a similar sector-based decomposition of IVT divergence for our arctic AR events in CARRA. As in Guan et al. (2020) and Cobb et al. (2021a), our decomposition relies on the IVT characteristics along the cross-section, which we depict for an exemplary cross-section in Fig. 4. The central AR core represents the region of strongest IVT, which is more than 80% of maximum IVT ($IVT_{max}$). East of the core we situate the pre-frontal sector and west the post-frontal sector. Yet, their outer edges are less trivial as ARs basically have open outer boundaries. To account for case-specific relative values, we assign frontal edges where $IVT \leq 0.33\ IVT_{max}$. As a secondary absolute threshold, we declare a moisture transport with $IVT \leq 100\ kg\ m^{-1}s^{-1}$ as too weak to be assigned as AR-IVT. Both form the outer edges of the AR where the pre- and post-frontal sectors end (Fig. 4). Note that the latter threshold to define the AR edges follows the approach of Cobb et al. (2021a). However, we lower their mid-latitude based IVT threshold from 250 to 100 $kg\ m^{-1}s^{-1}$. By this, we refer to common polar moisture transport magnitudes that exceed the 95th percentile of climatology and are declared as ARs in the detection of Guan and Waliser (2015). Otherwise, we would either exclude most ARs north of 70◦N, or would shrink the AR cross-section so much that most transport is ignored, as statistics in Fig. 2 suggest.**

**Applying the frontal classification to both cross-sections, we obtain three sectors. For the cross-sections of the AR, we locate the sondes so that six synthetic sondes (three each from the in- and outflow cross-section) span each frontal sector and calculate its IVT divergence, respectively (Fig. 5). Inspecting the sonde positions in Fig. 5, we emphasise that our IVT -determined frontal AR sectors along the flight track tilt while the internal IVT has a straight northward orientation. This arises from the north-eastward displacement of the AR filament over the course of the 2.5 h synthetic flight section (Sect. 2.3). Accordingly, Sect. 5 examines the extent to which sonde-based IV T divergence is affected by flight duration, as opposed to actually looking at the AR in an instantaneous snapshot."**

[Figure]

*Figure 4 (in manuscript): Frontal sector decomposition for an exemplary IVT cross-section (AR1) using the criteria described in Sect. 2.5. The colored shadings and text boxes indicate each frontal sector. The grey shading on the left represents moisture transport (i.e. IVT) that is not considered as AR because it is too weak.*

**Section 2 specific comments:**

*L202: I cannot see the three dropsondes that calculate IVT.*
**Response:** We specified our misleading explanation:
**Modifications (Line 279ff): "Applying the frontal classification to both cross-sections, we obtain three sectors. For the cross-sections of the AR, we locate the sondes in a way that six synthetic sondes (three from the in- and outflow cross-section each) calculate the IVT divergence for each frontal sector respectively (Fig. 5)."**

*L204 (putative? inconsistency?)*
**Response:** we deleted both words as they do not provide any added values.

**Section 3: Moisture transport in Arctic AR cross-sections from soundings**

**General remarks:**

*The paper lacks a thorough discussion of the results, either within the result section or in a separated section at the end. [..] a few references within the result section, however, not detailed enough (see above) so that the added value of the paper becomes clear.*
**Response:** We agree that the discussion of results is worth improving. We decided to manifold and strengthen the discussions in the respective result sections rather than merging them in a separated "discussion" section. Therefore, we restructured concerning paragraphs in order to unravel the discussions of results more stringently. We strengthen the interpretation of our results in a more connected comparison to findings from literature (mostly based on midlatitude ARs). In the following, you find specific responses for the relevant sections, whereby many reviewer remarks are applicable for several paragraphs throughout our results sections.

*Add more references to figure panels within the text whenever appropriate.*
**Response:** Yes, this improves readability. We added them especially in our result sections.

**Sect. 3.1: Shape of IVT across arctic ARs**

*I recommend adding more detailed discussion about determined characteristics of Arctic ARs.*
**Response:** We take up this point for Sec 3.1, as the IVT shape of ARs in the Arctic is here first presented in more detail. We find that comparisons are helpful here to categorize our cases. Accordingly, we added some more discussions about the IVT strength for our AR cases with respect to mid-latitude cases (Cobb et al, 2021) and arctic cases studies (Viceto et al, 2022). This is done as:
**Modifications (Line 299ff): "Summarising all cross-sections of our ARs from Sect. 2.2, most arctic AR cross-sections show this typical bell-shaped IVT curve over widths of roughly 400 -800 km and indicate pronounced IVT maxima in the core of 300-600 kg m$^{-1}$ s$^{-1}$ (not shown). Only for the weak AR8, this structure is less pronounced. We find that the arctic AR are not substantially narrower than the AR widths of global climatology (Guan et al., 2015) or observed mid-latitudes events (Cobb et al., 2021). Flight planning should thus consider cross-section distances of around 500-1000 km, similar to mid-latitude ARs. However, this only applies if the legs are not restricted to regions with IVT>250 kg m$^{-1}$ s$^{-1}$, which is a widely used threshold for mid-latitude ARs (e.g. in Ralph et al., 2019). In contrast, the maximum IVT for the arctic events, is roughly half as high as the majority of mid-latitude AR from airborne studies in Cobb et al., 2021. Moreover, the IVT magnitudes strongly differ between our cases and synoptic conditions. The strongest ARs with maximum IVT (IVT$_{max}$) exceeding 500 kg m$^{-1}$ s$^{-1}$ are found for intense Greenland troughs, while the ARs are weaker along the Siberian pathway (see Fig. 1). If we compare our ARs with those of other arctic case studies (e.g. Viceto et al., 2022), we are looking at rather strong ARs."**
**Additional Response:** We specified the comparison between ERA5 and CARRA: Modifications (Line 310ff): **"Viceto et al. (2022) documented the improved representation of arctic AR characteristics in ERA5 against coarser reanalysis data. In our comparison of CARRA and ERA5, the location and horizontal pattern of the ARs agree quite well (not shown). For all cross-sections, we ascertain plausible IVT values from CARRA with respect to ERA5. In particular, we highlight that maximum (mean) values of IVT per cross-section increase by roughly 9 % (8 %) from ERA5 to CARRA on average. CARRA further increases the IVT variability by roughly 11 %. We attribute this to horizontal resolution being higher than in ERA5."**

*Should there be a strategy to place one dropsonde at a simulated maximum IVT (L223)?*
**Response:** The restructured discussion of results follows this suggestion as:
**Modifications (Line 315ff): "Using a set of six synthetic sondes, a gaussian fit of IVT can reproduce the bell-shaped AR-IVT cross-section (Fig.6). This gaussian fit is very sensitive to the actual positions of dropsonde releases. While the centered sonde in Fig. 6 is positioned close to IVT$_{max}$, a slight shift of this sounding, which easily occurs in real observations, can quickly lead to an underestimation of the moisture transport in the AR core. Flight planning should thus imply a sonde release in the vicinity of predicted IVT$_{max}$ and place additional sondes symmetrically around the core. While sonde positions in Fig. 6 are suitable to represent the cross-section IVT, other AR cross-sections evince more complexity in being accurately represented by this number of soundings. We need further inspections on how sounding intervals deteriorate the AR moisture transport observability."**

**Sec. 3.2 Sonde-based total cross-section moisture transport**

*I suggest adding a recommendation for the spatial separation (L252, L425) instead of a number per flight which depends on the flight performance. Figure 6: Change "seconds" to "minutes".*

**Response:** Indeed, a recommendation for the spatial separation is more universal with respect to flight performance. We changed the descriptions in this way and we also changed the axis of Fig. 6 to distances (km). Yet, since measurement operators frequently rely on specified time intervals when performing sonde releases manually, we add a light secondary axis referring to the spacing time. It is true that the duration depends on the flight performance, however, the values are valid for a common groundspeed at cruising level above 10 km.

**Modifications: Updated version of Figure 6 (now Fig. 7):**

[Figure]

*Fig. 6: The median lines for the grey boxplots are hard to see. I guess that these distributions are calculated from the boot-strapping method (add information to caption). How many cross-sections? Please add what percentiles the box and whiskers represent.*

**Response:** We added the information about the median lines, that are now illustrated bolder. We change seconds to minutes. The qualitive meaning of the colour-coding is specified in the caption. The statistics are based on the boot-strapping approach considering hundred positions of sondes per cross-section. In total, this includes 900 cross-sections. The boxes show the quartiles while the whiskers extend to show the rest of the distribution, except for outliers (depicted as markers).

**Modifications (Figure 7 caption): "Relative error in TIVT as a function of sounding spacing in km for all AR cross-section representations (grey) and those including highest (75th percentile) IVT maxima (coloured). Statistics rely on the boots-trapping approach containing of 100 cross-section sonde representations per AR. The boxes show the quartiles while whiskers show the rest of the distribution, except for outliers (markers). For an assumed aircraft speed of 250 m/s, equivalent release intervals are given on the top x-axis."**

*How sensitive are these results to the length of the flight pattern?*

**Response:** Indeed, the TIVT values are always dependent on the flight lengths. We also compared the TIVT values of the arctic ARs in more detail with mid-latitude observations, where we also point out the different AR widths between arctic and mid-latitude ARs, if one would restrict to the same thresholds defining the outer edges. Regardless the actual AR width on which we also align the flight length, we stick to our recommendation of seven sondes that should be envisioned to be released in order to derive IVT divergence in the three different frontal sectors (pre- and post-frontal, and the core). Since the stronger ARs (in terms of $IVT_{max}$) are also broader, Fig. 7 demonstrates that the minimum required sonde spacing is less sensitive to the actual AR width. However, we admit that robust conclusions in this sense should involve a much higher number of AR events. The discussion of the arctic TIVT values is done as follows:

**Modifications (Line 339ff): "The TIVT uncertainty in Fig. 7 increases less rapidly with larger sonde spacing than derived for mid-latitude AR cases (see Ralph et al., 2017 and Guan et al., 2018). Total moisture transport in the arctic cases is, in turn, much smaller than in mid-latitude cases. The arctic TIVT values are roughly half as high as the sonde-based mean TIVT of 5 $10^8$ kg s$^{-1}$ ascertained by Ralph et al. (2017) from 21 mid-latitude ARs. The ARs we consider have, in turn, roughly two third the width of the ARs in Ralph et al. (2017) and Guan et al. (2018). Here, we remind that our threshold to define the outer AR edges is much lower to encounter for arctic AR conditions. Applying the mid-latitude thresholds (given in Sect. 2.5), mean AR widths would be in range of a few hundred kilometer and T IV T values much lower than in mid-latitude ARs."**

**Sec 3.3: Variability of moisture and wind in arctic ARs**

**Response (according to a more detailed discussion) and vague relation to other studies (L266f):** In this section, we also put more emphasis on clearly disentangling results (e.g. Fig. 7) and discussions that we manifold. We compare vertical profiles in Fig. 7 with those of radiosondes in an arctic early summer AR period studied in Viceto et al. (2022) and synthesize communalities and differences to mid-latitude AR soundings in more detail:

**Modifications (Line 368ff): The vertical moisture characteristics in Fig. 8b) resemble soundings of arctic early summer ARs at Ny-Alesund demonstrated in Viceto et al. (2022) who showed q values up to 5 g kg$^{-1}$. However, the winds in our AR cross-sections (Fig. 8a) are roughly twice as strong as given in the case study of Viceto et al. (2022). Note that the ground-based station from which Viceto et al. (2022) depicted soundings were basically located at the outflow edge of the AR (with IVT ≤ 250 kgm$^{-1}$s$^{-1}$) on the Luv side. Winds from the east were orographically slowed down by the massif of Svalbard. In this sense, the marine arctic ARs we consider are undisturbed. This enables stronger winds whose magnitude is rather comparable to mid-latitude AR conditions. Ralph et al. (2004) and Cobb et al. (2021b) report on mean low-level wind speeds from 10–25 m s$^{-1}$ for a large set of ARs over North-East Pacific. The slight local wind maximum at 900 hPa (Fig. 8a) arises from the presence of strong wind corridors designated as LLJs that represent a common feature in mid-latitude ARs (Ralph et al., 2004; Demirdjian et al., 2020). Their polar existence is verified in the case study of Terpstra et al. (2021). We find a very dominant LLJ inside our most intense AR (AR3 in Fig. 8). Above the local wind maximum, the vertical profile of wind speed remains more homogeneous than in sub-tropic/mid-latitude cases where Ralph et al. (2005) and Cobb et al. (2021a) registered a stronger intensification with height.**

*L274f: How can you see this in Fig.8 (7 before)? Winds also strongly vary and the transport distribution (grey shading) resembles the wind distribution (red shading). The sentence in L275f contains redundant information.*

**Response:** We referred imprecisely to the strong AR case (AR3) represented by error bars and not by the shadings in Fig. 8. Now, we compare our descriptions for the entirety of ARs and AR3 more obviously and rephrased the discussion of moisture transport variability and the role of wind and moisture:

**Modifications (Line 380ff): "The cross-section variability of both moisture and winds strongly affects IVT variability. The shadings in Fig. 8 indicate that the standard deviation of moisture transport resembles the standard deviation of the winds for the lower levels up to 850 hPa, before moisture transport variability is apparently driven by the standard deviation of moisture in upper levels, although the wind standard deviation becomes highest above 500 hPa. For the most intense AR3, the LLJ exhibits high wind speeds above 30 m s$^{-1}$ that cause strong moisture transport whereas moisture is more or less average. While strong moisture transport in AR3 originates from overall strong winds, moisture varies strongly and seemingly dominates the moisture transport variability. Hence, we hypothesise that in strong arctic ARs with intense winds, moisture**

**variability primarily steers IVT variability and leads to the bell-shaped IVT cross-section pattern (Sect. 3.1)."**

*L285f: Do you have an explanation for the increased variability in the free troposphere? Fig. 10 shows that your cross-sections pick up dry post-frontal subsidence regions and also dry Arctic air eastward of the AR feature, which likely impacts this result. Or maybe this is what your last sentence wants to say? How much sense does it make to calculate horizontal means for such heterogeneous sections?*

**Response:** Indeed, the subsidence of dry airmasses is one of the major explanations for the increased variability in q. As referred to Fig. 10 by the reviewer, we take up with this question again in Sec 4.4. Here, Fig. 11 enables an illustrative explanation.

The reviewer's last remark opens a very crucial discussion that results from the question of what do we consider as the "AR itself"? Here, the scientific perception strongly differs between the horizontal and vertical perspective. In the large majority (and as this study does), moisture transport with respect to vertical integrated quantities (IVT and/or IWV) is designated as AR where a certain threshold is exceeded. Even if the thresholds may change between AR detection algorithm for various reasons, they mostly have in common that they project the AR from 3D (horizontal and vertical) to 2D (horizontal). No matter if different air masses are entrained at certain vertical levels, the domain is still considered as AR, as long as moisture / or moisture transport are sufficiently high in the vertical integral.

The vertical atmosphere may thus still hold two airmasses (dry post-frontal subsidence regions and a moist air mass smaller than the 'plume' in the AR core). How both airmasses interact across this interface is a question for itself. The degree of mixing can have strong impact on cloud and precipitation formation (beyond the scope of this study). However, due to this fact, we pretend that is worth to consider such edges where we find a coexistence of air masses. Also, in the perspective of practical flight planning, forecasts of IWV and IVT represent the quick identification of the AR object to locate the flight tracks in.
**Modifications: no changes made**

*Wouldn't it be more interesting to focus on the AR itself and check how much the fluctuations at small scales contribute to IVT?*

We highlight that we applied our cross-sections with respect to the AR edges from the AR catalogue (Guan, 2022) and can confirm that less than 5% of the flight tracks reach out of moisture transport that is declared as AR. This can also be seen in Fig. 11, where the AR edges (outer edges of the frontal sectors) are partially even more restrictively defined than in Guan (2022). Accordingly, we assure our results are representative for AR internal variability. By our frontal decomposition, we also put more focus of the central AR core.
**Modifications: no changes made**

**Sect. 3.4: Coherence of moisture and wind**

*L289-291: Better explain the meaning of "correlated" and "coherent".*
**Response**: we specified that we mean the correlation of both variables along the cross-section and here now speak of "connected pattern". We stated more explicitly that the non-coherent transport consists of the individual means of moisture and winds ($\bar{q} \cdot \bar{v}$).
**Modifications (Line 400ff):"For the moisture transport, it is not only important whether moisture and wind anomalies are high separately (Sect. 3.3), but also how correlated they evolve along the AR cross-sections and whether connected pattern contribute significantly to AR-IVT (Q2). If both pattern do, carefully collocated observations are essential to determine TIVT , otherwise independent estimates of mean moisture and wind are sufficient. The overall moisture transport […] is basically a combination of transport by the mean quantities q and v and their correlated cross-section variability, i.e spatial fluctuations q′ and v′[…]"**

*Like for Fig 10: This should involve a discussion of the communalities and differences of the presented nine cases. The large case to case variability should be better discussed.*
**Response:** We restructured the paragraph describing now Fig. 11 and insist in more detail on large case-to-case variability with more direct relations to the references given. We extended the discussion to mid-latitude ARs and focused on the discrepancies to AR schematics as in Ralph et al., 2017 when arctic wind and moisture pattern do not coincide. We highlighted on the differences in AR corridors that are closer located to the AR center (e.g. AR5) against those corridors situated in the outflow region (e.g. AR7/9). We referred to Terpstra et al. 2021 that detected missing coincidence in a polar AR outflow, but rather in the vertical axis than we do in the horizontal.

With the added comparison, we specify the role of dry subsiding airmasses that become more effective if there is upper-level advection from Greenland air masses. We note that our analysis from Fig. 11 can still be manifold in various perspectives and details. At a certain point, however, we need to refocus on our research question and investigate the correlation between moisture and wind for the given cases and how the patterns contribute to IVT and its variability.
**Modifications: Rewritten part (Lines 432-453) according to descriptions above**

**Section 3 specific comments:**

L225 (maintain?)
**Response/Modifications: Changed to "show"**

*L322f: If there is little information from small scale fluctuations, why should one care about supplementary q observations?*
**Response:** Yes, this is valid point. However, if one is still willing to improve the measurements, then one should focus on supplementary moisture observations.
**Modification (Line 458): "An improvement for observing the moisture transport variability should be built upon supplementary moisture measurements rather than those of the winds."**

L260 (why intuitive?),
**Response/Modification: rephrased to "simplified"**

L268 (behaves more homogeneous?),
**Response/Modification (Line 378):** changed to: "**remains more homogeneous**"

L277 (How?)
**Response/Modification (Line 388f):** rephrased to: **"The identification of the more variable quantity can improve measurement strategies. Specifically, moisture can be derived from supplementary remote sensing devices on long-range research aircraft."**

L278 ("long-term aircraft"?)
**Response/Modification:** typo, changed to long-range

L289-291 (e.g., "carefully correlated observations?", "cross-sectoral")
**Response/Modification (see above Line 400ff):** changed **to "collocated" and "cross-section variability"**

L322 ("narrowed moisture columns here form"?)
**Response/Modification (Line 455f):** Rephrased to: **"Instead, narrow and high-reaching moisture plumes in the core control the moisture transport variability."**

**Section 4: Moisture transport divergence from sondes**

**4.1 Sectoral in- and outgoing moisture transport**

*Fig. 11 "frontal specific AR sectors" is unclear. I […] wonder that the dotted lines at negative distances are warm pre-frontal areas? I do not understand the two sentences "Leg specific … (lines)" – please rephrase. What is "corridor IVT convergence"?*

**Response:** We rephrased the figure caption.

**Modifications (Figure 12 caption): "Figure 12. IVT along inflow (outflow) section in blue (orange) for all nine ARs (Fig. 1). Changes in line styles denote the frontal sector classifications (Sect. 2.5): Dotted lines represent cross-section periods attributed to pre-frontal sectors, while dashed lines refer to post-frontal sectors. The legend depicts TIVT values for the in- and outflow cross-section parts within the AR. They include IVT purely internal of determined AR borders (Sect. 2.5). Arrows indicate the TIVT difference between in- and outflow leg scaled in length and width. The differences can be viewed as simple estimates of IVT divergence in between both legs, according to Sect. 2.4. Upward (downward) arrow scales represent estimated convergence (divergence) magnitudes. Note the x-axis orientation is from east (left) to west (right). "**

*It did not become clear to me how the cross-sectional IVT gradients (Fig. 11) are connected to the dynamical situation and the results are contrasting the impression that I got from Fig. 1.*

**Response:** We suppose that the axis orientation confuses the reader. Negative distance values refer to the eastern end, while positive values refer to the western end of our cross-section legs. Thus, the cross-section pattern should be mirrored when comparing with Figure 1. We added the clarifying explanation in the Figure caption (see above).

Then we cannot detect such inconsistencies to Figure 1. For example, the steep decline in the post-frontal sector of AR2 (Fig. 11b) is well in conjunction with IVT pattern shown in Fig. 1. Similarly, the gradients of in- and outflow legs for AR2 & AR7 are consistent with the outflow IVT pattern in Fig. 1. Later in the section, we highlighted that a comparison for estimating IVT divergence can be misleading in some cases. Note that IVT cross-sections are (although continuous) based on flight duration and not on an instantaneous snapshot as from Fig 1.

**Modification (Line 489ff): Nonetheless, although the IVT pattern of AR5 and AR6 (Fig. 1) allow slight divergence in the pre-frontal sector, we emphasize that a TIVT-based interpretation of predominant moisture transport divergence underlies strong idealisation. It considers neither moisture flow being non-perpendicular to the flight, nor it does separate contributions of moisture advection and mass convergence. Therefore, we insist on the regression approach to diagnose moisture transport divergence in each frontal sector of the arctic ARs.**

*The TIVT discussion (Fig. 11) could focus more on the AR area: Why is the divergence dominating in what you call warm sector – isn't that surprising?*

**Response:** We restructured the section: The first paragraph covers TIVT as a whole, the second paragraph the cross-section differences, the last paragraph investigates in more detail the TIVT gradients for the frontal sectors. We will highlight surprising findings more, but point out the limits in estimating divergence purely from IVT magnitudes.

**Modifications (L477ff): "Figure 12 further separates the AR cross-sections in the three sectors (pre-frontal, core, post-frontal). Although the AR cores are roughly 200–300 km narrow (slim lines in Fig. 12), they provide more than half of the entire AR-TIVT. This contribution of the AR core agrees with findings from Cobb et al. (2021a) in mid-latitude ARs. Except for AR2 and AR7, weaker slopes of IVT are generally in the cold sector as opposed to the warm sector. In turn, the steep post-frontal decline of *IVT* in AR2 and AR7 results from calm air masses on the backside of the AR (see also Fig 11).**

**Comparing both legs (Fig. 12), cross-section TIVT tends to decrease downstream in some arctic ARs. Yet, we likewise identify cases with weak stream-ward tendencies in total moisture transport or with slight increases. Moreover, the downstream difference**

**of TIVT is distributed unevenly over the cross-section IVT. It is mainly within the AR core where IVT decreases towards the outflow leg (e.g. AR3, AR9), thus suggesting internal convergence. However, counteracting behavior in the frontal sectors partially compensates the core and stream-ward decrease of IVT. Like in AR6, the increase of warm sector IVT towards the outflow conveys a seeming divergence in the warm sector. This is in contrast to the findings in Guan et al. (2020), where the pre-frontal sector is denoted as a region of moisture transport convergence. Nonetheless, although the spatial IVT pattern of AR5 and AR6 (Fig. 1) allow slight divergence in the pre-frontal sector, […]”**

*It is sometimes confusing what data is used. I actually thought that the flight duration was not considered for the "continuous" (L394, 426). "Continuous" was also used earlier (see e.g., Fig. 14 caption), however, I think it referred to the high-resolution cross section profiles. I suggest a clear structure and description.*

**Response:** We apologize for the missing clarity of the term "continuous". We added a clear statement in the method section (2.3.1) that underlines our definition of "continuous" cross-sections. The manuscript refers to the Sect. when speaking of "continuous" in the remainder of the study.

**Section 4.2: Sonde-based divergence and its representativeness**

*missing detail about the related work: L352f, L355f, L358f:*

**Response:** We specified our reference to related work and stated more precisely how we build on the precedent studies of Guan et al. (2020) and Norris et al. (2020).

**Modifications (Line 495ff): "This section specifies the IVT divergence *($\nabla$IVT)* in arctic ARs. Using the regression-based $\nabla$IVT (Sect. 2.4), moisture transport divergence is examined for the frontal sectors (Sect. 2.5) and for the decomposed terms, namely moisture advection *ADV* and mass convergence *CONV* (Eq. 5). Again, the results from the continuous cross-section flight legs (Sect. 2.3.1) represent our idealized reference. We compare them to regression-based results referring to seven synthetic sondes per cross-section (as in Fig. 5). This comparison assesses uncertainties of sonde-based $\nabla$IVT, representative for arctic ARs. In doing so, we build on Norris et al. (2020) who pioneered the airborne derivation of all moisture budget components, including moisture transport divergence, by sampling a mid-latitude AR event."**

**Modifications (Line 518ff): "The fact that the moisture transport divergence components differ across the frontal axis is in line with mid-latitude AR based statistics of Guan et al. (2020). In detail, the characteristics in AR3 described above differ quietly to the AR case observed by Norris et al. (2020). In their airborne study of an mid-latitude AR, they found moisture transport convergence to be strongest close the AR core and rather opposite signs for the pre- and post-frontal regions than us. Especially the lack of pre-frontal moisture advection in AR3, which Guan et al. (2020) actually robustly found in mid-latitude AR statistics, is worth-mentioning. In contrast to both Norris et al. (2020) and Guan et al. (2020), we do not identify a dominance of dynamical convergence over advection. The magnitudes of moisture transport divergence in AR3 are also much lower. Nonetheless, we remind that Norris et al. (2020) and Guan et al. (2020) consider even more intense mid-latitude AR near its centre. While AR3 is exceptionally strong for arctic conditions (Fig. 2), it is rather moderate for mid-latitude scales (Ralph et al., 2019)."**

**Section 4 specific minor comments:**

*L333ff (What is the "simplified understanding of divergence"?, "benchmarks…"?)*

**Response:** We merged both expressions in a connected sentence.

**Modifications (Line 470ff):** "The comparison of *TIVT* in both legs reveals first simplified estimates of the prevailing divergence. Idealising that no entrainment into the AR corridor (Sect. 3) takes place, Figure 12 contrasts TIVT of the in- and outflow cross-section to estimate whether convergence or divergence of moisture transport exists inside the AR corridor."

*L358 ("behave differently"):*
**Response/Modification (Line 502f):** formulation rephrased: "**ADV and CONV exhibit different vertical profiles throughout the frontal cross-section**"

*L360 ("lower atmosphere")*
**Response:** deleted

*L364f ("integrate along the vertical axis")*
**Response:** The respective sentence was removed as it is invalid for the updated results
*L373ff unclear*
**Response/Modifications (Line 531ff): We reformulated the sentences to: "When we place our sonde results in the context of the airborne study by Norris et al. (2020) using real dropsondes, we recognize the strength of true sondes with a high vertical resolution. They provide much greater vertical variability. Thus, it is likely that the quite low divergence displayed in Fig. 11 does not only result from less divergence prevailing in Arctic ARs compared to mid-latitude ARs, but may also be a consequence from the coarser vertical grid that average out larger values."**

*L392 ("our arctic AR composition"?)*
**Response:** changed to: "for our sample of arctic ARs"

**Section 5: Deterioration by non-instantaneous sounding**

**Section 5: Specific minor comments**

*L403f (unclear)*
**Response:** reformulated:
**Modifications (Line 582ff): "This section examines the extent to which the temporal AR evolution during flight affects the sonde-based representation of IVT divergence. Up to 3 hours are needed to fly over AR corridors and consecutively observe the in- and outflow (Sect. 2.3.1). Meanwhile, temporal AR evolution can distort the airborne (non-instantaneous) representation of IVT divergence in the AR."**

**Section 6: Summary and Conclusions**

*[…] should synopsize and synthesize the key results and identify the contribution to research on Arctic ARs. I think it will strongly profit from an improved discussion of the results.*
**Response:** In the following, we improved our conclusions in terms of clarity in structure and the discussions of our key results.
*So far, the first paragraph is a repetition of what was done. The second paragraph claims that higher resolution reanalyzes increase our understanding of arctic moisture transformation and precipitation efficiency, which I don't see is addressed.*
**Response:** We split up the first paragraph. In accordance with the remarks of reviewer 2, who suggests to put more emphasis on the general arctic AR conditions rather than airborne

perspective, we restructured our conclusions. First, we approach the concrete perspective of our study (the observability of arctic AR IVT divergence by dropsondes). In our opinion, it is helpful to briefly repeat how the synthetic soundings were established to answer our research questions. In the second paragraph, we synopsize the basic AR-IVT characteristics in the Arctic we found before specifying the airborne perspective. By that, we got rid of the initial second paragraph which lacked clear structure and key messages. Instead, we summarised our considered IVT magnitudes in the Arctic and how they differ to mid-latitude ARs

**Modifications (Line 637ff):**

**"This assessment study investigated the characteristics of the moisture transport divergence in arctic Atmospheric Rivers (ARs). We analysed the ARs from an airborne perspective to assess the dropsonde-based observability of moisture transport divergence of arctic AR. We characterised airborne uncertainties in sonde-based representation of the AR moisture transport divergence inside arctic ARs, focusing on two sonde-based limitations: subsampling by too large sounding spacing and the non-instationarity of the AR over the flight duration. For this, we followed a synthetic approach using reanalysis data a virtual truth. CARRA reanalysis data were interpolated on synthetic flight pattern that consist of two cross-sections covering frontal sectors over the entire AR transect. Single vertical profiles emulate dropsondes.**

**We considered nine arctic AR events over the Atlantic pathway to the Arctic ocean in the vicinity of the sea-ice edge from last decade. The values of Integrated Water Vapour Transport (IVT) in the AR cores range from 300-600 kg m$^{-1}$ s$^{-1}$, although the ARs are primarily examined north of their center. We thus classify these AR as rather strong for arctic conditions. Still, the bell shape of IVT across the AR varies strongly in between the AR cases. The considered cases cover a large variability and consist of various synoptic pattern (extended troughs, blocking situations, single cyclones) in which the AR are embedded. This study delivers benchmarks of uncertainties in the airborne representation of sonde-based AR moisture transport divergence. We conclude the four pursued questions (Q1-Q4) as:"**

The reason for summarising our synthetic framework is to remind the perspective that our feasibility study has chosen. We are confident that such a repetition of the synthetic approach facilitates the readability in later discussions, as well as the interpretation of our conclusions. Nevertheless, we erased too detailed information (e.g. the regression method used).

*The authors should try to better synthesize the central message of their results to each of the RQs in view of the gained knowledge. I suggest a separate discussion of how the obtained results may affect future flight planning and the deployment of dropsondes (Q1, Q4).*

**Response a):** We updated the central messages to our research questions and conducted a stronger connection to individual flight planning which is achieved by an improved structure. We extended the specifications in our recommendations for future dedicated flight planning.

**Modifications (Line 652ff): "For the sonde-based determination of Total Integrated Water Vapour Transport (TIVT) in arctic AR cross-sections, sonde spacings below 100 km robustly keeps TIVT errors below 10 % (Fig. 6). In strong ARs with IVT exceeding 500 kg m$^{-1}$s$^{-1}$, too coarse IVT representation at the AR core leads to TIVT underestimation. Gaussian fits help to reproduce the cross-section IVT shape but are sensitive to how sondes estimate maximum IVT and its location. Thus, precedent flight planning should aim for a sonde release at forecasted IVT maximum and place additional sondes symmetrically around. For arctic AR widths of 400-800 km, we suggest a minimum of seven soundings per cross-section (roughly 60 to 120 km spacing) to derive TIVT in both cross-section legs. The maximum IVT is more correlated to IVT variability than the AR width is. The planning of sonde releases should thus rely on the steepness of IVT along the cross-section. We highlight that the differences of TIVT between the in- and outflow cross-sections are in a range of 2-15% (Fig. 12). If we want to reliably estimate moisture transport divergence based on TIVT from both cross-sections, the sonde-based uncertainty of TIVT for a single leg must be considerably lower."**

**Response b):** We already gave some suggestions for future flight pattern in the preprint's conclusion, but in a poor structure. Now, we improved the structure of our concluding implications on flight planning emerging out of the conclusions from Q1-Q4.

**Modifications (Line 698ff):" We confirm the observability of moisture transport divergence in arctic AR corridors by releasing sondes in such dedicated flight patterns. A maximum sonde spacing of 100 km within the AR cross-section can in principle characterise the divergence between both cross-sections at the given uncertainties of ≤ 10 %. For the flight durations, we obtain the entire moisture transport divergence specified for the frontal sectors with an uncertainty in the range of 25-50 %. We deduce that sonde undersampling matters and recommend a sequence of at least seven sondes per section given the widths of arctic ARs. However, notwithstanding that we could release a higher number of sondes, it is the temporal AR evolution over flight duration that leads to higher deviations in the divergence components rather than sonde undersampling. The dedicated planning of such sonde-based purposes should not only include the positioning of the sondes, but also the minimisation of the flight duration. The placement of the cross-section legs and their spacing should carefully consider the AR displacement during flight. Shorter meridional distances between the cross-sections not only reduce the flight duration, but also the area enclosed by the sondes. Given the frontal-sector widths of the arctic ARs, both cross-sections should be no more than 200 km apart. For several of our cases, the meridional separation is higher and we have to expect considerable sub-grid scale variability. Collocated flights by two aircraft, with both cross-sections being not far apart and sampled simultaneously, is the optimal and still feasible strategy. When faced with a limited amount of dropsondes, supplementary measurements of moisture should be prioritized, as moisture represents the more variable quantity and moisture advection mostly dominates the moisture transport divergence in the AR corridors."**

**Response c):** After these implications we added a description of the limitations of our study that covers several points mentioned in the reviewer's remarks (vertical resolution of CARRA used, limits of regression approach. We used the constrains to provide specific suggestions for arctic AR follow-up studies using CARRA.

**Modifications (Line 713ff): "Additional limitations of our study need to be discussed. As our results are mainly based on corridors in the AR exit region, we strongly recommend extending our uncertainty assessment to other AR regions and expect the role of winds and mass convergence to increase in strong ARs. This becomes an even more important issue with respect to the tendency of arctic ARs to shift more northward and intensify under climate change (O'Brien et al., 2022). Furthermore, as we include a large variability of synoptic AR patterns but a small sample, we propose statistics with a larger number of AR events. The statistics can improve our understanding of the moisture transport divergence pattern in arctic ARs and attribute it to the dynamic and thermodynamic atmospheric conditions. Here, CARRA represents a very suitable reanalysis framework for this purpose in follow-up studies. Again, we encourage the use of the higher vertical resolution of the model levels rather than our chosen pressure levels, although sufficient for initial estimates. For real sondes, we emphasise the added value of their high vertical resolution. Sondes provide more accurate information on the vertical composition of ADV and CONV. The sonde-based approach is limited to regression-based divergence where we consider only rather large areas and open meridional boundaries. Even with continuous lateral sampling, the meridional gradients are only coarsely sampled.**
**Therefore, a follow-up study should investigate how the arctic AR moisture transport divergence acts internally of the flight corridor at grid-cell scales. This will allow two additional research topics to be addressed: First, the internal variability between both**

**cross-sections can be derived more precisely to improve the flight pattern, second the actual scales at which the moisture transport divergence varies significantly can be evaluated. This may also increase the divergence magnitudes, similarly to Norris et al. (2020) who found larger values of the divergence components. They considered smaller airborne AR corridors than the ERA-Interim pixels referred to in Guan et al. (2020)."**

**Response d):** Finally, we summarize the necessity of assessing the sonde-based observability and of deriving uncertainties for model-observation inter-comparison.
**Modifications (Line 731): Despite the aforementioned limitations, the orders of magnitudes for IVT variability and divergence that we provide are representative for arctic ARs and quantify benchmarks in the sonde-based derivation. Consistently mimicking the soundings is a fundamental step towards the understanding of the uncertainties when such airborne tactics are actually carried out. The benchmarks are not only useful for improving flight strategies, but also indicate deviations in corresponding model-observation comparisons. Only by illuminating the constraints on the AR representation from both models and observations, we establish a framework from which airborne observations can support modellers in terms of the resolution and complexity required for the parameterisation of moisture transformation processes caused by IV T divergence in arctic ARs.**

*L485 (unclear)*
**Response:** changed.
**Modifications (Line 673f):** "By contrasting in- and outflow TIVT through the AR transects, we expect an overall divergence in moisture transport."

**II) Response and Changes to the Comments from Anonymous Referee 2 (AC2)**

We thank the ACP associating editor, Geraint Vaughan, as well as the Anonymous Referee #2, for this inspiring review. Please find below our responses (in standard font) to the remarks from the Anonymous Referee #2 (in *italics*). The changes/modifications in the manuscript are specified below (**bold**). We structured this response in such a way that comments on the most important text blocks for improvement (e.g. motivation) are bundled and distinguishable from each other. In this response, we occasionally refer to our Author's responses for Referee 1 (from now on denoted as AC1), because several answers in AC1 consider the remarks given from Referee 2 and we intend to avoid repetition.

*This paper provides a contribution to advancing our understanding of arctic atmospheric rivers by presenting an analysis of them using different reanalysis products, and suggesting ideal targeting strategies for the purpose of understanding and closing the moisture budget.*
**Response:** We want to thank you for the detailed and inspiring feedback. Given your remarks, we realize that the perspective of our study had to be carved out more clearly. We are confident that the specification significantly improves the readability and clarity of the manuscript. Accordingly, we focused on improving the readability by a more elaborated structure which provides more precise motivation for sampling of ARs from airborne dropsondes.

**Major**

*In general, I think it is important for this paper to provide some additional context and motivation for the exercise of synthetic sampling.*

**Response:** We agree that the preprint lacks a well-elaborated motivation for our sonde sampling approach with respect to arctic ARs. Therefore, we carefully rewrote the introduction which, based on the reviewer remarks, we identified as one of the major weaknesses of our manuscript.

In detail, we specified our changes in structuring the motivation of AC1. At this point, we summarize some of the changes that we declare as most relevant according to Reviewer #2: The first paragraph now states more explicitly the presence and impact of Atmospheric Rivers (ARs) in the Arctic. We now refer to a broader collection of studies with respect to the Arctic and concretise their findings relevant for our motivation, rather than only list relevant literature (as done before). The second paragraph characterizes the moisture transport (IVT) and its divergence in ARs, how this becomes relevant for the transformation of moist air masses. Finally, it disentangles the issue and research gap, which is the lack of quantitative estimates for IVT divergence within Arctic ARs.

**Modifications: see AC1 in I) above**

"*Is there a possibility for in situ sampling of arctic ARs? Is the paper calling for this capability as a requirement for us to meaningfully further our understanding in this region?*"

**Response a):** The second paragraph is a logical transition to the description of the required measurement strategies to derive IVT and highlighting the need of in-situ sampling to obtain the moisture transport throughout the troposphere. In the third paragraph, we now address this point in more detail: We added that the observational radiosonde network in the Arctic (Dufour et al, 2016) allows the derivation of IVT divergence into the Arctic, but argue that this network is too coarse to resolve IVT variability and divergence within single AR events. This motivates the use of dropsondes from research aircraft specified in the following. This argumentation better highlights the ability of in-situ sampling which serves as a prerequisite to be able to meaningfully expand our knowledge of moisture transport in arctic ARs.

**Modifications (Line 49ff): "Radiosondes allow detailed insights of moisture transport profiles of arctic ARs at individual locations (e.g. Viceto et al., 2022), but their observation network in the Arctic is too sparse to obtain the divergence in single ARs (Dufour et al., 2016). Similarly, dropsondes released from research aircraft can also provide vertical profiles of relative humidity and wind speed with an accuracy of 1 % and 0.1 m/s, respectively (e.g. George et al., 2021; Konow et al., 2021)."**

**Response b):** Both referee reviews demonstrate the need of an improved motivation for our choice of a synthetic sampling approach. Therefore, the research focus and overarching motivation of our feasibility study are introduced earlier (4th paragraph). We mention that there currently exist arctic airborne flight campaigns using a long-range research aircraft proposed by Wendisch et al. (2021). This fact motivates a pre-assessment of the sonde-based observability of moisture transport divergence in arctic ARs. Not only, this will improve the interpretation of sonde-based observations gained in the HALO-(AC)3 campaign (Wendisch et al., 2021; Walbröl et al., 2023) which are currently under processing, our feasibility study aims at facilitating future flight mission planning that has a similar special focus on high-latitude ARs, e.g. like the NAWDIC campaign:

https://internal.wavestoweather.de/campaign/projects/nawdic/wiki)

**Modifications: no change made in the manuscript as too detailed and distracting**

**Response c):** The changed order in which we introduce our research questions sequentially, rather than at the end of the introduction as in the preprint, intends to strengthen our argumentation. In doing so, we expect that this leads to more clarity why we have chosen a synthetic approach in investigating the arctic ARs.

*How would this papers' findings be different if the synthetic sampling wasn't a part of it? Does this framing potentially distract from the findings regarding the structure of arctic ARs?*

**Response:** We fully agree that a study dealing with the structure of arctic ARs using reanalyses or models will represent a very fruitful scientific contribution. Nonetheless, we admit that we, the authors, are mostly situated in the observational scientific community and are confronted with the necessity of airborne data in arctic ARs. Since we do expect several studies to emerge from the previous HALO-(AC)³ flight campaign and upcoming campaigns regarding arctic/ high-latitude ARs, we see a benefit in our approach for future studies. For instance, in

the HALO-(AC)³ special issue of ACP, we envision a contribution of the novel airborne derivation of all moisture budget components (including IVT divergence) in an arctic AR that was observed during HALO-(AC)³ (Walbröl et al., 2023). For this, our feasibility study quantifies the magnitude of airborne misrepresentation in sonde-based moisture transport divergence. Correspondingly, our title immediately makes clear that our focus is in the observability.

Nonetheless, we take your suggestion into account in the manuscript. Our conclusions now emphasise the ability of investigating arctic ARs in a more general perspective using CARRA. As listed in the AC1, we promote follow-up studies that attribute dynamic and thermodynamic conditions to the AR characteristics.

**Modifications (Line 716ff) : "[…] as we include a large variability of synoptic AR patterns but a small sample, we propose statistics with a larger number of AR events. The statistics can improve our understanding of the moisture transport divergence pattern in arctic ARs and attribute it to the dynamic and thermodynamic atmospheric conditions. Here, CARRA represents a very suitable reanalysis framework for this purpose in follow-up studies. Again, we encourage the use of the higher vertical resolution of the model levels rather than our chosen pressure levels, although sufficient for initial estimates. "**

*I suggest the authors consider strengthening their case for structuring the paper in this way and referring to more papers studying arctic ARs and their structure in addition to observational studies covering the midlatitudes if they would like to keep this framing.*

**Response a):** We restructured our introduction in this way (see AC1 above). We included more literature findings from arctic ARs, not only in the introduction but also when comparing our results to polar AR characteristics investigated in other studies, such as Terpstra et al. (2021); Viceto et al., (2022); Lauer et al (2023). Please find detailed manuscript modifications in the AC1 above (especially for Section 3 and 4). In compliance with the remarks from Referee 1, we elaborated on the results concerning the general structure of arctic ARs in more detail in the respective sections, before moving on to the sonde-based representation.

This also applies for the presentation of our arctic AR cases with manifolded discussion of the synoptic conditions causing AR outbreak to the Arctic (see AC1, Sect. 2). In Sect 3.1, we compared the arctic AR-IVT shapes in more detail with those from mid-latitudes

**Modifications (Line 301ff): "We find that the arctic ARs are not substantially narrower than the AR widths of global climatology (Guan and Waliser, 2015) and observed mid-latitudes events (Cobb et al., 2021a). Flight planning should therefore consider cross-section distances of about 500-1000 km, similar to mid-latitude ARs. However, this only applies if the legs are not restricted to regions with IVT > 250 kg m$^{-1}$s$^{-1}$, which is a widely used threshold for mid-latitude ARs (e. g. Ralph et al., 2019). In contrast, the maximum IVT for the arctic events is roughly half as high as the majority of mid-latitude ARs from airborne studies in Cobb et al. (2021a). Moreover, the IVT magnitudes strongly differ between our cases and synoptic conditions. The strongest ARs, with IVT$_{max}$ exceeding 500 kg m$^{-1}$s$^{-1}$ are found for intense Greenland troughs, while the ARs are weaker along the Siberian pathway (see also Fig. 1). If we compare our ARs with those of other arctic case studies (e. g. Viceto et al., 2022), we are looking at rather strong ARs."**

Response b): We use this knowledge as a prerequisite for the following examination of the sonde-based AR-IVT representation. Not only for general AR characteristics, but also for the sonde-based representation, we enlarged and concretized the comparison to mid-latitude based studies. You can find concrete examples given in the AC1 above (e.g. referring to Sect 2.2, 3.1, 3.4, 4.1, 4.2). In particular, the modifications include more analysis of the phenomenology of arctic ARs in contrast to mid-latitude ARs. By this, we consider the knowledge based on mid-latitude ARs. Nonetheless, we keep our original framing/scientific perspective in principle. Still, we now see a progress of the manuscript in providing more details of general arctic AR characteristics. These details are used to additionally improve our argumentation and discussion in the sonde-based assessment. The first paragraph of the conclusions come back to this point and summarize the arctic AR characteristics.

**Modifications (Line 638ff): "This study investigated the characteristics of the moisture transport divergence in arctic Atmospheric Rivers (ARs). We elaborated on conditions**

**of the moisture transport, i.e. the Integrated Water Vapour Transport (IVT), where the high-resolution CARRA reanalyses formed our representation of IVT variability. We considered nine arctic AR events over the Atlantic pathway to the Arctic ocean in the vicinity of the sea-ice edge from last decade. The IVT values in the AR cores range from 300-600 kg m$^{-1}$ s$^{-1}$, although the ARs are primarily examined north of their center. We thus classify these AR as rather strong for arctic conditions. While specific humidity mostly remains below 6 g kg$^{-1}$, we cover AR events indicating the presence of a LLJ with wind speeds higher than 30 m s$^{-1}$. Still, the bell shape of IVT across the AR strongly varies in between the AR cases. The considered cases cover a large variability and consist of various synoptic pattern (extended troughs, blocking situations, single cyclones) in which the AR are embedded.**
**Given this variety, we analysed the ARs from an airborne perspective […]”**

*I suggest considering a reframing where the authors discuss what can be learned about arctic AR structure from appropriate reanalyses at different resolutions, and then recommend sampling strategies to verify/supplement this knowledge.*

**Response:** For our purposes, we see a certain risk in changing the whole structure in this direction, because then the main objective of this study (which is the assessment of sonde-based IVT observability and uncertainties in the sonde-based representation) would be underrepresented or the paper could become too long and overloaded. Instead, we sketched the impact of the reanalysis resolution on the arctic AR structure in more detail in Sect. 3.1 and referred to the current study of Viceto et al. (2022) in which they conducted a reanalysis comparison in a case study of arctic ARs. We included two parts:

**Modifications (Line 295ff): “[…] we recognise the bell-shaped IVT from both, CARRA and forcing ERA5. Within the cross-section centre which we declare as the AR core in Sect. 2.5, CARRA, however, shows stronger moisture transport with a more pronounced IV T maximum > 500 kg m$^{-1}$s$^{-1}$. Moreover, CARRA resolves more small-scale structures of the AR moisture transport. In particular, CARRA increases the cross-section variability for this case.”**

**Modifications Line (309ff): “[…] Viceto et al. (2022) documented the improved representation of arctic AR characteristics in ERA5 against coarser reanalysis data. In our comparison of CARRA with ERA5, the location and horizontal pattern of the ARs agree quite well (not shown). For all cross-sections, we ascertain plausible IVT values from CARRA with respect to ERA5. In particular, we highlight that maximum (mean) values of IV T per cross-section increase by roughly 9 % (8 %) from ERA5 to CARRA on average. CARRA further increases the IVT variability by roughly 11 %. We attribute this to the higher horizontal resolution than in ERA5.”**

*Do your results regarding non-instantaneous sampling change if you take into account the observations in time and space where and when they occur, as is possible in many assimilation systems now?*

**Response:** We remind that we interpolate the reanalysis data in time and space onto the flight track and compare this to the reanalysis output at centered hour (instantaneous snapshot).
Current methods to derive divergence from airborne soundings require atmospheric stationarity (e.g. Bony and Stevens, 2019), which can only be idealized in observations. To circumvent this issue, Norris et al. (2020) conducted a time-to-space adjustment of their sonde profiles. Our study aims to address the impact of instationarity for our divergence calculations by research question Q4. Here, we clearly see the limitations in the sonde-based derivation of moisture transport divergence from our flight pattern. Still, your recommendation is very useful for future steps in improving the regression methods to a multivariate regression involving the temporal component. For our purposes in which we refer to the state of the art in the calculation assumptions for the observations, assimilation methods remain out of scope of this study.

From the recent flight campaign HALO-(AC)3, we know that the dropsonde data has been integrated into the Global Telecommunications System and used for assimilation in ERA5. The upcoming investigation of the HALO-(AC)³ observations in arctic ARs should certainly take this into account. So far, we can only speculate about the outcome but see a very good agreement between ERA5 and the dropsondes due to the assimilation.
**Modifications: no changes made**

*I very much like how the authors identify key questions and then revisit them in the summary with their answers to synthesize the paper for the readers.*
**Response:** We are glad that the key questions are a suitable guideline throughout our manuscript. Under consideration of the remarks from Referee 1, the motivation improved unravelling and identifying our key questions. The synthesis of our answers now has a more precise focus on implications for future airborne measurement strategies.
**Modifications: see Responses AC1**

**Minor**

*Line 24 – flood may not be the best word choice here, please revisit ("affect"?)*
**Response:** We rephrased the first sentence and included a concise definition of Atmospheric Rivers (ARs):
**Modifications (Line 24f): see specification in AC1**

*Figure 1, Figure 4: locate us in space with lat/lon*
**Response/Modification (Fig. 1 & 5** [old 4]**):** we added lat/lon grid accordingly

*Figure 3 – suggest including a box in (a) to illustrate where the box in (b) comes from.*
**Response/Modifications:** We included a slight rectangle in the top view illustration (a) and renamed the boxes in (b) from "sections" to "corridors" in order to guarantee consistency between both subfigures.
*Line 252 – does this suggestion of 5 sondes at minimum depend on the AR width?*
**Response a):** Concerning both Referee remarks, we put more focus on actual sonde spacing (in distances) rather than number of sondes (see also AC1, Section 3.2).
**Modifications (Line356f), we reformulated: "Still, we emphasis that a minimum sounding spacing of 100-150 km has to be targeted for arctic ARs,[…]"**

**Response b):** This corresponds to 4-8 dropsondes for most of the AR widths in the range of 400-800 km. However in the conclusions we added that the sounding spacing is not only affected by the AR width, but also by the steepness of IVT. For example, in wider but weaker ARs, a spacing of 150 km may be sufficient to accurately derive TIVT, but in a narrower but stronger AR (greater steepness of IVT) we recommend a spacing of maximum 100 km.
**Modifications (Line 664ff) we added:" Larger AR width is not necessarily associated with higher IVT variability and maximum IVT is more correlated to IVT variability. The planning of sonde releases should thus rely on the steepness of IVT along the cross-section. "**

*Figure 6 – suggest this would be better presented as spatial interval to not require so much information regarding assumptions about plane speed etc. Indicate what the colors represent in the caption.*
**Response/Modifications:** X-axis is changed to spatial intervals. Further information in AC1.

*Line 268 – isn't this larger difference in q expected given the colder air?*
**Response/Modifications:** Yes, it is not surprising. We now constrain on the comparison to other arctic ARs and reformulated the paragraph:
**(Line 366ff): "Through the entire troposphere, q remains below 5 g kg⁻¹ in our arctic ARs. The vertical moisture characteristics in Fig. 8a) resemble soundings of arctic early**

summer ARs at Ny-Alesund demonstrated in Viceto et al. (2022) who showed q values up to 5 g kg⁻¹. However, the winds in our AR cross-sections (Fig. 8) are roughly twice as strong as given in the case study of Viceto et al. (2022). Note that the ground-based station from which Viceto et al. (2022) depicted soundings were basically located at the outflow edge of the AR (with IVT ≤ 250 kgm−1s−1) on the Luv side. Easterly winds were orographically slowed down by the massif of Svalbard. In this sense, the marine arctic ARs we consider are undisturbed. This enables stronger winds whose magnitude is rather comparable to mid-latitude AR conditions."

*If the AR is more moist or more windy, does that affect the spacing requirements to fully capture the structure?*

**Response:** From our sample, we cannot unambiguously specify that windier ARs requires more dropsondes than moister ARs or vice-versa. But as we explain for our strongest AR event (AR3), the winds here were rather constant (and high) along the horizontal AR transect while the internal moisture plume was primarily responsible for the moisture transport variability. Still in AR cases, where strong AR centers approach the sea-ice, we expect rising importance of the wind variability. Upcoming studies could investigate this in more detail in a larger sample.

**Modifications (Line724ff),** we added in the conclusions: **"As our results are mainly based on corridors in the AR exit region, we strongly recommend extending our uncertainty assessment to other AR regions and expect the role of winds and mass convergence to increase in strong AR centers."**

*Line 323 – constant winds in time or in space? Can you refer to one of your figures here?*

**Response:** Here, we spoke of the winds along the cross-sections. We rephrased the corresponding statement as follows:

**Modification (Line 384ff): "While strong moisture transport in AR3 originates from overall strong winds, moisture varies strongly and seemingly dominates the moisture transport variability (Fig. 8 b). Hence, we hypothesise that in strong arctic ARs with intense winds, moisture variability primarily steers IVT variability and leads to the bell-shaped IVT cross-section pattern depicted in Sect. 3.1"**

*Figure 13 – what is the purpose of the colors in the box-whiskers plot?*

**Response:** The colors refer to our colour-coded frontal sector composition as shown in Fig. 4.

**Modifications (caption Figure 14):** We updated the part of the Figure caption: **"Values specify both components (ADV , CONV ) for all frontal AR sectors (colour-coded)."**

**Editorial**

*A general quick read for grammar/word choice (clarity)/readability is warranted although generally the paper is in good shape. A few suggested changes are below (non-exhaustive).*

**Response**: For its final form, we conducted careful cross-reads to assure clarity and readability, and correctness in grammar and word choice.

**Modifications: see documented changes**

*Line 214 – suggest changing "infer" to "investigate"*

**Response/Modification (Line 293):** We changed the wording accordingly.

*Line 219 – suggest rephrase "arises the question, how" to "raises the question whether".*

**Response/Modification:** Due to the restructuring of the paragraph, the corresponding sentence was deleted.

*Line 264 – suggest rephrase "contributes to IVT with roughly 50%" to "contains roughly 50% of the IVT magnitude"*

**Response/Modifications (Line 364):** We changed it accordingly

*Line 265 – remove "even"*
**Response/Modification (Line 365):** removed

*Line 321 – suggest rephrase "are little coherent" to "exhibit little coherence"*
**Response/Modifications (Line 455):** We reformulated it accordingly

*Line 348 – suggest rephrase "neither it considers" to "it considers neither"*
**Response/Modification (Line 491):** we reformulated it accordingly

*Figure 11 – suggest removing "corridors in the" from the caption*
**Response/Modification:** We removed it from the caption

**III)  Additional author's changes**

Several parts of the manuscript were reformulated or restructured, as seen from above. This section lists some major changes that do not directly result from the comments of the reviewers. We only constrain on some major changes. Here, we do no list all minor changes comprising slight changes in phrasing or slight shortenings. Following the suggestions of the reviewer to carefully rewrite the manuscript, a list of all changes would go beyond the scope of this response list. All other changes can be found in the tracked changes file.

**Abstract:**

**Modifications in bold, inplace:**
**"**This study emulates dropsondes to elucidate how adequately sporadic airborne sondes represent divergence **()** of moisture transport in arctic Atmospheric Rivers (ARs). The convergence of vertically integrated moisture transport (IVT) plays a crucial role as it favours precipitation that significantly affects arctic sea ice properties. Long range research aircraft can transect ARs and  **drop sondes to** determine their IVT divergence.  In order to assess the representativeness of future sonde-based IVT divergence in arctic ARs, we disentangle errors arising from undersampling by discrete soundings and from the flight duration
Our synthetic study uses CARRA reanalyses to set up an idealised scenario for airborne AR observations. For nine arctic spring ARs, we mimic flights transecting each AR in CARRA and emulate sonde-based IVT representation by picking single vertical profiles. The emulation quantifies IVT divergence observability by two approaches. First, sonde-based IVT and its divergence are compared to the continuous IVT interpolated onto the flight cross-section. The comparison specifies uncertainties of discrete sonde-based IVT variability and divergence. Second, we determine how temporal AR evolution affects IVT divergence values by contrasting time-propagating sonde-based values with the divergence based on instantaneous snapshots.
For our arctic AR cross-sections, we find that  **coherent wind and moisture variability contribute by less than 10 % to the total transport** Both quantities  **are uncorrelated to a great extent** . Moisture turns out as the more **variable**  quantity. We show that sounding spacing greater than 100 km results in errors greater than 10 % of the total IVT along AR cross-sections. For IVT divergence, the arctic ARs exhibit similar **differences**  in moisture advection and mass convergence across the embedded front as mid-latitude ARs, but we identify moisture advection being dominant. **Overall,** we  confirm their observability with an uncertainty **of around 25-50 % using**  a sequence of at least seven sondes per cross-section. Rather than sonde undersampling, it is the temporal AR evolution over the flight duration that leads to high deviations in divergence components. Dedicated planning of sonde-based IVT divergence purposes should not only involve sonde positioning but rather pursue optimizing the flight duration. Our benchmarks quantify sonde-based uncertainties as a prerequisite to be used for future airborne moisture budget closure in arctic ARs.

**Divergence**

With the aid of the reviewer's remarks concerning the frontal gradients in moisture transport divergence and the emerging revision of our manuscript, we identified erroneous results in our divergence calculation. In specific, we accidently did not calculate the wind divergence from

using both u, v components but considered the absolute values of wind speed. By that our divergence results were direction-independent. In order to conduct a component-wise divergence calculation, we now had to rotate the u, v components in CARRA, as they are oriented along the local grid rotation and not the zonal/meridional direction. In doing so, the results in chapter 5 and chapter 6 (updated Figure 13-16) have changed moderately. The following, you find the updated figures, alongside a specification of differences to the preprint results. Bullet points highlight major differences to the preprint version. We reserve the right to insert here all the sub-chapters that have been rewritten for the sake of brevity.

**4.2 Sonde-based divergence and its representativeness**

**Erratum modifications**:

[Figure]

Figure 13: Vertical contributions from *ADV* (a) and *CONV* (b) to moisture transport divergence (c) for the frontal sectors in AR3. Bold lines represent the continuous AR representation while dashed lines depict the sonde-based representation with deviations as shadings.

**Changes to preprint results:**

- Slightly smaller magnitudes in all components
- warm sector and core values became more positive (less moisture advection and less mass convergence). In particular, the warm sector became rather divergent (Fig. 12c).
- Low-level mass convergence in the cold-sector (exhibited divergence before).

Explanations: Before, we did not consider the u and v components of the wind separately. This directionality, in turn, substantially affects the results. Since the winds are partly elongated with respect to the moisture pattern, we find relevant contributions of cross-section parallel gradients (e.g. in $u \cdot \frac{\delta q}{\delta x}$ and $q \cdot \frac{\delta u}{\delta x}$), that were not pronounced in the previous approach. The current dominant post-frontal mass convergence results from strong changes in wind direction and superimposes rather weak changes in wind speed.

[Figure]

*Figure 13: Box plot of moisture transport divergence contributions to daily moisture budget for all nine ARs. Values specify both components (ADV, CONV) for all frontal sectors (colour-coded). The continuous AR representation (coloured box-whiskers) is compared to sonde-based values (grey box-whiskers). The boxes refer to quartiles and horizontal lines specify the respective mean.*

**Changes to preprint results:**
- Smaller magnitudes in moisture budget contributions than before
- Less frontal gradient, meaning less positive contribution in the pre-frontal sector and less negative contribution in the post-frontal sector
- Mass convergence does not exhibit a clear gradient along the front. In particular, the post-frontal sector now shows the strongest mass convergence against all the other sectors (before divergent).

We updated the text in Sect. 4.2, accordingly. To account for the findings of related work in a more concise way, we distinguish the paragraphs that describe our results from the respective discussions referring to mid-latitude statistics given in Guan et al. 2020. This literature comparison is enlarged against the preprint version. Surprising findings, such as the predominant mass divergence in the core are carved out.

Like in Section 4, the divergence results for the instantaneous perspective have changed, causing modified figures. Both figures are displayed here equivalently.

**5 Sonde-based divergence and its representativeness**

[Figure]

*Figure 15: Comparison of divergence component contributions to daily moisture budget from spatially continuous AR representation referring to either evolving flight values (non-instantaneous) or to the values for the centered hour (instantaneous). Values are given for each frontal sector. Black error bars are identical to the coloured boxes in Fig. 14. Grey values represent centered hour-based values.* Note that for easy comparison, we here kept the legend entries equivalent to the preprint although slightly renamed in the revised manuscript.

**Changes to preprint results:**

- Instantaneous whiskers follow the modified frontal tendencies (smaller magnitudes and weaker gradients along the front)
- Mean errors in contribution (red-dots are less affected) by updated divergence calculations the post-frontal sector
- Highest mean error in cold sector advection remains. Robust dry advection visible in instantaneous view on post-frontal sector

[Figure]

*Figure 16: Total sonde error (orange) and individual errors by only discrete sondes (green) and by non-instantaneous sampling (grey) for daily IVT divergence in each frontal sector and divergence component (Eq.). For all AR cross-sections, positive bars indicate the root-mean square error while error markers and lines depict mean errors in combination with their*

*standard deviations*. Note that for easy comparison, we here kept the legend entries equivalent to the preprint although slightly renamed in the revised manuscript.

**Changes to preprint results:**

- Minor changes in magnitudes
- Equivalent key messages: Non-instationarity counts more than sounding frequency for the sonde-based misrepresentation of moisture transport divergence

We updated the text in Sect. 5, accordingly. The structure aligns to Section 4.2, where we distinguished the paragraphs in a description of the results followed by respective discussions referring to mid-latitude statistics given. This literature comparison is enlarged against the preprint version.

---

## Author Response (AR2)

**Second Point by Point Response for Manuscript, entitled 'Dorff, H. et al. 2023: Observability of Moisture Transport Divergence in Arctic Atmospheric Rivers by Dropsondes', for final publication**

**Table of Contents**

**I)    Response and Changes to the Comments from Anonymous Referee 1 (AC1)**

**Preface:**

We thank again the associating editor, Geraint Vaughan, as well as the Anonymous Referee #1, for the secondary detailed and helpful review. Please find below our responses (in standard font) to the remarks from the Anonymous Referee #1 (in *italics*). Our changes/ modifications in the manuscript are specified **bold**. Line numbers of reviewer comments (grey) are now referred to the lines in the second revised manuscript (**black, bold**).

**Erratum:**

In the budget contributions (in mm per day) from Sect 4 on, we divided by the gravity at two places, causing our very low values. New values were checked many times for correctness and agree more with literature. Verification with ERA5-fields of $\nabla IVT$ showed plausible values. We now provide the values in mm/h, which is more representative for the local time scales at which they contribute. Given this circumvent, we checked all our routines for correctness several times.

**Specific responses to AC1:**

*reconsidered after **major revisions***
*This is my second review of the manuscript by Dorff et al. Still, I think that the content of the paper is worth for publication and I want to acknowledge the authors for addressing many of my comments. The revision made the motivation and contribution of the paper better understandable and the revised structure improved the readability. Especially, the introduction and conclusion improved substantially. However, this work would profit from a clearer writing and communication of the results.*

**Response:** First of all, we are very grateful for acknowledging the improvement of the manuscript and its consideration for publication. Given your feedback, we put more focus on communicating the results in a more concise way. Your remarks significantly helped out us to improve the readability in the second revision. For all your comments you will find our corresponding responses below, structured in the way our sections are organised.

**Introduction:**

*L25 Change "high" to "strong"*
**Response/Modification: Changed correspondingly.**

*L30 what is "this"? "Extracted" means originating?*
**Response/Modification (L30f):** in agreement to AC2, we reformulated to: **"For the required moisture of ARs impacting the Arctic, the North Atlantic to the south is a dominant uptake zone (Vazquez et al., 2018)."**

*L34: specify what you mean with "air mass transformations" with respect to T, q etc.*
**Response/Modification: see comment AC2, we meant e.g. the thermodynamic vertical structure.**

*L37-39 these two sentences about IVT divergence do not fit here. Before and after IVT distribution is discussed. L38: "links (…) induction" - makes no sense to me*

**Response:** Given both comments, we reformulated the phrases.
**Modification (L36ff): "To illuminate moisture transformation processes occurring in arctic ARs, it is crucial to grasp spatial characteristics of the moisture transport, i.e. the vertically integrated water vapour transport. For instance, Seager et al. (2013) point out that the divergence of IVT links the local temporal development of moisture amount with the efficiency of precipitation formation. When we thus target to derive IVT divergence in arctic ARs, a prerequisite is to resolve the spatial variability of IVT."**

*L42 Rephrase "widely seen"*

**Response/Modification (L41ff):** we have rephrased as: **"Along such lateral cross-sections through the AR centre, airborne observations in mid-latitude ARs have shown a bell-shaped IVT distribution, having the strongest moisture transport in the AR core [..]."**

*L43 What does "this" refer to? What is a "heterogenous spatial variability"? Heterogenous distribution of IVT or high spatial variability would be clear.*

**Response/Modification (L44):** changed to the latter suggestion.

*L44 Define "arctic AR conditions", L45 Rephrase "reflects to" – do you mean "influences"?*

**Response/Modification (L44ff):** Here, we meant to state that the conditions omnipresent in arctic ARs are unclear, so that we reformulated:**" For the conditions in arctic ARs containing weaker moisture transport than in the mid-latitudes, […] and how this influences IVT divergence."**

*L46 delete "variability"*

**Response/Modification (L47ff):** changed according to AC2 to: **"High-resolution observations of IVT variability are not available for arctic ARs."**

*L50 What does "Similarily" refer to?*

**Response/Modifications (L51):** Changed to **"Based on a similar principle,"**

*L52 I think you cannot use "subject allows to verb" in English, it needs to be "subject allows object to verb"*

**Response/Modification (L53):** changed to **"allow derivation of"**

*L60: "with" should be "of"*

**Response/Modification:** Done

*L62-63 "Deteriorations of the representation…" and "The sondes may misinterpret…" are complicated sentences. Try to formulate in a clearer way that the IVT determined from a discrete number of sondes measured along a research flight may differ from instantaneous values on the high resolution reanalysis grid"*

**Response:** We reformulated L62-63 to be more precisely. However, we want to focus on the sonde measurement principle and find it slightly distracting to already refer to the reanalysis here. We look for the sonde-based deviations to the truth, and just use the reanalysis as a benchmark for this.
**Modification (L63ff): "A limited number of sondes can cause deviations in the airborne representation of AR moisture transport variability if the sounding spacing becomes**

**too coarse to reflect the spatial variability of IVT. Such deviations in IVT variability come with misinterpretation of the IVT divergence."**

*L65: What does "an agreement up to 3% for airborne results" mean?*
**Response/Modifications (L67)***: We removed "for airborne results".*

*L66: "Contrasting…" The sentence does not fit here.*
**Response/Modifications (L67ff):** We rephrased the sentence to: **"Accurate airborne estimates of TIVT, juxtaposed for two separate AR cross-section legs, then provide an initial estimate of IVT divergence in between both legs."**

*L69: Would be good to know the spacing in Ralph et al.*
**Response/Modifications (L69):** Both, Ralph et al. (2017) and Guan et al (2018) refer to the same observations. Thus, we added: **"Enlarging the aforementioned sonde spacing".**

*L74: Makes no sense – "but"?*
**Response/Modification (L75ff):** We reformulated the phrasing as: **"For instance, in a polar AR case study, Terpstra et al. (2021) identified incoherent patterns of moisture and wind forming the moisture transport pattern, that are less aligned than in mid-latitude ARs."**

*L75ff Explain the "disentanglement of both quantities" and how you think it may "facilitate flight strategies"? How do you "spend more investment on the airborne representation"?*
**Response/Modification (L77ff):** We modified the concerning paragraph as follows:
**"Unravelling moisture transport into wind and moisture can improve observational strategies of airborne moisture transport divergence in arctic ARs. Especially, if the moisture transport variability (and divergence) were e.g. mainly controlled by the moisture field, supplementary remote-sensing should be involved in the airborne representation of AR moisture. For this reason, it is important to determine whether moisture and wind are aligned in AR cross-sections and to ascertain:** *How correlated are moisture and winds in arctic ARs, and do coherent patterns contribute significantly to IVT (Q2)?*

*L83 Specify "vertical moisture and wind domains" and "frontal gradients in divergence characteristics"*
**Response/Modification (L81ff)***: We have rephrased and specified the aforementioned expressions for clarity and shortened other parts of the paragraph as:*
**"Knowing the spatial structure of IVT is a prerequisite for flight planning and reveals insights into the moisture transport divergence pattern in arctic AR cross-sections. Since ARs primarily occur at the interface of the cold front and warm conveyor belt in extratropical cyclones (Dacre et al., 2019), different dynamic and thermodynamic processes act on the moisture transport and its divergence across the embedded front (Cobb et al., 2021). For mid-latitude ARs, Cobb et al. (2021) found significant differences in vertical moisture and wind for different sectors across the front, which are reflected in gradients in IVT divergence before and rear the front (Guan et al., 2020). Using reanalysis data, Guan et al., (2020) specified lateral differences of moisture transport divergence across centres of ARs."**

*L91ff: "conducted airborne studies" I guess this can be deleted.*
**Response/Modification: deleted**

*"Such research flights" – there was no information about flights, yet. Be more specific about "interpret the discrepancies". "between" >> "between different". Why "In contrast"?*

**Response/Modification (L94ff):** we reformulated the corresponding part of the paragraph as:

**"Norris et al. (2020) determined IVT divergence in mid-latitude ARs from dropsondes. Their research flight allows interpreting the quantitive discrepancies of IVT divergence in ARs that Guan et al. (2020) found between different reanalyses. Norris et al. (2020) point to the large variability of IVT divergence at spatial scales of 50 km, which has implications for sonde-based sampling best practices."**

*L96: Please better explain "impaired by a time component"? I guess you want to say that the estimates derived from airborne observations cover a certain time period and as the state of the atmosphere may change with the evolving weather systems, these values may not be representative for instantaneous values which are typically derived from models.*

**Response/Modification (L98ff):** Exactly, this is what we want to introduce with L96, as we touch upon a new field (temporal evolution). Details were already given afterwards, but we now focus on more clarity. For this, we modified both phrases as:

**"[…] we hypothesise that airborne results are also impaired by the flight duration: Over the duration required to enclose an AR corridor, the state of the atmosphere changes, i.e. there is relevant temporal evolution of the AR (its life cycle and spatial displacement) that causes the sonde-based values to deviate from the instantaneous IVT divergence."**

*L101: These two paragraphs could be merged*

**Response:** We decided to keep them separated, since the following paragraph summarises how we realise our approach, i.e how we examine our research questions, while the last paragraph solely constitutes a brief outline of the manuscript.

*L102: What does to "to constrain on" mean? Do you want to say that you restrict your analysis to ARs in spring?*

**Response/Modification (L105)**: Changed to **"restrict"**.

**Section 2:**

*Section 2 better focuses on the methodical aspects. I suggest moving Sect. 2.2 (Fig. 1 and 2) and merging it with the discussion of Fig. 11 at the beginning of Sect. 3 (comment below).* (originally referred to Sect. 3), *Section 3 would profit from a joint discussion of Fig. 1, 2 and 11 at the beginning, which would ease the discussion of the remaining Figures.*

**Response:** We decided to keep the presentation of our AR cases (Fig. 1 and 2) in the second section, as it represents our data & method section. Furthermore, Fig. 1 is used to illustrate the flight patterns positions. Nonetheless, we took your suggestion into account and merged Fig. 1 & 2 and Fig. 11 in our manuscript, but in Section 2. We included the original Fig. 11 as Fig. 2, after introducing the AR patterns in terms of IVT and glimpse the vertical characteristics of moisture and winds. At this place, we shortened the description of Figure 11 and motivate that the case-to-case variability has to be considered in the following analysis, e.g. when interpreting the results in Sect 3.3/3.4. Afterwards, the original Fig. 2 (now Fig. 3) still suits for a precedent climatological framing of our cases that should be remembered in the following analysis. Another reason for keeping all three figures in Sect. 2 is that, in Sect. 3, we want to follow the stringent change of perspective from vertically integrated AR sonde representation (in terms of IVT) towards the decomposed variability/ coherence of wind and moisture in the vertical profiles across the AR filament.

**Modifications (L182ff).**: Placing the original Figure 11 as Figure 2, we inserted the following descriptive paragraphs: **"Inspecting the vertical curtains of AR cross-sections, that are based on the southern red transects in Fig. 1, the specific humidity exceeds 4 g kg$^{-1}$ in almost every cross-section (Fig. 2). This indicates that our events are rather moist for arctic AR conditions (e.g. Viceto et al., 2022), but still much drier than mid-latitude ARs where q easily exceeds 8 g kg$^{-1}$ (Cobb et al., 2021a). Nonetheless, Fig. 2 depicts several features that we normally know from mid-latitude ARs. For example, this comprises low-level jets (LLJs) that are strong low-level wind corridors (Ralph et al., 2004; Demirdjian et al., 2020). For the windy arctic AR events, e.g. AR3 and AR5, we detect the presence of LLJs stronger than 25 m s$^{-1}$. The LLJ is situated at a height of around 900 hPa, slightly lower than Cobb et al. (2021a) summarised for mid-latitude ARs. Ralph et al. (2004) and Cordeira et al. (2013) found a horizontally slanted vertical structure of moisture transport in mid-latitude AR cross-sections from dropsondes and reanalyses, where Ralph et al. (2017) verified the vertical interaction between the upper-level jet and the LLJ as dominant for the AR moisture transport. In Fig. 2, their conceptual depictions reflect mostly for AR5. Here, moist air masses residing in the cyclonic warm conveyor belt are lifted over the cold front sector. The downward intrusion of the upper-level jet on the western flank causes the slanted structure in the moisture transport.**

**In other arctic ARs than AR5, we find less agreement with the conceptual AR schemes presented in Ralph et al. (2017). This yields for the vertical structure of moisture, the presence and the intensity of the LLJ which is strongly distinctive in AR1, AR3, AR5, AR7, but missing there. In some cases, e.g. AR9, the upper-level intrusion is accompanied by strong dry air subsidence that reinforces the slanting of the moisture transport pattern in the mid and lower levels. The variety of characteristics of winds, moisture, and its transport comes with the different synoptic patterns (such as troughs, ridges, smaller cyclones embedded in a meridional, but weaker flow) that cause the arctic ARs. For example, we find the slanting most effective for ARs close to Eastern Greenland (AR2) or when the backside of the embedded cyclone strongly advects the dry Greenland air masses (AR9)."**

*The subdivision into 2.3.1. and 2.3.2 is not needed.*

**Response/Modifications:** We restructured Sect 2.3. Sect 2.3.1 was removed. Furthermore, we removed the short Sect. 2.3.2, but merged it with Sect 2.4 in order to keep the sonde-based perspective bundled. This caused the following renaming of the section names.

**Section 2.3: Emulating flight patterns to sample ARs, Section 2.4: Divergence derived from synthetic sondes, Section 2.5: AR sectors and sonde locations**

*L190f […] preceded by unnecessary introductory statements about what is done next.*

**Response:** This has been removed by merging section 2.3.1 and 2.3.2.

*It should be explained more carefully how flight pattern are defined.*

**Response:** We restructured the concerning paragraph and specified the flight pattern definitions more thoroughly as:

**Modifications (Line 224ff): "We place such zigzag flight patterns over the AR corridors at the sea ice edge (Fig. 1). We orientate the cross-section legs orthogonal to the major IVT direction and extend the legs such that they transect the entire AR, as long as the AR boundaries remain over open ocean and sea ice and disregard land. We obtain the boundaries from AR catalogue of (Guan,2022). The meridional distance between both cross-sections is assured to be larger than 200 km, but closer than half the lateral AR width. The final distance is chosen individually by visual inspection, as we allow flexibility for the actual flight planning. Due to their proximity to the sea ice edge, the transects of AR corridors are mainly located north of AR centres (horizontal lines in Fig. 3)."**

*It should be explained more carefully how […] the simulated dropsondes are placed (Sect. 2.3). This part is confusing. For example, you describe "six" sondes, but Fig. 5 shows seven sondes. Please explain how the number was defined and how this refers to the methods.*

**Response (i):** In the previous Sect. 2.3, we intended to only present the sonde emulation, without further specifying where actually placing the sondes. In contrast, in Sect. 2.5, the description of gradients along the AR cross-sections due to the presence of embedded cold-front structures allows for further specification on how to place the sondes in the sectors. This has been modified to be more explicit (see comments for Sect. 2.5 below). Note that in Sect. 4.1 and 4.2 the sonde placing is equidistantly, as the sector classification is less relevant when the total cross-section IVT is examined.

**Response (ii):** When we speak of six sondes with respect to Fig. 6, we mean three sectors from each cross-section (six in total) that a span a single sector, and this corresponds to seven sondes per cross-section.

**Modifications (i):** We renamed Sect 2.5 to **"AR sectors and sonde locations".**

**Modifications (ii, Line 302ff):** we reformulated: **"Using seven synthetic sondes per cross-section of the AR, we locate the sondes so that three sondes each in the in- and outflow cross-section span one out of three frontal sector (Fig. 6) and calculate its IVT divergence, respectively."**

*The sector classification (Sect. 2.5) should be more specific. Throughout the manuscript you often talk about a "frontal classification" or "frontal sectors", although it is never described how the classes based on IVT thresholds relate to the cold front location. Additionally, the boundary between core and pre-frontal sector is not a frontal boundary.*

**Response (I):** We specified the frontal characteristics that are omnipresent in the vicinity of the AR and further explained the expected location of the cold front. We further checked throughout the manuscript if the term "sector classification" is less misleading than "frontal".

**Modification (I, 277ff):** We rewrote the first paragraph as: **"Research considering IVT divergence in ARs suggests distinguishing between different sectors along the lateral**

AR cross-sections. Guan et al. (2020) highlight that different dynamics take place across the cold-frontal structures that are commonly embedded in the AR, which itself is situated at the western end of warm conveyor belts (Dacre et al., 2019). Hence, Guan et al. (2020) separate IVT divergence calculations across the major AR axis and the AR embedded front. Similarly, Cobb et al. (2021a) classified different sectors in ARs based on the position of the AR embedded cold front and IVT shape of airborne observations of a large set of pacific AR cross-sections. Both approaches distinguish between frontal sectors, namely a pre-frontal (warm) sector, the AR core with highest IVT, near which the cold front is expected (Ralph et al., 2017), and the post-frontal (cold) sector behind the cold front. Since there exist significant differences in moisture transport divergence between the sectors (Guan et al., 2020), a large part of the variability is smoothed out when calculating IVT divergence for entire cross-sections."

**Response (II):** Indeed, the terms are conventions from precedent studies, and we agree that the boundaries cannot be considered as real robust frontal boundaries. However, we can be certain that our so-called pre-frontal sector mostly contains warm airmasses ahead of the front, while the post-frontal sector primarily contains colder airmasses prevailing behind the front. As already done in the first paragraph (see above), we inserted an additional clarifying statement in the second paragraph, but kept the terminologies in the remainder of the manuscript to ease readability. We agree that the explanation of the outer edges of the pre- and post-frontal sectors is misleading, as they actually focus on significant IVT that belongs to the AR.

**Modifications (II, Line 290ff):** For our sector classification we reformulated: **"Following Ralph et al. (2017), we expect the cold front in the vicinity of the AR core. We denote the region east of the core as the pre-frontal sector mainly containing warm airmasses and west of the core as the post-frontal sector that reaches out of the cold front in colder airmasses. Since we focus on the AR relevant regions with high IVT, we restrict the outer extents of both extra-frontal sectors. For both sectors, we assign the outer edges where IVT >0.33 IVT$_{max}$ to account for case-specific relative values (Fig. 5).**

**(Line 299ff) […] Both thresholds form the outer edges of the AR where our pre- and post-frontal AR sectors end. Note that our sector terminologies only categorise the prevailing air masses of an AR, but should not be viewed as a synoptic cold-front identification.**

*Figure 4 is confusing as it extends from left (east) to right (west). I wonder why you don't not use the same case in Fig. 4 and Fig. 5.*

**Response/Modification:** We now have chosen the same case for Fig. **5**(4) and Fig. **6**(5) and added a clarification in Fig. **5**(4) as: **"The orientation of the x-axis is in flight direction, from west to east."**

*The sondes are not equidistant and do not lie at the boundaries of the sectors (at the intermediate time). Explain!*

**Response/Modification:** We clarified the explanations, that were too confusing before, as: **"Given the varying sector lengths, the sonde spacing is not equidistant in Fig. 6. Additionally, the comparison to the IVT contours in Fig. 6 reveals that the sondes do not lie at the sector boundaries at the intermediate timestep. Our IVT-based AR sectors, i.e. sonde positions, are defined for each airborne cross-section representation individually and thus do not refer to IVT conditions at an intermediate time step. In fact, there is a north-eastward displacement of the AR filament over the course of the 2.5 synthetic flight pattern. For this reason, the sectors along the flight track tilt while the internal IVT has a northward orientation."**

*How would the placement of the sondes be considered in flight planning to cover the sectors best?*

We come back to this in the conclusions where we mention concrete suggestions. Overall, as stated before when focusing on placing a sonde at the forecasted IVT maximum and placing the others symmetrically around is a simple and quite promising approach. Further specifications are out of the scope of the feasibility during flight. Given our thresholds, it is recommended to place all sondes in regions where IVT is simulated to exceed 100 kgm$^{-1}$s$^{-1}$.

**Section 3**

*L287-291 [...] preceded by unnecessary introductory statements about what is done next.*

**Response:** We agree that the paragraph was overloaded with outlining statements. We carefully revised this part, but still placed some overarching statements, that, to our opinion, ease the logical order and readability of the remaining section. We link each of our research questions $Q_i$ to the corresponding Section. Given the recommendations by AC2, we remind for the large case-to case variability of ARs, and we state again that CARRA represents our truth in which we mimic the soundings.

**Modification (Line 312ff):** We reformulated as: **"To examine the moisture transport variability in arctic ARs, we follow a two-fold approach. First, we stick to the plane perspective and determine the maximum distance between sondes needed to derive the total IVT in AR cross-sections accurately (Q1). The synthetic soundings assess the observability of AR moisture transport by discrete sondes. Since we lack real observations, we declare the AR representation in CARRA as our truth. Second, we analyse to what extent coherent patterns in moisture and wind speed anomalies contribute to moisture transport and its variability (Q2). It is crucial to link the results to the large case-to-case variability with respect to IVT magnitude (Fig. 1) and the vertical moisture and wind fields (Fig. 2)."**

*would profit from a joint discussion of Fig. 1, 2 and 11 at the beginning, which would ease the discussion of the remaining Figures.*

**Response:** The shift to Sect. 2.3 of Fig 11 already helped to merge the discussion. At the beginning of Sect 3, we will only place a short reference back to the case-to-case variability. The detailed linking to the remaining Figures of Sect 3 and, in particular, Sect 3.3 and 3.4 are still drawn in the corresponding sections. Thus, we remained at inserting the statements shown in our precedent answer.

*In Sect. 3.2 the lengthy and complicated introduction of Fig. 7 (L322-335) should be shortened. The comparison to different thresholds is not revealing and impairs the description of Fig. 7.*

**Response:** We have removed too much details in the first comparison referring to AR1. Furthermore, we have shortened the second paragraph in its introduction of Fig. 7 and focused on the key messages (relevant sonde spacing) earlier.

**Modifications (L347ff):** First two paragraphs:**" The accuracy in sonde-based TIVT of an AR cross-section depends on the number of sondes across the AR, i.e. their spacing (Ralph et al., 2017). Larger spacing of sondes affects the derived moisture transport variability, whereby the sonde location becomes increasingly relevant. For example, the equidistant placing of six sondes within AR1, as shown in Fig. 7, underestimates TIVT by roughly 10 % against the continuous IVT representation. For all of our ARs, we assess the sounding spacing, needed to gain adequate TIVT estimates, by varying the density of synthetic sondes and by comparing their TIVT values with those of the continuous AR cross-section representation in CARRA. To account for the dependency on sounding location, we conduct a bootstrapping approach in which we sample the cross-sections with varying release spacings and varying sounding locations, from which we derive TIVT. From this, the grey box-whiskers in Fig. 8, showing the distribution of sonde-based errors of TIVT, reveal that the relative error of TIVT against the continuous AR representation increases significantly for sounding spacing ≥ 150 km. This corresponds to roughly five sondes for the given cross-section lengths, and release intervals above 10 min at a cruising speed of 250 m s$^{-1}$). For sonde spacing ≥ 200 km, sonde-based TIVT can substantially deviate."**

*I do not understand the analysis of moisture variability in Sect. 3.3 extending into the post-frontal cold sector where dry descending air is located (Fig. 11). This certainly doesn't characterize the relative role of q and winds for variability of strong moisture transport in the AR (L385).*

**Response:** We checked that less than 10% of our cross-section lengths reach out of the AR edges. Our cross-sections are aligned to the AR shapes defined in the AR catalogue of Guan et al. 2022, that refers to vertically integrated IVT thresholding, representing the most common AR classification. There may occur coexisting air masses along the vertical profile. Accordingly, in AR domains, there can exist substantial dry descending air above, while the moisture transport underneath is still designated as AR. If we neglected these regions, we would also neglect large cross-section parts being designated as AR from the IVT perspective. Therefore, we argue that our inclusion of the vertical columns with dry air intrusions are valid.

**Modification (Figure 2, caption):** To achieve clarity for the reader in advance, we added the AR pattern based on the AR catalogue of Guan et al. (2022) in Figure 1 and referred to it in the caption of Figure 2 as: **"As visible in Fig. 1, some ends of the cross-sections already reach out of the ARs, but this constitutes less than 10% of the cross-section lengths."**

*Consider shortening the discussion about orographic effects at Svalbard, which seems not to be relevant.*

**Response/Modifications (L383ff):** We shortened this part as follows **"However, the winds in our AR cross-sections (Fig. 9a) are roughly twice as strong as given in their case study, which reports an orographic deceleration by Svalbard. For our arctic ARs, the open ocean enables stronger winds, rather comparable to the wind conditions in the mid-latitude ARs.**

*Are "coherence" and "coherent" and "non-coherent" transport established terminologies in this context? I wonder why the transport driven by small scale fluctuation is named coherent?*

**Response/Modifications:** Yes, these terms are established to emphasise that the small-scale fluctuations of wind and humidity must be correlated. Random fluctuations cannot generate a flux. Only coherent anomalies or patterns of these variables lead to moisture transport.

*Moving Fig. 11 at the beginning of Sect. 3 would ease the interpretation in Sect. 3.4. The discussion is rather long and repeats things that were already discussed (e.g. ~L370).*

**Response:** The discussion in Sect 3.4 now has shorten, as some text parts introducing the AR curtains are now listed in Sect 2.2. Consequently, we have several cross-references to the original Fig. 11 (now Fig. 2) to interpret our results.

**Modification (L436ff):** See for example: **"Apart from AR5, other ARs exhibit less coherent patterns where wind and moisture do not necessarily correlate with each other (see also Fig. 2). Valid for most of the ARs, the correlation between moisture and wind peaks in the LLJ height. The negative correlation in Fig. 11 refers to AR9 that indicates a clear horizontal displacement of the wind and moisture fields (Fig. 2). Here, subsiding dry air masses in the cold sector counteract the westward increase of wind speeds.**

**Summarizing Fig.10 and Fig. 11, the moisture variability mainly steers the moisture transport variability above the marine boundary layer. This shows analogously in more horizontal overlap between fields of moisture and moisture transport as against the wind fields (Fig. 2). Especially, AR1 and AR3 exhibit small horizontal variability in the wind field, as winds are almost constant along the entire cross-section (> 25 m s−1). The ARs, being variable in moisture, consist of an elevated moist plume only residing in the AR core that is surrounded by dry air."**

*Why is the strength of the subsidence determining the slanting (L442)?*

**Response:** We actually intend to connect the zone of dry subsidence with the tilt of the moisture transport. We see the slanting of the moisture transport mostly overlapping with the moisture gradients, resulting from the dry air subsidence caused by the cold front. However, our connection was imprecise. It was thus deleted, as described appropriately in the modifications above.

*How do you know about the distribution of warm conveyor belt air, a concept which was not introduced or used before?*

**Response:** As specified in our responses for Sect 2, we now introduced the warm conveyor belt location with respect to the AR location and refer to Dacre et al. (2019). For our knowledge of the distribution of warm conveyor belt air, we visually inspected reanalysis-based Theta-E charts at 850 hPa to identify its general region. In the manuscript, we do not specify this identification for brevity and easy readability, as we would then distract more from the AR analysis.

**Section 4:**

**Response/Modifications (L462ff):** We agree that there are many introductory sentences, but due to the amount of assumptions and methods used, some of which change between the sections, we want to ensure clarity for the reader as how the following results have to be treated. Nonetheless, we rephrased the section introduction and reduced the outline of all steps, in order to focus more on the connection to the precedent section. For this, we also moved the last statements of Sect 3, as follows:

**"The incoherent cross-section patterns of moisture and wind fields (Sect. 3) suggest lateral differences in the moisture transport divergence components (Eq. 5), and motivate investigating the divergence in separate sectors across the front embedded in the AR. Showing the limits in TIVT-based divergence, we investigate whether high moisture advection occurs more frequently in strong moisture-dominated AR sectors and whether mass convergence dominates in windy AR sectors. By categorising our results based on the AR sectors (Sect. 2.5), we examine how the divergence of moisture transport is characterised along cross-sections of arctic ARs (Q3), and evaluate how the sondes reproduce the features of the continuous cross-section representation."**

*"Idealising (…) takes place" - I cannot find where in Sect. 3 you showed details about fluxes across the eastern and western boundaries. It would be good to know how valid this assumption is. How do these results agree with findings from the other approach?*

**Response (I):** We had an erroneous section reference included. We specified that we meant (Sect. 2.4), in which we, in the first paragraph, describe the simplification of IVT divergence by contrasting out- and inflow TIVT, assuming that no flux across the boundaries takes place. We further slightly modified the corresponding sentence in Sect 2.4:

**Modifications (I, L): "Neglecting the moisture flux apart from perpendicular to the flight track, i.e missing fluxes across the eastern and western boundaries, we can approximate $\nabla$IVT in an AR corridor by the difference of out- minus ingoing TIVT of the cross-sections."**

**Modifications (II, L477): By contrasting the in- and outflow cross-section legs (Fig.12), it can be estimated whether convergence or divergence of moisture transport inside the AR corridor exists, under the idealisation that no lateral entrainment into the AR corridor occurs (Sect. 2.4).**

**Response (II):** It is especially the confluence of post-frontal air masses that is responsible for considerable deviations of the TIVT- based moisture transport divergence against the regression-based divergence (see Sect 4.2). The arctic ARs show significant entrainment through the western boundaries. Moreover, IVT is only an integrated quantity, and thus also of the mean wind direction. In our case, we do also not separate between zonal or meridional $IVT_x$ or $IVT_y$.

**Modification:** In restructuring Sect. 4.1, we put more focus on argueing why TIVT-based divergence is not promising.

**Response/Modifications (L493ff):** When deleting the introductory phrases completely, we perceive the explanations of the following results as too unclear, since the reader is not aware of the all included observation perspectives/assumptions. Instead, we shortened the paragraph as:

**"To derive the IVT divergence ($\nabla$IVT), we thus use the regression-based approach (Sect. 2.4) for moisture advection ADV and mass convergence CONV. The results from the**

**continuous cross-sections are compared to results based on the synthetic sondes that sample the cross-sections (as illustrated in Fig. 6)."**

*The exemplary case study (Fig. 13) and the average integrated values (Fig. 14) could better connected.*

**Response/Modifications (L537ff):** We improved the connection between both figures, and referred back to Figure 13 in a dedicated paragraph: **"The overall pre-frontal moisture advection in Fig. 14 is aligned with the profiles of AR3 (Fig. 13). Pre-frontal moisture advection primarily occurs in the mid-levels. The mass divergence in the core is surprising as most arctic ARs contain LLJs (Fig.2), which are associated with high mass convergence in mid-latitude AR. However, the low-level mass convergence below 800 hPa, found in many of our AR cases like AR3 (Fig. 13), is often superimposed by mid- and upper-level mass divergence above the LLJ (e.g. Fig. 13), as the AR spreads out. The mass convergence in the post-frontal sector marks the highest inter-case variability. The high values of mass convergence, mostly from low-levels as in Fig. 13, mainly arise from two cases (AR3, AR7). Here, we find changes in wind direction, as visible from the surface isobars in Fig. 1, inducing the confluence of moist marine boundary layer air masses. For the sign of post-frontal advection, the effectiveness of subsidence of dry air overrunning the western AR edge in the mid-levels becomes crucial (see Fig. 2 and 13)."**

*The increased moisture convergence is restricted to the marine boundary layer - why is this the case and how are the other features related to the vertical distribution of moisture transport and the synoptic situation?*

**Response:** We briefly sketched this in the manuscript by describing that the low-level mass convergence results from confluence as also remarkable from the surface isobars in Figure 1 (see above). This becomes more effective due to the high specific humidity in the marine boundary layer, that is however less variable in the marine BL, so that moisture advection remains weaker.

*You state that the negative contribution of the core region (Fig. 14) to the moisture budget is different to the extratropics and thus surprising. Can you explain what that means?*

**Response/Modification (L551ff):** It means less favourable conditions for the formation of precipitation. Indeed, inspecting the reanalysis precipitation, we see precipitation foremost from the western half of the AR center, constrained to a narrow swath. Consequently, we inserted into the manuscript:

**"Unlike the mid-latitudes, the upper-level dominating mass divergence in the core of arctic AR lowers the triggering of precipitation by convection. Instead, major precipitation fields are often shifted towards higher reaching convergence west of the IVT maximum (not shown)."**

*To what extent is the cold sector moisture vertically transported and producing rain?*

**Response:** This is a very interesting question following our added argumentation of less favourable precipitation triggering in the core. Still, a robust answer requires supplementary trajectory analyses following the air masses. We consider this as out of the scope of this manuscript. Furthermore, the description of the AR IVT divergence conditions for Figure 14 is already quite detailed so that such analyses would further distract from our primary focus being the sonde-based observability / reproducibility of the moisture transport divergence.

Your question is somewhat related to the answer of the beforementioned comment, showing the western shift of the precipitation field. Nonetheless, in a follow up study comparing the moisture budget components for a specific airborne AR case, we will come back to your point, as we there find similar IVT divergence features.

*Where is that mid-level mass divergence coming from? I wonder whether you can better connect these results to the case study showing the vertical distribution of the CONV and ADV?*

**Response/Modifications (L539ff):** We added the reference to the case of AR3, where we see the same feature: **"However, the low-level mass convergence is found in many of our AR cases, but often superimposed by mid-level mass divergence above the LLJ (e.g. Fig.13), as the ARs spread out."**

*"we recognize (…)" (L532) - why should the divergence be underestimated due to lower vertical resolution?*

**Response/Modifications (L527ff):** Coarse vertical resolution may cause spatial aliasing. This become effective in lower levels where divergence shows larger amplitudes especially in terms of low-level convergence in the vicinity of the LLJ. The divergence in mid-levels exhibits more homogeneity (see Fig. 13). Accordingly, we briefly added in the manuscript: **"Thus, the rather low divergence shown in Fig. 13 is probably not only the result of true lower divergence that prevails in arctic ARs compared to mid-latitude ARs. It can also result from the coarser vertical resolution leading to spatial aliasing in narrow convergent low levels."**

*L556-565: If it can't be compared I suggest skipping the whole discussion.*

**Response/Modifications (L561ff):** The first expression was misleading, as we solely wanted to emphasize that there are numerous differences in our AR sectors compared to those of Guan et al. However, we fully see the validity for the comparison of our values with Guan et al (2020) in principle (paragraph before), since we built on their methods and derive equivalent quantities for the arctic ARs. Instead, we reformulated: **"Nonetheless, the precedent comparison of our sector-based values of moisture transport divergence in arctic ARs to those in Guan et al. (2020) has to consider additional aspects."**

**Section 5:**

*Can you really speak of distortion or error? Isn't it just a difference related to the temporal evolution of the AR in strength and location?*

**Response:** We agree that the term "error" is quite strong and we removed it. However, we are convinced that distortion remains a valid term, as the sonde-based results are generally used to interpret the actual divergence features in air masses. These airborne values can be distorted by the temporal evolution of the AR, causing different gradients in the AR as actually present in the AR corridor at an intermediate time step. In particular, this may cause erroneous conclusions for the analysis of air mass transformation with respect to the moisture budget components in the ARs.

**Modifications (throughout):** Still, we replaced the term "error" for "deviations" due to the temporal evolution, and placed "distortion" or mostly "deviation" instead (also applies for axes in Fig. 15 and 16).

*Please explain carefully how you treated the sonde locations for this comparison? Do both versions use the same locations or are they relocated when using instantaneous IVT?*

**Response:** First in Figure (15), we compare both continuous representations to neglect any sonde spacing dependencies. For our following comparison, we kept the equivalent sonde locations in order to rule out further dependencies in our final analysis. We tested the relocation of the sondes for our IVT-thresholding in both time perspectives, and the divergence values and deviations to the continuous perspective slightly change for the individual cases. However, they do not change to an extent that our conclusions need to be modified. Keeping the sonde locations fixed, we really intercompare the local effect of AR evolution on given sounding positions. This may be of higher interest for later uncertainty assessments in flight campaigns in which the sondes were already released. Accordingly, we inserted in the manuscript:

**Modifications (Line 608f): "To purely attribute the non-instantaneous effect on the divergence estimates at specific sonde locations, we hold the sonde positions fixed in both time perspectives. Thus, we do not relocate the sondes, once the sector-based IVT thresholds are exceeded different locations in the instantaneous representation occur."**

*Why "ideal representation"? I would understand if you talk about a representation of the continuous and instantaneous IVT values by a certain number of sondes (Fig. 16).*

**Response:** The representation, that is instantaneous and continuous, is the optimum airborne perspective for the moisture budget assessment. However, it would require an infinitely fast aircraft conducting soundings continuously and thus represents an idealization. Moreover, it would fulfil instantaneous airmass investigation for connecting the derived moisture transport divergence with local change of water vapour, precipitation and evaporation.

**Modification:** To ensure more intuitive readability, we now term the **"optimum airborne representation"** to ensure that our perspective remains in the observations and their ideal realisation.

**Conclusions:**

*This section significantly improved, but could be shortened at the end. L698-710 are a lot of repetition and the purpose or difference to the preceding conclusions should be made clear.*

**Response:** We shortened this paragraph, and merged some of its sentences with the bullet point questions above. However, although the paragraphs resemble the bullet points, their statements generalise/merge the key messages of the research questions in an overarching way for implications and recommendations for actual flight planning.

**Modifications (L701ff):** "**The synthetic study confirms the observability of moisture transport divergence in arctic AR corridors by releasing sondes in such dedicated flight patterns. Notwithstanding that we could release more sondes, it is the temporal evolution of the AR over the flight duration that leads to higher deviations in the divergence components rather than sonde undersampling. These deviations range from 25--50% of the divergence values. Therefore, the dedicated planning of such sonde-based purposes should not only include the sonde positioning, but also the minimisation of the flight duration. The placement of cross-section legs and their spacing should carefully consider the AR displacement during flight. Shorter distances between the cross-sections not only reduce the flight duration, but also the area enclosed by sondes. Given the widths of the arctic AR sectors, both cross-sections should be no more than 200 km apart. For several of our cases, the meridional separation is higher, and we have to expect considerable subgrid scale variability. The optimal and still feasible strategy represents collocated flights by two aircraft, with both cross-sections being not far apart and sampled simultaneously. Supplementary measurements of moisture should be prioritised due to its higher variability, and its advection dominating** $\nabla IVT$.**

*It also should be mentioned that your conclusions are drawn from simulations and should first be verified by observations, as you do not know to what extent the small-scale variability is reproduced in the reanalysis.*

**Response/Modifications (L712ff):** We added this very valid point at the beginning of the paragraph evaluating the limitations of our study as:

"**Additional limitations of our study need to be discussed. All our conclusions, especially the described AR characteristics, are drawn from simulations and should be verified with observations, as the extent to which the simulations reproduce the small-scale variability is uncertain.**"

*For Q4 I would like to know how the sonde placement and the number of sondes would affect the differences between the instantaneous "truth" and the simulated sonde-based approximation.*

**Response:** We have merged Q4 with some of the statements/repetitions in L698-710 and rephrased/restructured its content in order to better address your valid expectations.

**Modification (L687ff):** "**For reproducing IVT divergence, the undersampling by a limited number of sondes matters. We recommend a sequence of at least seven sondes per AR cross-section. Given the widths of arctic ARs, this corresponds to a maximum sonde spacing of 100 km. Symmetrically placed around the maximum IVT, three sondes per AR sector leg are capable of reproducing the sector characteristics of moisture transport divergence components with similar magnitudes. The mean absolute deviations to a continuous AR representation along the flight reach up to 0.1 mm h$^{-1}$ (Fig. 14). However, the deviations for moisture transport divergence by undersampling are minor compared to the deviations induced by the flight duration that mostly range from 25-50% of the actual divergence values. Non-instantaneous sonde-based estimates deviate the most in the post-frontal cold sector, where we detect steeper**

**gradients in moisture and winds than in the pre-frontal sector. Here, the AR displacement during flight, which is not necessarily along the moisture transport direction, as well as the intensity of dry intrusions on the backside of the AR is relevant for more than twice the deviations in ADV and CONV, partly exceeding 50% of the actual values. Unlike the undersampling, non-instantaneous effects deteriorate the results more consistently. The moisture advection is overall most sensitive to the airborne sampling. In fact, the post-frontal divergence (from ADV and CONV) and the pre-frontal moisture advection are stronger than assumed by the sondes during flight. Although mass confluence is relevant in the post-frontal sector, it is overestimated by sondes**

*Line 731ff: "Consistently mimicking the soundings is a fundamental step towards the understanding of the uncertainties when such airborne tactics are actually carried out." The sentence is correct but very vague. It is not clear to me what "consistently mimicking" means. What is the "fundamental step towards the understanding of the uncertainties" you made – of what uncertainties? And what "airborne tactics" are you talking about?*

**Response/Modifications (L733ff):** We intended to link the added value of our assessment for the future measurement strategies and the analysis of sonde-based results from real campaigns. Thus, we reformulated: **"Emulated soundings assess possible airborne misrepresentation in moisture transport divergence and will improve the interpretation of future real soundings interpretation of future real soundings aiming for the airborne closure of the moisture budget in ARs."**

*"Only by illuminating the constraints on the AR representation from both models and observations, we establish a framework from which airborne observations can support modellers in terms of the resolution and complexity required for parameterisation of moisture transformation processes caused by IVT divergence in arctic ARs.": What means "illuminating the constraints on the AR representation from both models and observations"? – what is your work contributing to model and observation differences? What exactly is `the "framework" that supports modellers in terms of resolution? What "parameterisation of moisture transformation processes" are you referring to?*

**Response/Modification (L736ff):** For more clarity, we reformulated: **"With quantified limitations in the sonde-based AR representation of IVT divergence in arctic ARs, future airborne observations can better assist modellers in terms of the resolution and complexity required to represent ongoing moisture transformation processes."**

**II) Response and Changes to the Comments from Anonymous Referee 2 (AC2)**

**Preface:**

We thank again the associating editor, Geraint Vaughan, as well as the Anonymous Referee #2, for the secondary motivating and helpful review. Please find below our responses (in standard font) to the remarks from the Anonymous Referee #2 (in *italics*). Our changes/ modifications in the manuscript are specified below. Line numbers of minor comments (grey) are now referred to the lines in the second revised manuscript (**black, bold**).

**Specific responses to AC2:**

*accepted subject to **minor revisions***

*I appreciate the authors' efforts to address the first round of suggested revisions. I think they have done a nice job providing more background and context, and clarity on the questions they are seeking to answer with this study. I have some remaining suggestions to improve on the presentation/language, and I think the writing could still be substantially improved overall. These suggestions are examples and similar edits (e.g., appropriate word choice, simplifying sentence structure) should be considered throughout. I look forward to seeing this contribution in the literature.*

**Response:** First of all, we are very grateful for acknowledging the improvement of the manuscript and its consideration for publication. Given your feedback, we put more focus on the writing and presentation style.

**Abstract:**

*"adequately sporadic" does this mean "adequately spaced"?*
**Response/Modifications (L1f):** No, it was meant that "adequate" refers to the representation, but it was imprecise. We reformulated the sentence as: **"This study emulates dropsondes to elucidate the extent to which sporadic airborne sondes adequately represent divergence of moisture transport in arctic ARs."**

*Line 20 – "rather" change to "also"*
**Response/Modification (L20):** Done
*Line 21 – "quantify" change to "identify"*
**Response/Modification (L21):** Done

**Introduction:**

*Line 30: – does this mean that the moisture is from the south? Please rephrase for clarity. Suggestion: "The North Atlantic to the south is a dominant moisture update zone for ARs affecting the Arctic"*
**Response:** Yes, we meant this. We slightly changed your suggestion as follows:
**Modifications (L30f): "For the required moisture of ARs impacting the Arctic, the North Atlantic to the south is a dominant uptake zone (Vazquez et al., 2018)."**

*Line 31 – does "in an interplay" mean when ARs are associated with the other features mentioned? Suggest clarifying here.*
**Response/Modifications (L31):** Changed to **"Embedded in"**

*Line 34 – suggest defining "air mass transformation" briefly even if you are reiterating how it is used in You et al. 2022.*

**Response/Modification (L34):** We added a specification stating that is meant in the vertical structure of the atmospheric conditions: **"[…] air masses are subject to transforming thermodynamic vertical structures. "**

*Line 39 – "considered" change to "used"*
**Response/Modification (L40):** Done

*Line 40 – "specify" change to "identify" or "investigate"*
**Response/Modification (L40):** we chose **"investigate"**

*Line 45 – "reflects" do you mean "influences*
**Response/Modification (L46):** changed to **"influences"**

*Line 46, suggest rephrase to "High-resolution observations of IVT variability are not available for the Arctic"*
**Response/Modification (L47):** changed accordingly.

*Line 47 – by "sporadic" do you mean infrequent here?*
**Response/Modification (L47):** changed accordingly.

*Line 52 "allow to derive" change to "allow derivation of"*
**Response/Modification (L53):** changed accordingly.

*Line 56: "The overall goal of the study is to assess the observability of moisture transport divergence in arctic ARs by dropsondes. This assessment comprises (a) The role of dropsonde frequency,(b) The influence of correlated moisture and wind fields on moisture transport, (c) Dropsonde capacity to reproduce IVT divergence in arctic ARs, (d) Impact of extended flight duration". I think this could be clarified – bullet 3 is basically the same as the overall goal. Item 2 should be clarified in terms of how that relates to the observability of moisture transport.*
**Response/Modifications (L57ff):** We rephrased the long sentence and redefined bullet point 3 as: **"The overall goal of this study is to assess the observability of moisture transport in arctic ARs by dropsondes. The assessment targets the facilitation of measurement strategies in dedicated research flight campaigns, as e.g. proposed by Wendisch et al. (2021). It includes (a) the role of sonde frequency, (b) concretising the need for supplementary measurements based on moisture and wind variability, and c) the impact of extended flight duration under evolving AR conditions on the dropsonde capacity to reproduce IVT divergence in arctic ARs."**

*Line 65 – remove "being"*
**Response/Modification (L66):** Done

*Line 94 – does this mean sonde-based sampling best practices? Or results? If results, consider clarifying what is meant.*
**Response/Modifications (L96f):** We want to more focus on the best practices here and rephrased: **"[…] which has implications for sonde-based sampling best practices."**

*Line 102 – suggest "constrain our analysis" or "focus" instead of "constrain"*
**Response/Modifications (L105):** In accordance with AC1, we rephrased: **"restrict our analysis to"**

**Section 2:**

*Line 155 – "principle" should be "principal"- unless you mean its observability in principle*
**Response/Modifications (L158)**: We changed to **"principal"**

*Line 158 – "restrict to conditions and events" change to "selected events"*
**Response (L161):** Done

*Line 158 – not sure what "Reacts very prone" means – does it mean the sea-ice is more vulnerable somehow to the influence of ARs during this season? Consider clarifying this language.*
**Response/Modification (L160f):** We rephrased: **"[…] when maximum sea-ice extent in the Arctic Ocean starts to break-up and is more vulnerable to the intrusion of warm and moist air (Rostosky et al., 2023)."**

*Line 163 – "to be conducted" change to "to conduct".*
**Response (L166): Done.**

*Line 164 – "2020, presented in Fig. 1" change to "2020 (Fig. 1)"*
**Response (L167): Done.**

*Line 165 – remove "commonly"*

**Response: Done.**

*Line 167-168 – "we confirm a large case-to-case variability regarding the synoptic situation" "we confirm large case-to-case synoptic variability"*
**Response (L170): Done.**

*Line 201 – why can the internal line but not the cross- sections be used for precipitation rate, evaporation, or water load? Couldn't the internal line also get IVT (albeit not along a cross section)?*
**Response/Modification (L219ff):** For sure, all legs can be used to measure all quantities (IVT, IWV, precipitation & evaporation). However, with the physical conception of a budget box (Fig. 4b), the legs have certain focusses. We thus reformulated: **"The boundary cross-section legs perpendicular to the major flow focus on quantifying the corridor in- and outflow, i.e. in- and outgoing IVT over the entire lateral AR extension and enable simplified divergence calculations. Note that the diagonal internal legs can focus on assessing precipitation rate, evaporation or water load inside the AR corridor so that this pattern allows quantifying the remaining moisture budget components of the budget box, i.e. AR corridor (Fig. 4b)."**

*Include a list of things you are neglecting – e.g., dropsonde drift and perhaps a note on why investigating other limitations, e.g., the instantaneous sampling of the full cross section is more important.*
**Response/Modifications (L246):** According to AC1, we reorganized the sections, and placed the emulation of sondes in Sect 2.4, where we more concisely listed our assumptions and focusses as: **"We synthetically refer to the measurement principle of dropsondes (Sect. 1). Along the continuous airborne AR representation of the cross-sections (Sect. 2.3), we depict profiles as synthetic soundings for which we neglect any vertical drift or fall time. We also neglect any measurement uncertainties. Such effects are out of the scope of this study, and assumed to cause lower deviations than our considered effects. Our sondes observe exact IVT values at the release position. Instead, we focus on the spatial representativeness of sporadic sonde-based IVT and evaluate the uncertainties in the**

**lateral variability of moisture transport, and how these uncertainties affect the airborne non-instantaneous perspective on IVT divergence in arctic ARs."**

*Line 259 – not a complete sentence, rephrase, maybe "Current research considering IVT divergence in ARs suggests that it is essential to distinguish between different sectors along the lateral AR cross-sections"*

**Response/Modifications (L277f):** changed to **"Research considering IVT divergence in ARs suggests that is essential to distinguish between different sectors along the lateral ARs cross-sections."**

*Line 269/throughout – I think maybe you mean "example" rather than exemplary? Consider rephrasing for clarity.*

**Response/Modification (throughout):** We checked our manuscript and rephrased accordingly where appropriate.

**Section 3:**

*Section 3.1 – consider noting that this is assuming that ground-truth is as appears in the reanalysis, and perhaps note that we may learn more as we start making observations in these regions. /throughout*

**Response:** Since this is an overarching information holding for the entire Sect 3 (and the remaining results), we placed it in the introductory part of Sect 3 where we also refer back to Fig. 1 and 2, i.e. the AR cases and their case-to-case-variability as desired by AC1.

**Modification (L312ff):** We reformulated: **" To examine the moisture transport variability in arctic ARs, we follow a two-fold approach. First, we stick to the plane perspective determine the maximum distance between sondes needed to derive the total IVT in AR cross-sections accurately (Q1). The synthetic soundings assess the observability of total moisture transport by discrete sondes. Since we lack real observations, we declare the AR representation in CARRA as our truth. Second, we analyse to what extent coherent patterns in moisture and wind speed anomalies contribute to moisture transport and its variability (Q2). It is crucial to link our results to the large case-to-case variability with respect to IVT magnitude (Fig. 1) and the vertical moisture and wind fields (Fig. 2), and to attribute the results to synoptic AR precursors."**

*Line 301 – remove "Only"*

**Response/Modification:** we removed the entire phrase at it is too redundant here and thus distracting.

*Line 347 – suggest rephrase to "Too coarse sonde spacing enhances the likelihood that the sampling will not capture the strongest IVT" or something similar. I don't think you mean to say that you will miss the IVT overall with the transect focusing on the IVT.*

**Response/Modification (L365ff):** We inserted: "**Too large sonde spacing enhances the likelihood that the sampling will not capture the region of strongest IVT.**"

*Sometimes the authors state "Arctic ARs" and other times "arctic ARs" consider being consistent with this throughout.*

**Response/Modification (throughout):** We cross-checked the manuscript for consistency and speak of **"arctic ARs"**.

*Line 389 – what are the supplementary remote sensing devices? Are they successful at getting near-surface moisture fields in the presence of precipitation? If not maybe state "the ability of supplementary remote sensing devices […] to derive moisture fields should/could be explored"*

**Response/Modification (L399ff):** changed to: **"The identification of the more variable quantity can improve measurement strategies. Specifically, in case of a moisture dominance, the ability of supplementary remote sensing devices from the research aircraft to derive moisture fields could be explored. "**

*Line 445 – do you mean causing here? Or "associated with"?*

**Response/Modification (L197f):** This sentence has moved to Sect 2.2, introducing the AR cases and their variability. We really meant "causing", but reformulated: **"The variety of characteristics of winds, moisture, and its transport comes with the different synoptic patterns that cause the arctic ARs (troughs, ridges, smaller cyclones embedded in a meridional, but rather weak flow)."**

**Section 4:**

**Response/Modification (L474):** Done.

*Line 519 – not sure what you mean by "differ quietly to" – I think it should be rephrased to "differ from"*
**Response/Modification (L512f):** Changed accordingly.

*Line 522 – this full sentence is confusing, rephrase – potentially "The lack of pre-frontal moisture advection in AR3, found robustly in mid-latitude AR statistics (Guan et al., 2020), is worth mentioning."*
**Response/Modification (L514f):** Changed accordingly.

*Line 568 – do you mean the observability, in principle? Or principal observability?*
**Response/Modification:** We meant the latter and modified accordingly.

*Line 478 – "narrow" change to "wide":*

**Conclusions:**

**Response (L698):** Done.

**Response/Modification (L701f):** we reformulated: **"The synthetic sondes confirm the observability of moisture transport divergence in arctic AR corridors by releasing real sondes in zigzag flight patterns in the future."**

---

## Author Response (AR3)

**Third Point by Point Response for Manuscript, entitled 'Dorff, H. et al. 2023: Observability of Moisture Transport Divergence in Arctic Atmospheric Rivers by Dropsondes', for final publication**

**I)   RESPONSE AND CHANGES TO THE COMMENTS FROM ANONYMOUS REFEREE 1 (AC1)**

**Preface:**

We express our condolences that the Anonymous Referee #1 (AC1) invest this effort in continuously providing us these very detailed and helpful comments, which significantly improve the manuscript. We have adressed all remarks from AC1 and have adjusted all relevant text passages accordingly. The manuscript was read again as urged by both the Referee #1 and the handling editor. We additionally thank the editor for his continuous support.

We here only list the manuscript changes that require further clarification. All other edits are conducted as clearly suggested by AC1. Please find below our responses (in standard font) to the remarks from the Anonymous Referee #1 (in *italics*). Our changes/ modifications in the manuscript are specified **bold**. Line numbers of reviewer comments (grey) are now referred to the lines in the second revised manuscript (**black, bold**).

**Specific responses to AC1:**

*reconsidered after* ***major revisions***
*This is my third review of Dorff et al. and I thank the authors for carefully addressing all of my comments. Although the manuscript improved, I still see deficits in the presentation of the results, especially grammatical edits would be valuable to get more clarity of the arguments. I went through the manuscript again and I suggest that the authors consider the comments carefully to improve the readability.*

**Response:** First of all, we are very grateful for acknowledging the improvement of the manuscript and its consideration for publication. Given your feedback, we checked and modified all the given points listed above. Your remarks significantly helped out us to improve the readability. For all your comments, you will find our corresponding responses below.

*L5/L70: It is not needed to change "error" everywhere but it should be defined what it is. If you talk about deviation then it should be explained from what?*

**Response/Modification (L5f):** we rephrased: **"[…] we disentangle the sonde-based deviations from an ideal instantaneous IVT divergence, which result from undersampling by a limited number of sondes and from the flight duration."**

*L17: What is "their" referring to?*

**Response/Modification (L17f):** changed to: **"we confirm the observability of IVT divergence with an uncertainty […]".**

*L19/20: "Dedicated planning of (…) purposes (…)" makes no sense to me.*

**Response/Modification (L20f):** We have rephrased: **"In order to realise the estimation of IVT divergence from dropsondes, flight planning should consider not only the positioning of the sonde, but also the minimisation of the flight duration."**

*L21: Please clarify what you mean with this sentence! How can a prerequisite be used?*

**Response/Modification (L21f):** We rephrased: **"Our benchmarks quantify sonde-based uncertainties as an essential preparatory work for the upcoming airborne closure of the moisture budget in arctic ARs."**

*L58-60: What are the research flights proposed by Wendisch et al. and when will they be conducted?*

**Response/Modification**: The research flights were planned for the HALO-(AC)³ campaign, and budget component analyses are currently under investigation and initially introduced in Wendisch et al. (2024, https://doi.org/10.5194/egusphere-2024-783). For the sake of brevity, we do not provide precise specifications in our manuscript, as all the information can be extracted from Wendisch et al. (2021).

*L59 What does "It" refer to? I guess it remains unclear at point what you mean with "individual moisture and wind variability"*

**Response/Modification (L59f):** We rephrased: **"We include a) […], a concretisation of the need for supplementary measurements based on spatial variability of moisture and wind."**

*L67-71: These lines were revised since the last time but did not change for the better. Please revise a) "estimates (…) provide an initial estimate" and b) "spacing between the sonde spacing (…).Instead of referring four times to a "spacing", please give a value used in Ralph et al.*

**Response/Modifications (L67ff):** We rephrased (a) as: **"Accurate airborne TIVT, juxtaposed for two separate AR cross-section legs, gives an initial estimate of IVT divergence in between both legs."** We rephrased L69f (b) as: **"When doubling the initial sonde spacing, which averaged about 80 km, by reducing the number of sondes included, they found a mean deviation of at least 5% for TIVT."**

*L94-97: I still think you can be more specific about the findings in the referred papers and identify results of relevance to your work.*

**Response/Modification (L94ff):** We rephrased the corresponding sentences as: **"Comparing two reanalyses, Guan et al. (2020) found differences in IVT divergence that, in the AR centres, reach up to 30 % the magnitude of IVT divergence itself. Norris et al. (2020) determined IVT divergence in a mid-latitude AR from dropsondes that allows interpreting the discrepancies of IVT divergence in ARs found by Guan et al. (2020). In particular, Norris et al. (2020) point to the large variability of IVT divergence at spatial scales of 50 km. This also has implications for sonde-based sampling best practices."**

*L174: What does "synoptic compositions lead to AR dispersion" mean?*

**Response/Modification (L175):** We rephrased: **"The synoptic compositions distribute the ARs over the North Atlantic and Arctic Ocean."**

*L188-190: It is not clear how these findings are relevant and what Ralph et al. (2017) found. Instead I suggest the authors describe the conceptual model that the refer to in L191 and L194.*

**Response/Modification (L187ff):** We readapted the connection of our statements as:
**"Figure 2 shows some of the features that we know about mid-latitude ARs. Over the course of various research flight campaigns over the eastern Pacific, Ralph et al. (2017) have developed a conceptual scheme for the cross-sections of ARs. This scheme includes, for example, a low-level jet (LLJ), which is a strong low-level wind corridor (Ralph et al., 2004; Demirdjian et al., 2020). For the windy arctic AR events, e.g. AR3 and AR5, we detect the presence of LLJs stronger than 25 m s−1. The LLJ is located at a height of about 900 hPa, slightly lower than the mean height reported by Cobb et al.**

**(2021) for mid-latitude ARs. As another feature of the AR cross-section scheme, Ralph et al. (2004) and Cordeira et al. (2013) found a horizontally slanted vertical structure of moisture transport in mid-latitude ARs from dropsondes and reanalyses. Ralph et al. (2017) confirmed the vertical interaction between the upper-level jet and the LLJ as a dominant effect for AR moisture transport. In Fig. 2, this is particularly evident for AR5. Here, moist air masses residing in the cyclonic warm conveyor belt are lifted over the cold front sector. The downward intrusion of air from the upper-level jets on the western flank causes the slanted structure of moisture transport.***"***

*L195/196: I do not understand the sentence. What means "yield" in this context? "but missing there" where?*

**Response/Modification (L198ff):** We have rephrased: **"In arctic ARs other than AR5, we find less agreement with the conceptual AR schemes. This is the case for the vertical structure of moisture, or the presence and the intensity of the LLJ, which is only strongly distinctive in AR1, AR3, AR5, AR7, but absent in all other cases."**

*L213/214: The causality of the first sentence is unclear. "specific corridors" - I thought the AR itself is a corridor.*

**Response/Modification (L216ff):** We have exchanged "corridor" by "area" at this point. Accordingly, we reformulated: **"To evaluate the airborne observability of ∇IVT within arctic ARs, we search for a suitable flight pattern. Such a pattern must capture well certain areas of the ARs. Flight tracks that enclose such areas, like circles, best allow divergence calculations and are often used for such purposes (e.g. Bony and Stevens, 2019)."**

We now use the term "flight corridor" to briefly refer to the area of the AR that is captured by the flight pattern. We find appropriate to call this region a flight corridor, even though the AR itself is occasionally described as a corridor in literature, too. To prevent misunderstandings, we clearly define this term in **L222f** as: **"For the sake of brevity, we speak of an AR flight corridor in the following, when we mean the area of the AR that is captured by the flight pattern."**

*L225ff: The description "as ong as (…) Fig.3)" needs to be improved. What does it mean that you obtain the boundaries from the catalogue? What does "closer" means "less"? What is the "final distance"? What is that "visual inspection" about, which would also be done during a real field campaign flight planning process?*

**Response/Modification (L228ff):** We have reformulated the paragraph as follows: **"We place the AR flight corridors close to the sea-ice edge. We orientate the cross-section legs orthogonal to the major IVT direction and extend the legs such that they transect the entire AR. One requirement is that the transect, and thus the lateral AR extension, is completely over open-ocean or sea-ice, so that we neglect landfalling regions of ARs. To obtain the spatial extent of the AR for a given case, we consider the shapes of the ARs defined in the AR catalogue of Guan et al. (2020). The meridional distance between the two cross-section legs is assured to be larger than 200 km but closer than half the lateral AR width. The decision for the placement and meridional distance of the cross-section legs is based on visual inspection of the reanalysis taking in particular the movement of the IVT filament into account. "**

For the real flight campaign, we recommend to include IVT forecasts into the flight planning and to look for AR like structures with IVT exceeding 100 kgm$^{-1}$s$^{-1}$. However, in this section of the manuscript, we do not find it useful to provide such information. We think that they rather belong in the outlook.

*L235-237 and L239/240: I do not get the need of the 1 Hz resolution that is then reduced to 1 minute. I suggest deleting this information as it is confusing and seems not to be relevant here.*

**Response/Modification (L241ff)**: First, we only describe the flight performance, i.e how the aircraft position on its flight is defined. We rephrased: **"Based on the aforementioned flight performance, the aircraft location is represented in 1Hz resolution, as common remote-sensing products on research aircraft (e.g. Mech et al., 2014; Konow et al., 2019), that can complement dropsonde data, have a comparable time-resolution and require the information of the aircraft location."**

Then, we come to the point on how to project the meteorological data onto the "aircraft". For this, the reanalyses are interpolated in time onto 1min as we find this as good compromise to not overfit the data. From this coarser time evolving representation of the AR, we spatially interpolate via harvesine distances.

*L241: Does "evolving representation of met. values" mean the temporally interpolated values?*

**Response/Modification (L248f)**: Yes, we changed it to: **"This spatio-temporally interpolated representation, […]"**

*L242: Clarify what "assure model physics" means and delete "as".*

**Response/Modification**: We meant that, while the models are tuned for consistency at their model output, this cannot be assured once we interpolate linearly among the quantities. Still, we agree that this sentence is confusing and not very relevant here. We have removed it for the sake of clarity.

*L249: "Our (…) position" - What does that mean?*

**Response/Modification (L255f)**: We rephrased: "**The synthetic sondes profile the atmosphere fully-vertically from their release location. IVT is thus defined as the integral of the fixed vertical column from the respective reanalysis cells."**

*L246-265 You explain two methods and should explain the reader which one is used or if they are compared, etc.:*

**Response/Modification**: We added in **L259f** and **L266f**: **"To derive ∇IVT in AR flight corridors from sondes, we compare two approaches. The first one is a simplified approximation based on the derivation of the TIVT […] Given this limitation, Lenschow et al. (2007) alternatively suggests a regression-approach, which marks our second approach."**

*Fig.5 If the x-axis runs from west to east, why is the cold post-frontal sector to the right?*

**Response/Modification**: We are very sorry for this wrong statement. In fact, we wanted to say from east to west and corrected it accordingly.

*L300: I suggest deleting "where our (…) ends". Change "our" to "the". Be more specific about "only categorize the prevailing air masses of an AR".*

**Response/Modification (L307ff)**: We rephrased: *"**Both thresholds form the outer boundaries of the AR and of the pre-frontal and post-frontal sectors. Note that the sector terminologies are only a generalised categorisation of the surrounding air masses in the vicinity of the cold front inside ARs, but should not be viewed as a synoptic cold front identification."**

*L302: It is unclear what "we (…) sectors" means. It is still unclear whether the release locations are selected at one particular time step (L305-307 is not clear on that) and how this is done in real case scenarios?*

**Response/Modification (310f):** In Sect. 2.4, we mention that we place the sondes along the airborne continuous AR representation. We added in L300: **"All threshold criteria are applied to the continuous AR representation along the flight time, as in a realistic post-analysis from real research flights."**

In **L312ff**, we remind for this time perspective by: **"Using […], we locate the sondes along the flight time in a way that three sondes each in the in- and outflow cross-section span one out of the three pre-defined frontal sectors (Fig.6), and calculate its IVT divergence, respectively. Note there is probably more variation in the actual release position in real flights due to forecast uncentainities, even when the releases are planned using the threshold criteria based on forecasted IVT. However, we here stick to these pre-defined locations for comparability between the AR cases. "**

*L312: What is "plane perspective" exactly meaning? Is this the temporally interpolated data?*

**Response/Modification (L325f):** We rephrased: **"First, we stick to the vertically-integrated perspective."** The second question is addressed in the following comment.

*L315: Change "our" to "the". Do you mean the temporally interpolated representation in CARRA?*

**Response/Modification (L327f):** We rephrased: **"Since we lack real observations, we declare the spatio-temporally interpolated AR representation in CARRA as the truth."**

*L335: Be more precise about what you want to say with "location and horizontal patterns agree quite well".*

**Response/Modification (L350):** We rephrased: **"In our comparison of CARRA with ERA5, the location of the ARs and the horizontal IVT patterns match quite well (not shown)."**

*L340: "(…) before dedicated observations." I don't get the coherence in this sentence*
**Response/Modification:** We have deleted this part.

*L341: I still find it confusing that six dropsondes are used here. This and also the motivation for using equidistant dropsondes should be clarified.*

**Response/Modification (L355f):** We updated Fig. 7 and now included seven sondes for better consistency. Furthermore, in this part we are not interested in any sector specifications and only consider the overall moisture transport variability inside the AR. For this reason, we reformulated: **"When the observational focus is on the IVT variability, in general, not on sector-based characteristics, we can simply place the sondes equidistantly."**

*L350 and 352: Before, you talked about "continuous" in a different context. Maybe better to specify this here as 1-min resolution profiles, if that is the case.*

**Response/Modification:** In **L366**, we added a reference to the section to make clear, that *continuous* here also means our time-evolving flight representation:

**"[…] by comparing their TIVT values to a control case which is based on the continuous representation of the AR cross-sections along the flight (Sect. 2.3)."**

We also updated our definition in **L248f**: **"This spatio-temporally interpolated representation of meteorological values and AR characteristics along the flight will from now on be referred to as "continuous AR representation".**

*.*

*L358f: What is the maximum spacing recommended for mid-latitudes? Would be good to know. See also comment referring to your conclusion.*

**Response/Modification (L390f):** They did not have a clear maximum spacing recommendation. However, their mean spacing was about 80 km, which we now explicitly refer to in the final statement as: **"[…], which is less than mean sonde spacings of 80 km as conducted for mid-latitude ARs in Ralph et al. (2017)."**

*L380: Revise "(…) dries with height. The height decreasing moisture (…)". Actually moisture decreases with altitude due to the decreasing temperatures. Doesn't a drying imply a temporal change?*

**Response/Modification (L395f):** We rephrased: **"[..] while moisture decreases. The decrease of moisture with altitude leads to a decline of moisture transport."**

*L388: This sentence implies that a more homogeneous distribution at upper levels implies a weaker wind intensification with height? Does that make sense?*

**Response/Modification (L403ff):** We simplified the phrasing to: **"Above the local wind maximum, the increase of wind speed with height is less than in sub-tropic/mid-latitude cases. Ralph et al. (2005) and Cobb et al. (2021) report on a stronger increase with height due to a more intense upper level jet."**

*L425: Although these terminologies are established and I understand that the fluctuations need to be coherent, I do not get why transport by the mean quantities is called non-coherent?*
*You should comment about whether the variability in CARRA might still be lower than in observations and thus is underestimating this effect of fluctuations. I guess it would need a spectral analysis to see what scales are dominating the covariance and compare this for observations and simulations?*

**Response/Modification (L444):** We have included a remark to make sure for the reader that we cannot fully answer this question with a reanalysis where we lack observations:
**"Assuming that CARRA resolves the scales of dominant fluctuations, […]"**

*L456: Wouldn't it be better to talk about "fluctuations" instead of "patterns"*

**Response/Modification:** Since the fluctuations are spatially clustered, we find pattern more appropriate.

*L475: What are "calm air masses"? Now it is more confusing to understand Fig. 12 as it is different to Figure 5 regarding the axis orientation. In addition, it is confusing me where the post frontal cold sector actually is (see comment to Fig.5).*

**Response/Modification (L493):** See comment for Fig. 5, they are in the same axis orientation. All cross-sections are shown in flight direction. We rephrased: **"The steep post-frontal IVT decline in AR2 and AR7 results from weak low-level winds west of the AR."**

*L488: What means "are suggesting"?*

**Response/Modification:** We found "to suggest" more appropriate as "to indicate", since we do not purely trust the TIVT-based explanations of divergence/convergence. We checked the sentences for correct grammar.

*L489f: What means "decided to use"?*

**Response/Modification (L506f):** Changed to: **"Therefore, we choose the regression approach [..] to diagnose"**

*L499: I do not understand what you mean with "most crucial"? Do you mean that the strongest divergence occurs in the cold sector? Be specific about where.*

**Response/Modification (L516):** We rephrased: *"The moisture transport divergence is strongest in the post-frontal sector with both signs."*

*L503: "locate at" change to "are located at". Be specific about "the heights"*

**Response/Modification:** We verified that we actually mean that they are NOT located at the dominant heights of wind and moisture, as moisture is highest in the marine boundary layer and winds peak in the low-level jet and the upper level jet. The dominant heights of wind and moisture have been described already, so that we here only describe the vertical characteristics of the divergence components.

*L507: Why "dominates" and where?*

**Response/Modification (L524f):** Our statement was misleading so that we rephrased: **"The vertically integrated moisture transport convergence (divergence) is highest in the cold post-frontal (warm pre-frontal) sector of AR3 (Fig. 13), while the post-frontal sector shows the strongest variations with height."**

*L524f: "When..." Consider revising the complicated sentence.*

**Response/Modification (L542f):** We rephrased: **"Comparing our synthetic results with the mid-latitude based airborne study of Norris et al. (2020), which used real dropsondes, we see the strength of real sondes in their high vertical resolution."**

*L526-528: You mean the vertical resolution of the model? What about missing variability in the model?*

**Response/Modification (L543f):** Yes, we clarified: **"Real dropsondes provide much larger vertical variability than CARRA."**

*L540: "as the AR spread out" – what does that mean?*

**Response/Modification (L558f):** We rephrased: **"However, the low-level mass convergence below 800 hPa [..] is often superimposed by mid- and upper-level mass divergence [...], where the AR widens causing directional divergence. "**

*L549: Please specify "frontal characteristics of CONV"*

**Response/Modification (L568ff):** We rephrased: **"In turn, the fact that CONV is found to be divergent in the prefrontal sector and core in arctic ARs contradicts the findings of Guan et al. (2020), who emphasised a dominant convergence of mass in and ahead of the AR-embedded front for mid-latitude ARs."**

*L585: Remind the reader what the "spatially continuous representation" is by adding the profile distance and temporal resolution.*

**Response/Modification (L602ff):** We added spatially explicitly since we were not seeking for temporal interpolation at this time. For better clarification, we rephrased: "**To quantify the deviations in IVT divergence due to non-instantaneous observations, we contrast the spatially continuous representation, which is the spatially interpolated CARRA data at the aircraft location for each point, in two temporal perspectives. This is done by establishing an instantaneous reference. The instantaneous reference is based on the spatially continuous airborne representation, but only for the CARRA output at the central hour of the flight, without interpolation in time. This can be thought of as a continuously sampling and infinitely fast aircraft, so we refer to it as the optimum airborne representation. We contrast the sector-based $\nabla IVT$ of this reference with the**

**one of the non-instantaneous continuous representation defined in Sect. 2.3 and analysed in the previous sections, which takes the flight time into account. "**

*L660: I wonder about the factor of 2 in the range of the spacing of the dropsondes. I would have expected you can be more precise. Maybe add the respective error ranges. Compare to the midlatitude value (see comment above).*

**Response/Modification (L684):** We give an updated recommendation of 100 +/- 20 km.

*L661: What does "rely on steepness" mean in the context of flight planning?*

**Response/Modification (L685f):** We found the "steepness" as to vague. With a reference to flight planning, we rephrased: **"The planning of sonde releases should thus rely on the forecasted gradients of IVT along the cross-sections.**

*L666: Delete "height and less at other height" and make clear if 0.5 is the highest mean correlation coefficient.*

**Response/Modification (L691f):** We rephrased: "**Moisture and wind in arctic ARs are only moderately correlated along the flight transects, with a maximum mean correlation of 0.5 at about 850 hPa height, but much less at other heights."**

*L670: What does "collocated sampling of (..) not of first priority" mean? Is that connected to the last sentence in L676. Could you comment on how supplementary observations would help?*

**Response/Modification (L695f):** We rephrased: **"We draw the conclusion that collocated sampling of wind and moisture is not a priority."** At the end of the bullet point, we give information on how supplementary observations may help to estimate IVT variability. Furthermore, when listing the implications for future flight planning in **L739f**, we added: **"In addition to the sondes, complementary measurements of moisture should be prioritised due to its higher variability, and its advection dominating moisture transport divergence."**

*L676ff: Avoid "is expected" and write what you have done. What are "Arctic AR edges? Are there more than one "post frontal sector"? Be specific about the dominating process and the overall contribution of the individual sectors to the moisture budget.*

**Response/Modification (L700ff):** We rephrased this bullet point for more specific clarification: **"Contrasting the ingoing and outgoing *TIVT* of the AR flight corridor using the two cross-sections suggests an overall divergence in *IVT*. However, the ARs show different characteristics of IVT divergence ($\nabla$IVT) in specific sectors across the AR-embedded cold front, especially when we decompose $\nabla$IVT into moisture advection (ADV) and mass convergence (CONV). The advection term contributes most to the entire moisture transport divergence across the AR, especially in the pre- and post-frontal sectors (Fig. 15). The pre-frontal AR sector contributes via moist advection, while the post-frontal sector generally shows dry advection. Across the front, the total contribution of IVT divergence to the moisture budget is up to +1 mm h$^{-1}$ (pre-frontal moisture advection) to -2 mm h$^{-1}$ (post-frontal dry advection). This is slightly less than the magnitudes in mid-latitude ARs. However, in contrast to mid-latitude ARs, mass convergence is much less dominant in the arctic ARs apart from the post-frontal sector. Although the convergence of mass is generally dominant below 850 hPa, upper-level divergence often superimposes it. The advection term dominates at levels higher than 850 hPa. For the post-frontal sector, this is mostly dry advection of cold airmasses west of the AR."**

*L687ff: I think I still do not understand this result completely. What are potential explanations for the large differences between instantaneous and non-instantaneous for values Arctic AR? If you look at the total over all sectors - what is the difference? Does that relate to the northern location of the sections at the ice edge? How do these findings affect planning of flights (L630)? What can you learn from observations about IVT divergence?*

**Response/Modification (L718ff):** We put more focus on describing the overall differences between both samplings (which are basically the weaker gradients across the AR) and explain the reasons more thoroughly as: **"The AR instationarity over flight time, including a displacement of the AR not necessarily along the moisture transport direction, deteriorates the results more than undersampling and leads to an underestimation of the sector-based gradients in moisture transport divergence. In fact, the pre-frontal moisture advection and the post-frontal sector divergence (from *ADV* and *CONV*) are stronger than assumed by sondes. Sonde-based values deviate the most in the post-frontal cold sector, where the AR has stronger gradients in moisture and winds than in the pre-frontal sector. The eastward displacement of the AR during flight deteriorates the post-frontal wind and moisture conditions seen from the sondes. Over flight time, the northern cross-section becomes drier due to dry intrusions of air masses from west of the AR. The emerging negative meridional moisture gradient between both cross-sections, that is then seen by the sondes, suggests a meridional advection of moisture that partially compensates the actual dry advection. This misrepresents sonde-based ADV and frequently causes deviations higher than 50% of the actual values."**
How this affects flight planning is further described in **L733ff**.

*L709: "Still feasible" in what sense?*

**Response/Modification (L738f):** We have rephrased: **"The optimal and practically realisable strategy is to have collocated flights by two aircraft, with both cross-sections not far apart and sampled simultaneously."**